# Direction-oriented Multi-objective Learning: Simple and Provable Stochastic Algorithms

**Peiyao Xiao[1], Hao Ban[2], Kaiyi Ji[1]***

[1]Department of CSE, University at Buffalo

[2] Zuoyi Technology

{peiyaoxi,kaiyiji}@buffalo.edu, bhsimon0810@gmail.com

## Abstract

Multi-objective optimization (MOO) has become an influential framework in many machine learning problems with multiple objectives such as learning with multiple criteria and multi-task learning (MTL). In this paper, we propose a new direction-oriented multi-objective formulation by regularizing the common descent direction within a neighborhood of a direction that optimizes a linear combination of objectives such as the average loss in MTL or a weighted loss that places higher emphasis on some tasks than the others. This formulation includes GD and MGDA as special cases, enjoys the direction-oriented benefit as in CAGrad, and facilitates the design of stochastic algorithms. To solve this problem, we propose Stochastic Direction-oriented Multi-objective Gradient descent (SDMGrad) with simple SGD type of updates, and its variant SDMGrad-OS with an efficient objective sampling. We develop a comprehensive convergence analysis for the proposed methods with different loop sizes and regularization coefficients. We show that both SDMGrad and SDMGrad-OS achieve improved sample complexities to find an $\epsilon$-accurate Pareto stationary point while achieving a small $\epsilon$-level distance toward a conflict-avoidant (CA) direction. For a constant-level CA distance, their sample complexities match the best known $\mathcal{O}(\epsilon^{-2})$ without bounded function value assumption. Extensive experiments show that our methods achieve competitive or improved performance compared to existing gradient manipulation approaches in a series of tasks on multi-task supervised learning and reinforcement learning. Code is available at https://github.com/ml-opt-lab/sdmgrad.

## 1 Introduction

In recent years, multi-objective optimization (MOO) has drawn intensive attention in a wide range of applications such as online advertising [1], hydrocarbon production [2], autonomous driving [3], safe reinforcement learning [4], etc. In this paper, we focus on the stochastic MOO problem, which takes the formulation of

$$\min_{\theta \in \mathbb{R}^m} \boldsymbol{L}(\theta) := (\mathbb{E}_\xi[L_1(\theta; \xi)], \mathbb{E}_\xi[L_2(\theta; \xi)], ..., \mathbb{E}_\xi[L_K(\theta; \xi)]), \tag{1}$$

where $m$ is the parameter dimension, $K \geq 2$ is the number of objectives and $L_i(\theta) := \mathbb{E}_\xi L_i(\theta; \xi)$. One important example of MOO in eq. (1) is the multi-task learning (MTL) [5, 6], whose objective

---

*First two authors contributed equally.

37th Conference on Neural Information Processing Systems (NeurIPS 2023).

| Algorithms | Batch size | Nonconvex | Bounded function value | Sample complexity | CA distance |
|---|---|---|---|---|---|
| SMG [11] | $\mathcal{O}(\epsilon^{-2})$ | ✗ | ✗ | $\mathcal{O}(\epsilon^{-4})$ | N/A |
| CR-MOGM [12] | $\mathcal{O}(1)$ | ✓ | ✓ | $\mathcal{O}(\epsilon^{-2})$ | N/A |
| MoCo [13] | $\mathcal{O}(1)$ | ✓ | ✓ | $\mathcal{O}(\epsilon^{-2})$ | N/A |
| MoDo [14] | $\mathcal{O}(1)$ | ✓ | ✗ | $\mathcal{O}(\epsilon^{-2})$ | $\mathcal{O}(1)$ |
| SDMGrad (Theorem 3) | $\mathcal{O}(1)$ | ✓ | ✗ | $\mathcal{O}(\epsilon^{-2})$ | $\mathcal{O}(1)$ |
| SDMGrad-OS (Theorem 4) | $\mathcal{O}(1)$ | ✓ | ✗ | $\mathcal{O}(\epsilon^{-2})$ | $\mathcal{O}(1)$ |
| MoCo [13] | $\mathcal{O}(1)$ | ✓ | ✗ | $\mathcal{O}(\epsilon^{-10})$ | $\mathcal{O}(\epsilon)$ |
| MoDo [14] | $\mathcal{O}(1)$ | ✓ | ✗ | $\mathcal{O}(\epsilon^{-8})$ | $\mathcal{O}(\epsilon)$ |
| SDMGrad (Corollary 1) | $\mathcal{O}(1)$ | ✓ | ✗ | $\mathcal{O}(\epsilon^{-6})$ | $\mathcal{O}(\epsilon)$ |
| SDMGrad-OS (Theorem 2) | $\mathcal{O}(1)$ | ✓ | ✗ | $\mathcal{O}(\epsilon^{-6})$ | $\mathcal{O}(\epsilon)$ |

Table 1: Comparison of different algorithms for stochastic MOO problem to achieve an $\epsilon$-accurate Pareto stationary point. MoDo [14] is a concurrent work. Definition of CA distance can be found in Definition 2. $\mathcal{O}(\cdot)$ omits logarithm factors.

function is often regarded as the average loss over $K$ objectives.

$$\theta^* = \arg \min_{\theta \in \mathbb{R}^m} \left\{ \bar{L}(\theta) \triangleq \frac{1}{K} \sum_{i=1}^{K} L_i(\theta) \right\}, \tag{2}$$

where $L_i(\theta)$ is the loss function for task $i$. However, solving the MOO problem is challenging because it can rarely find a common parameter $\theta$ that minimizes all individual objective functions simultaneously. As a result, a widely-adopted target is to find the *Pareto stationary point* at which there is no common descent direction for all objective functions. In this context, a variety of gradient manipulation methods have been proposed, including the multiple gradient descent algorithm (MGDA) [7], PCGrad [8], CAGrad [9] and Nash-MTL [10]. Among them, MGDA updates the model parameter $\theta$ using a time-varying multi-gradient, which is a convex combination of gradients for all objectives. CAGrad further improves MGDA by imposing a new constraint on the difference between the common descent direction and the average gradient to ensure convergence to the minimum of the average loss of MTL. However, these approaches mainly focus on the deterministic case, and their stochastic versions still remain under-explored.

Stochastic MOO has not been well understood except for several attempts recently. Inspired by MGDA, [11] proposed stochastic multi-gradient (SMG), and established its convergence with convex objectives. However, their analysis requires the batch size to increase linearly. In the more practical nonconvex setting, [12] proposed a correlation-reduced stochastic multi-objective gradient manipulation (CR-MOGM) to address the non-convergence issue of MGDA, CAGrad and PCGrad in the stochastic setting. However, their analysis requires a restrictive assumption on the bounded function value. Toward this end, [13] recently proposed a method named MoCo as a stochastic counterpart of MGDA by introducing a tracking variable to approximate the stochastic gradient. However, their analysis requires that the number $T$ of iterations is big at an order of $K^{10}$, where $K$ is the number of objectives, and hence may be unsuitable for the scenario with many objectives. Thus, it is highly demanding but still challenging to develop efficient stochastic MOO methods with guaranteed convergence in the nonconvex setting under mild assumptions.

## 1.1 Our Contributions

Motivated by the limitations of existing methods, we propose Stochastic Direction-oriented Multi-objective Gradient descent (SDMGrad), which is easy to implement, flexible with a new direction-oriented objective formulation, and achieves a better convergence performance under milder assumptions. Our specific contributions are summarized as follows.

**New direction-oriented formulation.** We propose a new MOO objective (see eq. (6)) by regularizing the common descent direction $d$ within a neighborhood of a direction $d_0$ that optimizes a

linear combination $L_0(\theta)$ of objectives such as the average loss in MTL or a weighted loss that places higher emphasis on some tasks than the others. This formulation is general to include gradient descent (GD) and MGDA as special cases and takes a similar direction-oriented spirit as in CAGrad [9]. However, different from the objective in CAGrad [9] that enforces the gap between $d$ and $d_0$ to be small via constraint, our regularized objective is easier to design near-unbiased multi-gradient, which is crucial in developing provably convergent stochastic MOO algorithms.

**New stochastic MOO algorithms.** The proposed SDMGrad is simple with efficient stochastic gradient descent (SGD) type of updates on the weights of individual objectives in the multi-gradient and on the model parameters with (near-)unbiased (multi-)gradient approximations at each iteration. SDMGrad does not require either the linearly-growing batch sizes as in SMG or the extra tracking variable for gradient estimation as in MoCo. Moreover, we also propose SDMGrad-OS (which refers to SDMGrad with objective sampling) as an efficient and scalable counterpart of SDMGrad in the setting with a large number of objectives. SDMGrad-OS samples a subset of data points and objectives simultaneously in all updates and is particularly suitable in large-scale MTL scenarios.

**Convergence analysis and improved complexity.** We provide a comprehensive convergence analysis of SDMGrad for stochastic MOO with smooth nonconvex objective functions. For a constant-level regularization parameter $\lambda$ (which covers the MGDA case), when an $\epsilon$-level conflict-avoidant (CA) distance (see Definition 2) is required, the sample complexities (i.e., the number of samples to achieve an $\epsilon$-accurate Pareto stationary point) of the proposed SDMGrad and SDMGrad-OS improve those of MoCo [13] and MoDo [14] by an order of $\epsilon^{-4}$ and $\epsilon^{-2}$ (see Table 1), respectively. In addition, when there is no requirement on CA distance, the sample complexities of SDMGrad and SDMGrad-OS are improved to $\mathcal{O}(\epsilon^{-2})$, matching the existing best result in the concurrent work [14]. For an increasing $\lambda$, we also show this convergent point reduces to a stationary point of the linear combination $L_0(\theta)$ of objectives.

**Promising empirical performance.** We conduct extensive experiments in multi-task supervised learning and reinforcement learning on multiple datasets and show that SDMGrad can achieve competitive or improved performance compared to existing state-of-the-art gradient manipulation methods such as MGDA, PCGrad, GradDrop, CAGrad, IMTL-G, MoCo, MoDo, Nash-MTL and FAMO, and can strike a better performance balance on different tasks. SDMGrad-OS also exhibits a much better efficiency than SDMGrad in the setting with a large number of tasks due to the efficient objective sampling.

## 2 Related Works

**Multi-task learning.** One important application of MOO is MTL, whose target is to learn multiple tasks with possible correlation simultaneously. Due to this capability, MTL has received significant attention in various applications in computer vision, natural language process, robotics, and reinforcement learning [5, 15, 16, 17]. A group of studies have focused on how to design better MTL model architectures. For example, [18, 19, 20] enhance the MTL models by introducing task-specific modules, an attention mechanism, and different activation functions for different tasks, respectively. Another line of research aims to learn a bunch of smaller models of local tasks split from the original problem, which are then aggregated into a single model via knowledge distillation [21]. Recent several works [22, 23] have explored the connection between MTL and gradient-based meta-learning. [24] and [25] highlight the importance of task grouping and unrelated tasks in MTL, respectively. This paper focuses on a single model by learning multiple tasks simultaneously with novel model-agnostic SDMGrad and SDMGrad-OS methods.

**Gradient-based MOO.** Various gradient manipulation methods have been developed to learn multiple tasks simultaneously. One popular class of approaches re-weight different objectives based on uncertainty [26], gradient norm [27], and training difficulty [28]. MOO-based approaches have received more attention due to their principled designs and training stability. For example, [6] viewed MTL as a MOO problem and proposed an MGDA-type method for optimization. A class of approaches has been proposed to address the gradient conflict problem. Among them, [8] proposed PCGrad by projecting the gradient direction of each task on the norm plane of other tasks. GradDrop randomly dropped out highly conflicted gradients [29], and RotoGrad rotated task gradients to alleviate the conflict [30]. [9] proposed CAGrad by constraining the common direction direction within a local region around the average gradient. These approaches mainly focus on the determin-

istic setting. In the stochastic case, [13] proposed MoCo as a stochastic counterpart of MGDA, and provided a comprehensive convergence and complexity analysis. More recently, a concurrent work [14] analyzed a three-way trade-off among optimization, generalization, and conflict avoidance, providing an impact on designing the MOO algorithm. In addition, [31] found that scalarization SGD could be incapable of fully exploring the Pareto front compared with MGDA-variant methods. In this paper, we propose a new stochastic MOO method named SDMGrad, which benefits from a direction-oriented regularization and an improved convergence and complexity performance.

**Concurrent work.** A concurrent work [14] proposed a multi-objective approach named MoDo, which uses a similar double sampling strategy (see Section 4.2). However, there are still several differences between this work and [14]. First, our method benefits from a direction-oriented mechanism and is general to include MGDA and CAGrad as special cases, whereas MoDo [14] does not have such features. Second, MoDo takes a single-loop structure, whereas our SDMGrad features a double-loop scheme with an improved sample complexity. Third, [14] focuses more on the trade-off among optimization, generalization and conflict-avoidance, whereas our work focues more on the optimization efficiency and performance balance in theory and in experiments.

## 3 Preliminaries

### 3.1 Pareto Stationarity in MOO

Differently from single-objective optimization, MOO aims to find points at which all objectives cannot be further optimized. Consider two points $\theta$ and $\theta'$. It is claimed that $\theta$ dominates $\theta'$ if $L_i(\theta) \leq L_i(\theta')$ for all $i \in [K]$ and $L(\theta) \neq L(\theta')$. In the general nonconvex setting, MOO aims to find a Pareto stationary point $\theta$, at which there is no common descent direction for all objectives. In other words, we say $\theta$ is a Pareto stationary point if range$(\nabla L(\theta)) \cap (-\mathbb{R}_{++}^K) = \emptyset$ where $\mathbb{R}_{++}^K$ is the positive orthant cone.

### 3.2 MGDA and Its Stochastic Variants

**Deterministic MGDA.** The deterministic MGDA algorithm was first studied by [7], which updates the model parameter $\theta$ along a multi-gradient $d = \sum_{i=1}^K w_i^* g_i(\theta) = G(\theta)w^*$, where $G(\theta) = (g_1(\theta), g_2(\theta), ..., g_K(\theta))$ are the gradients of different objectives, and $w^* := (w_1^*, w_2^*, ..., w_K^*)^T$ are the weights of different objectives obtained via solving the following problem.

$$w^* \in \arg\min_w \|G(\theta)w\|^2 \quad s.t. \ w \in \mathcal{W} := \{w \in \mathbb{R}^K | \mathbf{1}^T w = 1, w \geq 0\}, \tag{3}$$

where $\mathcal{W}$ is the probability simplex. The deterministic MGDA and its variants such as PCGrad and CAGrad have been well studied, but their stochastic counterparts have not been understood well.

**Stochastic MOO algorithms.** SMG [11] is the first stochastic variant of MGDA by replacing the full gradient $G(\theta)$ in eq. (3) by its stochastic gradient $G(\theta; \xi) := (g_1(\theta; \xi), ..., g_K(\theta; \xi))$, and updates the model parameters $\theta$ along the direction given by

$$d_\xi = G(\theta; \xi)w_\xi^* \quad s.t. \ w_\xi^* \in \arg\min_w \|G(\theta; \xi)w\|^2.$$

However, this direct replacement can introduce **biased** multi-gradient estimation, and hence SMG required to increase the batch sizes linearly with the iteration number. To address this limitation, [13] proposed MoCo by introducing an additional tracking variable $y_{t,i}$ as the stochastic estimate of the gradient $\nabla L_i(\theta)$, which is iteratively updated via

$$y_{t+1,i} = \Pi_{\mathcal{Y}_i}\big(y_{t,i} - \beta_t(y_{t,i} - h_{t,i})\big), i = 1, 2, ..., K, \tag{4}$$

where $\beta_t$ is the step size, $\Pi_{\mathcal{Y}_i}$ denotes the projection on a bounded set $\mathcal{Y}_i = \{y \in \mathbb{R}^m | \|y\| \leq C_{y,i}\}$ for some constant $C_{y,i}$, and $h_{t,i}$ denotes stochastic estimator of $\nabla L_i(\theta)$ at t-th iteration. Then, MoCo was shown to achieve an asymptotically unbiased multi-gradient, but with a relatively strong assumption that the number $T$ of iterations is much larger than the number $K$ of objectives. Thus, it is important but still challenging to develop provable and easy-to-implement stochastic MOO algorithms with mild assumptions.

# 4 Our Method

We first provide a new direction-oriented MOO problem, and then introduce a new stochastic MOO algorithm named SDMGrad and its variant SDMGrad-OS with objective sampling.

## 4.1 Direction-oriented Multi-objective Optimization

MOO generally targets at finding a direction $d$ to maximize the minimum decrease across all objectives via solving the following problem.

$$\max_{d \in \mathbb{R}^m} \min_{i \in [K]} \left\{ \frac{1}{\alpha} (L_i(\theta) - L_i(\theta - \alpha d)) \right\} \approx \max_{d \in \mathbb{R}^m} \min_{i \in [K]} \langle g_i, d \rangle, \tag{5}$$

where the first-order Taylor approximation of $L_i$ is applied at $\theta$ with a small stepsize $\alpha$.

In some scenarios, the target is to not only find the above common descent direction but also optimize a specific objective that is often a linear combination $L_0(\theta) = \sum_i \widetilde{w}_i L_i(\theta)$ for some $\widetilde{w} \in \mathcal{W}$ of objectives. For instance, MTL often takes the averaged loss over tasks as the objective function, and every element in $\widetilde{w}$ will be set as $\frac{1}{K}$. In addition, it is quite possible that there is a preference for tasks. In this case, motivated by the framework in [32], we can regard $\widetilde{w}$ as a preference vector and tend to approach a preferred stationary point along this direction. To address this problem, we propose the following multi-objective problem formulation by adding an inner-product regularization $\lambda \langle g_0, d \rangle$ in eq. (5) such that the common descent direction $d$ stays not far away from a target direction $g_0 = \sum_i \widetilde{w}_i g_i$.

$$\max_{d \in \mathbb{R}^m} \min_{i \in [K]} \langle g_i, d \rangle - \frac{1}{2} \|d\|^2 + \lambda \langle g_0, d \rangle. \tag{6}$$

In the above eq. (6), the term $-\frac{1}{2} \|d\|^2$ is used to regularize the magnitude of the direction $d$ and the constant $\lambda$ controls the distance between the update vector $d$ and the target direction $g_0$.

*Compared to CAGrad.* We note that CAGrad also takes a direction-oriented objective but uses a different constraint-enforced formulation as follows.

$$\max_{d \in \mathbb{R}^m} \min_{i \in [K]} \langle g_i, d \rangle \quad \text{s.t.} \quad \|d - h_0\| \le c\|h_0\| \tag{7}$$

where $h_0$ is the average gradient and $c \in [0, 1)$ is a constant. To optimize eq. (7), CAGrad involves the evaluations of the product $\|h_0\|\|g_w\|$ and the relation $\frac{\|h_0\|}{\|g_w\|}$ with $g_w := \sum_i w_i g_i$, both of which complicate the designs of unbiased stochastic gradient/multi-gradient in the $w$ and $\theta$ updates. As a comparison, our formulation in eq. (6), as shown later, admits very simple and provable stochastic algorithmic designs, while still enjoying the direction-oriented benefit as in CAGrad.

To efficiently solve the problem in eq. (6), we then substitute the relation that $\min_{i \in [K]} \langle g_i, d \rangle = \min_{w \in \mathcal{W}} \langle \sum_{i=1}^{K} g_i w_i, d \rangle = \min_{w \in \mathcal{W}} \langle g_w, d \rangle$ into eq. (6), and obtain an equivalent problem as

$$\max_{d \in \mathbb{R}^m} \min_{w \in \mathcal{W}} \langle g_w + \lambda g_0, d \rangle - \frac{1}{2} \|d\|^2.$$

By switching min and max in the above problem, which does not change the solution due to the concavity in $d$ and the convexity in $w$, we finally aim to solve $\min_{w \in \mathcal{W}} \max_{d \in \mathbb{R}^m} \langle g_w + \lambda g_0, d \rangle - \frac{1}{2} \|d\|^2$, where the solution to the min problem on $w$ is

$$w_\lambda^* \in \arg \min_{w \in \mathcal{W}} \frac{1}{2} \|g_w + \lambda g_0\|^2, \tag{8}$$

and the solution to the max problem is $d^* = g_{w_\lambda^*} + \lambda g_0$.

*Connection with MGDA and GD.* It can be seen from eq. (8) that the updating direction $d^*$ reduces to that of MGDA whenwe set $\lambda = 0$, and is consistent with that of GD for $\lambda$ large enough. This consistency is also validated empirically in Appendix A.2 by varying $\lambda$.

## 4.2 Proposed SDMGrad Algorithm

The natural idea to solve the problem in eq. (8) is to use a simple SGD-type method. The detailed steps are provided in algorithm 1. At each iteration $t$, we run $S$ steps of projected SGD with warm-start initialization and with a double-sampling-based stochastic gradient estimator, which is unbiased by noting that

$$\mathbb{E}_{\xi,\xi'}[G(\theta_t;\xi)^T\big(G(\theta_t;\xi')w_{t,s} + \lambda g_0(\theta_t;\xi')\big)] = G(\theta_t)^T\big(G(\theta_t)w_{t,s} + \lambda g_0(\theta_t)\big),$$

where $\xi, \xi'$ are sampled data and $g_0(\theta_t) = G(\theta_t)\widetilde{w}$ denotes the orientated direction. After obtaining the estimate $w_{t,S}$, SDMGrad updates the model parameters $\theta_t$ based on the stochastic counterpart $G(\theta_t;\zeta)w_{t,S} + \lambda g_0(\theta_t;\zeta)$ of the direction $d^* = g_{w_\lambda^*} + \lambda g_0$ with a stepsize of $\alpha_t$. It can be seen that SDMGrad is simple to implement with efficient SGD type of updates on both $w$ and $\theta$ without introducing other auxiliary variables.

---

**Algorithm 1** Stochastic Direction-oriented Multi-objective Gradient descent (SDMGrad)

---

1: **Initialize:** model parameters $\theta_0$ and weights $w_0$
2: **for** $t = 0, 1, ..., T-1$ **do**
3:    **for** $s = 0, 1, ..., S-1$ **do**
4:       Set $w_{t,0} = w_{t-1,S}$ (warm start) and sample data $\xi, \xi'$
5:       $w_{t,s+1} = \Pi_{\mathcal{W}}\big(w_{t,s} - \beta_{t,s}[G(\theta_t;\xi)^T\big(G(\theta_t;\xi')w_{t,s} + \lambda g_0(\theta_t;\xi'))\big)]\big)$
6:    **end for**
7:    Sample data $\zeta$ and $\theta_{t+1} = \theta_t - \alpha_t\big(G(\theta_t;\zeta)w_{t,S} + \lambda g_0(\theta_t;\zeta)\big)$
8: **end for**

---

## 4.3 SDMGrad with Objective Sampling (SDMGrad-OS)

Another advantage of SDMGrad is its simple extension via objective sampling to the more practical setting with a large of objectives, e.g., in large-scale MTL. In this setting, we propose SDMGrad-OS by replacing the gradient matrix $G(\theta_t;\xi)$ in Algorithm 1 by a matrix $H(\theta_t;\xi)$ with randomly sampled columns, which takes the form of

$$H(\theta_t;\xi,\mathcal{S}) = \big(h_1(\theta_t;\xi), h_2(\theta_t;\xi), ..., h_K(\theta_t;\xi)\big), \tag{9}$$

where $h_i(\theta_t;\xi) = g_i(\theta_t;\xi)$ with a probability of $n/K$ or 0 otherwise, $\mathcal{S}$ corresponds to the randomness by objective sampling, and $n$ is the expected number of sampled objectives. Then, the stochastic gradient in the $w$ update is adapted to

$$\text{(Gradient on } w) \quad \frac{K^2}{n^2}H(\theta_t;\xi,\mathcal{S})^T\big(H(\theta_t;\xi',\mathcal{S}')w_{t,s} + \lambda h_0(\theta_t;\xi',\mathcal{S}')\big),$$

where $h_0(\cdot) = H(\cdot)\widetilde{w}$. Similarly, the updating direction $d = \frac{K}{n}H(\theta_t;\zeta,\widetilde{\mathcal{S}})w_{t,S} + \frac{K}{n}\lambda h_0(\theta_t;\zeta,\widetilde{\mathcal{S}})$. In the practical implementation, we first use *np.random.binomial* to sample a 0-1 sequence $s$ following a Bernoulli distribution with length $K$ and probability $n/K$, and then compute the gradient $g_i(\theta;\xi)$ of $G(\theta;\xi)$ only if the $i^{th}$ entry of $s$ equals to 1. This sampling strategy greatly speeds up the training with a large number of objectives, as shown by the reinforcement learning experiments in Section 6.3.

# 5 Main results

## 5.1 Definitions and Assumptions

We first make some standard definitions and assumptions, as also adopted by existing MOO studies in [11, 12, 13]. Since we focus on the practical setting where the objectives are nonconvex, algorithms are often expected to find an $\epsilon$-accurate Pareto stationary point, as defined below.

**Definition 1.** *We say $\theta \in \mathbb{R}^m$ is an $\epsilon$-accurate Pareto stationary point if $\mathbb{E}\big[\min_{w \in \mathcal{W}} \|g_w(\theta)\|^2\big] \leq \epsilon$ where $\mathcal{W}$ is the probability simplex.*

MGDA-variant methods find a direction $d(\theta)$ (e.g., $d^* = g_{w_\lambda^*} + \lambda g_0$ with $w_\lambda^*$ given by eq. (8)) that tends to optimize all objective functions simultaneously, which is called a conflict-avoidant (CA)

direction [14]. Thus, it is important to measure the distance between a stochastic direction estimate $\widehat{d}(\theta)$ and the CA direction, which we call as CA distance, as defined below.

**Definition 2.** $\|\mathbb{E}_{\widehat{d}(\theta)}[\widehat{d}(\theta)] - d(\theta)\|$ *denotes the CA distance.*

The following assumption imposes the Lipschitz continuity on the objectives and their gradients.

**Assumption 1.** *For every task* $i \in [K]$, $L_i(\theta)$ *is* $l_i$*-Lipschitz continuous and* $\nabla L_i(\theta)$ *is* $l_{i,1}$*-Lipschitz continuous for any* $\theta \in \mathbb{R}^m$.

We next make an assumption on the bias and variance of the stochastic gradient $g_i(\theta; \xi)$.

**Assumption 2.** *For every task* $i \in [K]$, *the gradient* $g_i(\theta; \xi)$ *is the unbiased estimate of* $g_i(\theta)$. *Moreover, the gradient variance is bounded by* $\mathbb{E}_\xi[\|g_i(\theta; \xi) - g_i(\theta)\|^2] \leq \sigma_i^2$.

**Assumption 3.** *Assume there exists a constant* $C_g > 0$ *such that* $\|G(\theta)\| \leq C_g$.

The bounded gradient condition in Assumption 3 is necessary to ensure the boundedness of the multi-gradient estimation error, as also adopted by [12, 13].

## 5.2 Convergence Analysis with Nonconvex Objectives

We first upper-bound the CA distance for our proposed method.

**Proposition 1.** *Suppose Assumptions 1-3 are satisfied. If we choose* $\beta_{t,s} = c/\sqrt{s}$ *and* $S > 1$, *then*

$$\|\mathbb{E}_{\zeta, w_{t,S}|\theta_t}[G(\theta_t; \zeta)w_{t,S} + \lambda g_0(\theta_t; \zeta)] - G(\theta_t)w_{t,\lambda}^* - \lambda g_0(\theta_t)\| \leq \sqrt{\left(\frac{2}{c} + 2cC_1\right)\frac{2 + log(S)}{\sqrt{S}}},$$

*where* $C_1 = \mathcal{O}(K(1 + \lambda))$ *and* $c$ *is a constant.*

Proposition 1 shows that CA distance is decreasing with the number $S$ of iterations on $w$ updates. Then, by selecting a properly large $S$, we obtain the following general convergence result.

**Theorem 1** (SDMGrad). *Suppose Assumption 1-3 are satisfied. Set* $\alpha_t = \alpha = \Theta((1 + \lambda)^{-1}K^{-\frac{1}{2}}T^{-\frac{1}{2}})$, $\beta_{t,s} = c/\sqrt{s}$, *where* $c$ *is a constant, and* $S = \Theta((1 + \lambda)^{-2}T^2)$. *Then the outputs of the proposed SDMGrad algorithm satisfy*

$$\frac{1}{T}\sum_{t=0}^{T-1}\mathbb{E}[\|G(\theta_t)w_{t,\lambda}^* + \lambda g_0(\theta_t)\|^2] = \widetilde{\mathcal{O}}((1 + \lambda^2)K^{\frac{1}{2}}T^{-\frac{1}{2}}), \tag{10}$$

*where* $\widetilde{\mathcal{O}}$ *omits the order of* $\log T$.

Theorem 1 establishes a general convergence guarantee for SDMGrad along the multi-gradient direction $d^* = g_{w_\lambda^*} + \lambda g_0$. Building on Theorem 1, we next show that with different choices of the regularization parameter $\lambda$, two types of convergence results can be obtained in the following two corollaries.

**Corollary 1** (Constant $\lambda$). *Under the same setting as in Theorem 1, choosing a constant-level* $\lambda > 0$, *we have* $\frac{1}{T}\sum_{t=0}^{T-1}\mathbb{E}[\|G(\theta_t)w_t^*\|^2] = \widetilde{\mathcal{O}}(K^{\frac{1}{2}}T^{-\frac{1}{2}})$, *where* $w_t^* \in \arg\min_{w \in \mathcal{W}}\frac{1}{2}\|G(\theta_t)w\|^2$. *To achieve an* $\epsilon$*-accurate Pareto stationary point, each objective requires* $\mathcal{O}(K^3\epsilon^{-6})$ *samples in* $\xi$ ($\xi'$) *and* $\mathcal{O}(K\epsilon^{-2})$ *samples in* $\zeta$, *respectively. Meanwhile, the CA distance takes the order of* $\widetilde{\mathcal{O}}(\epsilon)$.

Corollary 1 covers the MGDA case when $\lambda = 0$. In this setting, the sample complexity $\mathcal{O}(\epsilon^{-6})$ improves those of MoCo [13] and MoDo [14] by an order of $\epsilon^{-4}$ and $\epsilon^{-2}$, respectively, while achieving an $\epsilon$-level CA distance.

**Corollary 2** (Increasing $\lambda$). *Under the same setting as in Theorem 1 and choosing* $\lambda = \Theta(T^{\frac{1}{2}})$, *we have* $\frac{1}{T}\sum_{t=0}^{T-1}\mathbb{E}[\|g_0(\theta_t)\|^2] = \mathcal{O}(K^{\frac{1}{2}}T^{-\frac{1}{2}})$. *To achieve an* $\epsilon$*-accurate stationary point, each objective requires* $\mathcal{O}(K^2\epsilon^{-4})$ *samples in* $\xi$ ($\xi'$) *and* $\mathcal{O}(K\epsilon^{-2})$ *samples in* $\zeta$, *respectively. Meanwhile, the CA distance takes the order of* $\widetilde{\mathcal{O}}(\sqrt{K})$.

Corollary 2 analyzes the case with an increasing $\lambda = \Theta(T^{\frac{1}{2}})$. We show that SDMGrad converges to a stationary point of the objective $L_0(\theta) = \sum_i \widetilde{w}_i L_i(\theta)$ with an improved sample complexity, but with a worse constant-level CA distance. This justifies the flexibility of our framework that the $\lambda$ can balance the worst local improvement of individual objectives and the target objective $L_0(\theta)$.

**Convergence under objective sampling.** We provide a convergence analysis for SDMGrad-OS.

**Theorem 2** (SDMGrad-OS). *Suppose Assumption 1-3 are satisfied. Define* $\gamma = \frac{K}{n}$, $\alpha_t = \alpha = \Theta((1+\lambda^2)^{-\frac{1}{2}}\gamma^{-\frac{1}{2}}K^{-\frac{1}{2}}T^{-\frac{1}{2}})$, $\beta_{t,s} = c/\sqrt{s}$ *where c is a constant, and* $S = \Theta((1+\lambda^2)^{-2}\gamma^{-2}T^2)$. *Then, by choosing a constant $\lambda$, the iterates of the proposed SDMGrad-OS algorithm satisfy*

$$\frac{1}{T}\sum_{t=0}^{T-1}\mathbb{E}[\|G(\theta_t)w_t^*\|^2] = \widetilde{\mathcal{O}}(K^{\frac{1}{2}}\gamma^{\frac{1}{2}}T^{-\frac{1}{2}}).$$

Theorem 2 establishes the convergence guarantee for our SDMGrad-OS algorithm, which achieves a per-objective sample complexity of $\mathcal{O}(\epsilon^{-6})$ comparable to that of SDMGrad, but with a much better efficiency due to the objective sampling, as also validated by the empirical comparison in Table 4.

### 5.3 Lower sample complexity but constant-level CA distance

Without the requirement on the $\epsilon$-level CA distance, we further improve the sample complexity of our method to $\mathcal{O}(\epsilon^{-2})$, as shown in the following theorem.

**Theorem 3.** *Suppose Assumptions 1-3 are satisfied. Set* $S = 1$, $\alpha_t = \alpha = \Theta(K^{-\frac{1}{2}}T^{-\frac{1}{2}})$, $\beta_t = \beta = \Theta(K^{-1}T^{-\frac{1}{2}})$ *and $\lambda$ as constant. The iterates of the proposed SDMGrad satisfy*

$$\frac{1}{T}\sum_{t=0}^{T-1}\mathbb{E}[\|G(\theta_t)w_t^*\|^2] = \mathcal{O}(KT^{-\frac{1}{2}}).$$

Theorem 3 shows that to achieve an $\epsilon$-accurate Pareto stationary point, our method requires $T = \mathcal{O}(K^2\epsilon^{-2})$. In this case, each objective requires a number $\mathcal{O}(K^2\epsilon^{-2})$ of samples in $\xi(\xi')$ and $\zeta$.

**Convergence under objective sampling.** We next analyze the convergence of SDMGrad-OS.

**Theorem 4.** *Suppose Assumptions 1-3 are satisfied. Set* $S = 1$, $\gamma = \frac{K}{n}$, $\alpha_t = \alpha = \Theta(K^{-\frac{1}{2}}\gamma^{-\frac{1}{2}}T^{-\frac{1}{2}})$, $\beta_t = \beta = \Theta(K^{-1}\gamma^{-1}T^{-\frac{1}{2}})$ *and $\lambda$ as a constant. The iterates of the proposed SDMGrad-OS algorithm satisfy*

$$\frac{1}{T}\sum_{t=0}^{T-1}\mathbb{E}[\|G(\theta_t)w_t^*\|^2] = \mathcal{O}(K\gamma T^{-\frac{1}{2}}).$$

Theorem 4 shows that to achieve an $\epsilon$-accurate Pareto stationary point, our algorithm requires $T = \mathcal{O}(\gamma^2 K^2\epsilon^{-2})$, and each objective requires a number $\mathcal{O}(\gamma^2 K^2\epsilon^{-2})$ of samples in $\xi(\xi')$ and $\zeta$.

## 6 Experiments

In this section, we first describe the implementation details of our proposed methods. Then, we demonstrate the effectiveness of the methods under a couple of multi-task supervised learning and reinforcement settings. The experimental details and more empirical results such as the two-objective toy example, consistency with GD and MGDA, and ablation studies over $\lambda$ can be found in the Appendix A.

### 6.1 Practical Implementation

**Double sampling.** In supervised learning, double sampling (i.e., drawing two samples simultaneously for gradient estimation) is employed, whereas in reinforcement learning, single sampling is used because double sampling requires to visit the entire episode twice a time, which is much more time-consuming.

**Gradient normalization and rescale.** During the training process, the gradient norms of tasks may change over time. Thus, directly solving the objective in eq. (8) may trigger numerical problems.

Inspired by CAGrad [9], we normalize the gradient of each task and rescale the final update $d$ by multiplying a factor of $\frac{1}{1+\lambda}$ to stabilize the training.

**Projected gradient descent.** The computation of the projection to the probability simplex we use is the Euclidean projection proposed by [33], which involves solving a convex problem via quadratic programming. The implementation used in our experiments follows the repository in [34], which is very efficient in practice.

| Method | Segmentation | | Depth | | MR ↓ | $\Delta m\%$ ↓ |
|---|---|---|---|---|---|---|
| | mIoU ↑ | Pix Acc ↑ | Abs Err ↓ | Rel Err ↓ | | |
| STL | 74.01 | 93.16 | 0.0125 | 27.77 | | |
| LS | 75.18 | 93.49 | 0.0155 | 46.77 | 8.50 | 22.60 |
| SI | 70.95 | 91.73 | 0.0161 | 33.83 | 11.50 | 14.11 |
| RLW [35] | 74.57 | 93.41 | 0.0158 | 47.79 | 11.25 | 24.38 |
| DWA [36] | 75.24 | 93.52 | 0.0160 | 44.37 | 8.50 | 21.45 |
| UW [26] | 72.02 | 92.85 | 0.0140 | **30.13** | 7.75 | **5.89** |
| MGDA [7] | 68.84 | 91.54 | 0.0309 | 33.50 | 12.00 | 44.14 |
| PCGrad [8] | 75.13 | 93.48 | 0.0154 | 42.07 | 8.75 | 18.29 |
| GradDrop [29] | 75.27 | 93.53 | 0.0157 | 47.54 | 7.75 | 23.73 |
| CAGrad [9] | 75.16 | 93.48 | 0.0141 | 37.60 | 7.25 | 11.64 |
| IMTL-G [37] | 75.33 | 93.49 | 0.0135 | 38.41 | 5.25 | 11.10 |
| MoCo [13] | 75.42 | 93.55 | 0.0149 | 34.19 | 4.00 | 9.90 |
| MoDo [14] | 74.55 | 93.32 | 0.0159 | 41.51 | 10.75 | 18.89 |
| Nash-MTL [10] | **75.41** | **93.66** | **0.0129** | 35.02 | **2.75** | 6.82 |
| FAMO [38] | 74.54 | 93.29 | 0.0145 | 32.59 | 7.75 | 8.13 |
| SDMGrad | 74.53 | 93.52 | 0.0137 | 34.01 | 6.25 | 7.79 |

Table 2: Multi-task supervised learning on Cityscapes dataset.

## 6.2 Supervised Learning

For the supervised learning setting, we evaluate the performance on the Cityscapes [39] and NYU-v2 [40] datasets. The former dataset involves 2 pixel-wise tasks: 7-class semantic segmentation and depth estimation, and the latter one involves 3 pixel-wise tasks: 13-class semantic segmentation, depth estimation and surface normal estimation. Following the experimental setup of [9], we embed a MTL method MTAN [36] into our SDMGrad method, which builds on SegNet [41] and is empowered by a task-specific attention mechanism. We compare SDMGrad with Linear Scalarization (LS) which minimizes the average loss, Scale-invariant (SI) which minimizes the average logarithmic loss, RLW [35], DWA [36], UW [26], MGDA [7], PCGrad [8], GradDrop [29], CAGrad [9], IMTL-G [37], MoCo [13], MoDo [14], Nash-MTL [10], and FAMO [38]. Following [42, 9, 13, 10, 38], we compute two metrics reflecting the overall performance: **(1)** $\Delta \mathbf{m}\%$, the average per-task performance drop versus the single-task (STL) baseline $b$ to assess method $m$: $\Delta m\% = \frac{1}{K}\sum_{k=1}^{K}(-1)^{l_k}(M_{m,k} - M_{b,k})/M_{b,k} \times 100$, where $K$ is the number of metrics, $M_{b,k}$ is the value of metric $M_k$ obtained by baseline $b$, and $M_{m,k}$ obtained by the compared method $m$. $l_k = 1$ if the evaluation metric $M_k$ on task $k$ prefers a higher value and $0$ otherwise. **(2) Mean Rank (MR)**: the average rank of each method across all tasks.

We search the hyperparameter $\lambda \in \{0.1, 0.2, \cdots, 1.0\}$ for our SDMGrad method and report the results in Table 2 and Table 3. Each experiment is repeated 3 times with different random seeds and the average is reported. It can be seen that our proposed method is able to obtain better or comparable results than the baselines, and in addition, can strike a better performance balance on multiple tasks than other baselines. For example, although MGDA achieves better results on Surface Normal, it performs the worst on both Segmentation and Depth. As a comparison, our SDMGrad method can achieve more balanced results on all tasks.

## 6.3 Reinforcement Learning

For the reinforcement learning setting, we evaluate the performance on the MT10 benchmarks, which include 10 robot manipulation tasks under the Meta-World environment [43]. Following the experiment setup in [9, 13, 10], we adopt Soft Actor-Critic (SAC) [44] as the underlying training algorithm. We compare SDMGrad with Multi-task SAC [43], Multi-headed SAC [43], Multi-task SAC + Task Encoder [43], PCGrad [8], CAGrad [9], MoCo [13], Nash-MTL [10] and FAMO [38]. We search $\lambda \in \{0.1, 0.2, \cdots, 1.0\}$ and provide the success rate and average training time (in sec-

| Method | Segmentation | | Depth | | Surface Normal | | | | | MR ↓ | $\Delta m\%$ ↓ |
| | | | | | Angle Distance ↓ | | Within $t°$ ↑ | | | | |
| | mIoU ↑ | Pix Acc ↑ | Abs Err ↓ | Rel Err ↓ | Mean | Median | 11.25 | 22.5 | 30 | | |
|---|---|---|---|---|---|---|---|---|---|---|---|
| STL | 38.30 | 63.76 | 0.6754 | 0.2780 | 25.01 | 19.21 | 30.14 | 57.20 | 69.15 | | |
| LS | 39.29 | 65.33 | 0.5493 | 0.2263 | 28.15 | 23.96 | 22.09 | 47.50 | 61.08 | 11.33 | 5.59 |
| SI | 38.45 | 64.27 | 0.5354 | 0.2201 | 27.60 | 23.37 | 22.53 | 48.57 | 62.32 | 10.33 | 4.39 |
| RLW [35] | 37.17 | 63.77 | 0.5759 | 0.2410 | 28.27 | 24.18 | 22.26 | 47.05 | 60.62 | 13.89 | 7.78 |
| DWA[36] | 39.11 | 65.31 | 0.5510 | 0.2285 | 27.61 | 23.18 | 24.17 | 50.18 | 62.39 | 10.11 | 3.57 |
| UW [26] | 36.87 | 63.17 | 0.5446 | 0.2260 | 27.04 | 22.61 | 23.54 | 49.05 | 63.65 | 9.89 | 4.05 |
| MGDA[7] | 30.47 | 59.90 | 0.6070 | 0.2555 | **24.88** | **19.45** | 29.18 | **56.88** | **69.36** | 7.33 | 1.38 |
| PCGrad [8] | 38.06 | 64.64 | 0.5550 | 0.2325 | 27.41 | 22.80 | 23.86 | 49.83 | 63.14 | 10.44 | 3.97 |
| GradDrop [29] | 39.39 | 65.12 | 0.5455 | 0.2279 | 27.48 | 22.96 | 23.38 | 49.44 | 62.87 | 9.44 | 3.58 |
| CAGrad [9] | 39.79 | 65.49 | 0.5486 | 0.2250 | 26.31 | 21.58 | 25.61 | 52.36 | 65.58 | 6.44 | 0.20 |
| IMTL-G [37] | 39.35 | 65.60 | 0.5426 | 0.2256 | 26.02 | 21.19 | 26.20 | 53.13 | 66.24 | 5.67 | -0.76 |
| MoCo [13] | 40.30 | **66.07** | 0.5575 | 0.2135 | 26.67 | 21.83 | 25.61 | 51.78 | 64.85 | 6.33 | 0.16 |
| MoDo [14] | 35.28 | 62.62 | 0.5821 | 0.2405 | 25.65 | 20.33 | 28.04 | 54.86 | 67.37 | 8.89 | 0.49 |
| Nash-MTL [10] | 40.13 | 65.93 | 0.5261 | 0.2171 | 25.26 | 20.08 | 28.40 | 55.47 | 68.15 | 3.33 | -4.04 |
| FAMO [38] | 38.88 | 64.90 | 0.5474 | 0.2194 | 25.06 | 19.57 | **29.21** | 56.61 | 68.98 | 4.22 | -4.10 |
| SDMGrad | **40.47** | 65.90 | **0.5225** | **0.2084** | 25.07 | 19.99 | 28.54 | 55.74 | 68.53 | **2.33** | **-4.84** |

Table 3: Multi-task supervised learning on NYU-v2 dataset.

onds) per episode in Table 4. As shown, SDMGrad achieves the second best success rate among the compared baselines.

We also validate the efficiency of the proposed objective sampling strategy against other acceleration strategies including CAGrad-Fast [9], Nash-MTL with updating once per {50, 100} iterations [38]. Following [9], we choose the task sampling size $n = 4$. As shown in Table 4, our SDMGrad-OS with objective sampling achieves approximately $1.4\times$ speedup on MT10, while achieving a success rate comparable to that of SDMGrad. We also observe that although our SDMGrad-OS requires more time than CAGrad-Fast [9] and Nash-MTL [38] (every 100), it reaches a higher or comparable success rate.

| Method | Metaworld MT10 | |
| | success rate (mean ± stderr) | time |
|---|---|---|
| SAC STL (upper bound) | 0.90 ± 0.03 | – |
| Multi-task SAC [43] | 0.49 ± 0.07 | – |
| Multi-task SAC + Task Encoder [43] | 0.54 ± 0.05 | – |
| Multi-headed SAC [43] | 0.61 ± 0.04 | – |
| Nash-MTL⋆ [10] | **0.91** ± 0.03 | – |
| PCGrad [8] | 0.72 ± 0.02 | 11.6 |
| CAGrad [9] | 0.83 ± 0.05 | 13.5 |
| MoCo [13] | 0.75 ± 0.05 | 11.5 |
| Nash-MTL [38] | 0.80 ± 0.13 | 87.4 |
| FAMO [38] | 0.83 ± 0.05 | **4.2** |
| SDMGrad | 0.84 ± 0.10 | 13.6 |
| CAGrad-Fast [9] | 0.82 ± 0.04 | 8.6 |
| Nash-MTL [38] (every 50) | 0.76 ± 0.10 | 9.7 |
| Nash-MTL [38] (every 100) | 0.80 ± 0.12 | 9.3 |
| SDMGrad-OS | 0.82 ± 0.08 | 9.7 |
| SDMGrad-OS (S=1) | 0.80 ± 0.12 | 6.8 |

Table 4: Multi-task reinforcement learning on Metaworld MT10 benchmarks. Nash-MTL⋆ [10] denotes the results reported in the original paper. Nash-MTL [38] denotes the reproduced result.

## 7 Conclusion

In this paper, we propose a new and flexible direction-oriented multi-objective problem formulation, as well as two simple and efficient MOO algorithms named SDMGrad and SDMGrad-OS. We establish the convergence guarantee for both algorithms in various settings. Extensive experiments validate the promise of our methods. We anticipate that our new problem formulation and the proposed algorithms can be applied in other learning applications with multiple measures, and the analysis can be of independent interest in analyzing other stochastic MOO algorithms.

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

# Supplementary Materials

## A  Experiments

### A.1  Toy Example

To demonstrate our proposed method can achieve better or comparable performance under stochastic settings, we provide an empirical study on the two-objective toy example used in CAGrad [9]. The two objectives $L_1(x)$ and $L_2(x)$ shown in Figure 1 are defined on $x = (x_1, x_2)^\top \in \mathbb{R}^2$,

$$L_1(x) = f_1(x)g_1(x) + f_2(x)h_1(x)$$
$$L_2(x) = f_1(x)g_2(x) + f_2(x)h_2(x),$$

where the functions are given by

$$f_1(x) = \max\big(\tanh(0.5x_2), 0\big)$$
$$f_2(x) = \max\big(\tanh(-0.5x_2), 0\big)$$
$$g_1(x) = \log\Big(\max\big(|0.5(-x_1 - 7) - \tanh(-x_2)|, 0.000005\big)\Big) + 6$$
$$g_2(x) = \log\Big(\max\big(|0.5(-x_1 + 3) - \tanh(-x_2) + 2|, 0.000005\big)\Big) + 6$$
$$h_1(x) = \big((-x_1 + 7)^2 + 0.1(-x_1 - 8)^2\big)/10 - 20$$
$$h_2(x) = \big((-x_1 - 7)^2 + 0.1(-x_1 - 8)^2\big)/10 - 20.$$

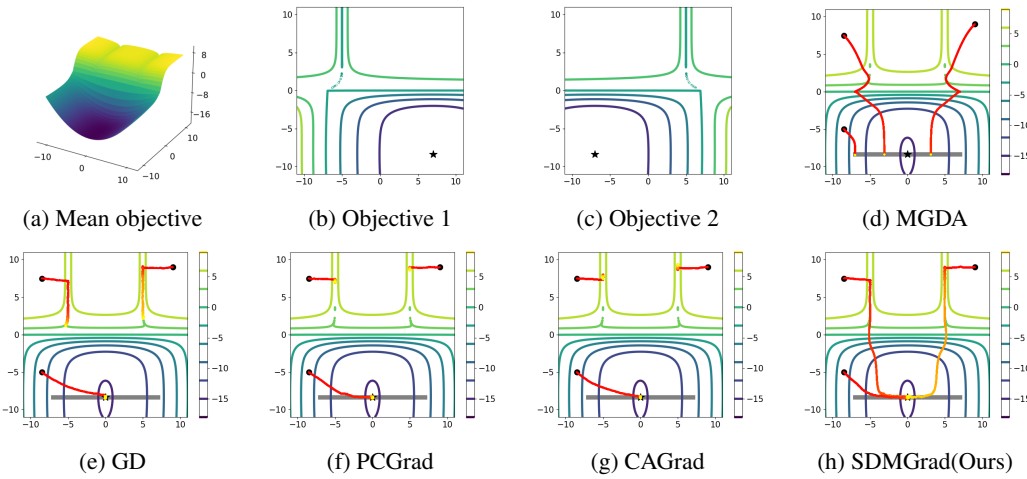

Figure 1: A two-objective toy example.

We choose 3 initializations

$$x_0 \in \{(-8.5, 7.5), (-8.5, 5), (9, 9)\}$$

for different methods and visualize the optimization trajectories in Figure 1. The starting point of every trajectory in Figure 1d-Figure 1h is given by the ● symbol, and the color of every trajectory changes gradually from red to yellow. The gray line illustrates the Pareto front, and the ⋆ symbol denotes the global optimum. To simulate the stochastic setting, we add zero-mean Gaussian noise to the gradient of each objective for all the methods except MGDA. We adopt Adam optimizer with learning rate of 0.002 and 70000 iterations for each run. As shown, GD can get stuck due to the dominant gradient of a specific objective, which stops progressing towards the Pareto front. PCGrad and CAGrad can also fail to converge to the Pareto front in certain circumstances.

## A.2 Consistency Verification

We conduct the experiment on the multi-task classification dataset Multi-Fashion+MNIST [45]. Each image contained in this dataset is constructed by overlaying two images randomly sampled from MNIST [46] and FashionMNIST [47] respectively. We adopt shrinked Lenet [48] as the shared base-encoder and a task-specific linear classification head for each task. We report the training losses obtained from different methods over 3 independent runs in Figure 2. As illustrated, the performance of SDMGrad with large $\lambda$ is similar to GD, and the performance when $\lambda$ is small resembles MGDA. With properly tuned $\lambda$, lower average training loss can be obtained. Generally, the results confirm the consistency of our formulation with the direction-oriented principle.

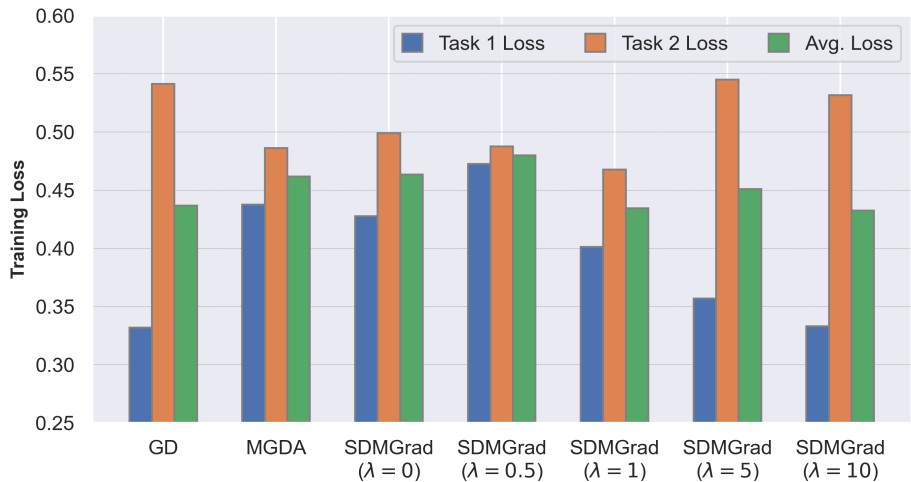

Figure 2: Consistency verification on Multi-Fashion+MNIST dataset.

The Multi-Fashion+MNIST[45] includes images constructed from FashionMNIST[47] and MNIST[46]. First, select one image from each dataset randomly, then transform the two images into a single image with one put in the top-left corner and the other in bottom-right corner. The dataset contains 120000 training images and 20000 test images. We use SGD optimizer with learning rate 0.001 and train for 100 epochs with batch size 256. We use multi-step scheduler with scale factor 0.1 to decay learning rate every 15 epochs. The projected gradient descent is performed with learning rate of 10 and momentum of 0.5 and 20 gradient descent steps are applied.

## A.3 Supervised Learning

We implement the methods based on the library released by [10]. Following [9, 13, 10], we train our method for 200 epochs, using Adam optimizer with learning rate 0.0001 for the first 100 epochs and 0.00005 for the rest. The batch size for Cityscapes and NYU-v2 are 8 and 2 respectively. We compute the averaged test performance over the last 10 epochs as final performance measure. The inner projected gradient descent is performed with learning rate of 10 and momentum of 0.5 and 20 gradient descent steps are applied. The experiments on Cityscapes and NYU-v2 are run on RTX 3090 and Tesla V100 GPU, respectively. We also report additional experiment results over different $\lambda$ and $S = 1$ in Table 5 and Table 6.

## A.4 Reinforcement Learning

Following [9, 13, 10], we conduct the experiments based on MTRL codebase[49]. We train our method for 2 million steps with batch size of 1280. The inner projected gradient descent is performed with learning rate of 10 for MT10 benchmark and 20 gradient descent steps are applied. The method is evaluated once every 10000 steps and the best average test performance over 10 random seeds over the entire training process is reported. We search $\lambda \in \{0.1, 0.2, \cdots, 1.0\}$ for MT10 benchmark and the highest success rate is achieved when $\lambda = 0.6$. For our objective sampling strategy, the number of sampled objectives is a random variable obeying binomial distribution whose expectation is $n$.

| Method | Segmentation | | Depth | | $\Delta m\% \downarrow$ |
|---|---|---|---|---|---|
| | mIoU ↑ | Pix Acc ↑ | Abs Err ↓ | Rel Err ↓ | |
| STL | 74.01 | 93.16 | 0.0125 | 27.77 | |
| SDMGrad ($\lambda = 0.1$) | 72.56 | 92.68 | 0.0156 | 40.89 | 18.65 |
| SDMGrad ($\lambda = 0.2$) | 74.79 | 93.30 | 0.0149 | 32.46 | 8.62 |
| SDMGrad ($\lambda = 0.3$) | 74.53 | 93.52 | 0.0137 | 34.01 | **7.79** |
| SDMGrad ($\lambda = 0.4$) | 75.10 | 93.48 | 0.0137 | 35.66 | 9.11 |
| SDMGrad ($\lambda = 0.5$) | 74.63 | 93.46 | 0.0131 | 38.99 | 11.09 |
| SDMGrad ($\lambda = 0.6$) | 74.42 | 93.22 | 0.0138 | 38.79 | 12.30 |
| SDMGrad ($\lambda = 0.7$) | 75.06 | 93.42 | 0.0158 | 39.98 | 17.24 |
| SDMGrad ($\lambda = 0.8$) | 74.99 | 93.40 | 0.0155 | 39.65 | 16.30 |
| SDMGrad ($\lambda = 0.9$) | 75.60 | 93.50 | 0.0134 | 43.52 | 15.39 |
| SDMGrad ($\lambda = 1.0$) | 74.50 | 93.47 | 0.0142 | 42.80 | 16.41 |
| SDMGrad ($\lambda = 10$) | 74.17 | 93.13 | 0.0154 | 41.77 | 18.36 |
| SDMGrad ($\lambda = 0.3, S = 1$) | 75.41 | 93.62 | 0.0139 | 38.83 | 12.22 |

Table 5: Addtional supervised learning experiments on Cityscapes dataset.

| Method | Segmentation | | Depth | | Surface Normal | | | | | $\Delta m\% \downarrow$ |
|---|---|---|---|---|---|---|---|---|---|---|
| | | | | | Angle Distance ↓ | | Within $t°$ ↑ | | | |
| | mIoU ↑ | Pix Acc ↑ | Abs Err ↓ | Rel Err ↓ | Mean | Median | 11.25 | 22.5 | 30 | |
| STL | 38.30 | 63.76 | 0.6754 | 0.2780 | 25.01 | 19.21 | 30.14 | 57.20 | 69.15 | |
| SDMGrad ($\lambda = 0.1$) | 40.23 | 66.01 | 0.5360 | 0.2268 | 25.03 | 19.99 | 28.45 | 55.80 | 68.65 | -3.86 |
| SDMGrad ($\lambda = 0.2$) | 39.23 | 65.67 | 0.5315 | 0.2189 | 25.13 | 20.02 | 28.12 | 55.71 | 68.46 | -3.66 |
| SDMGrad ($\lambda = 0.3$) | 40.47 | 65.90 | 0.5225 | 0.2084 | 25.07 | 19.99 | 28.54 | 55.74 | 68.53 | **-4.84** |
| SDMGrad ($\lambda = 0.4$) | 40.68 | 66.53 | 0.5248 | 0.2199 | 25.21 | 20.01 | 27.69 | 55.72 | 68.58 | -4.14 |
| SDMGrad ($\lambda = 0.5$) | 41.08 | 66.82 | 0.5184 | 0.2116 | 25.65 | 20.68 | 26.70 | 54.27 | 67.46 | -3.33 |
| SDMGrad ($\lambda = 0.6$) | 41.20 | 66.86 | 0.5258 | 0.2175 | 25.85 | 21.03 | 26.47 | 53.51 | 66.82 | -2.39 |
| SDMGrad ($\lambda = 0.7$) | 41.00 | 66.31 | 0.5224 | 0.2202 | 25.60 | 20.64 | 27.64 | 54.30 | 67.15 | -3.16 |
| SDMGrad ($\lambda = 0.8$) | 39.88 | 66.13 | 0.5406 | 0.2266 | 26.20 | 21.57 | 25.67 | 52.33 | 65.65 | -0.09 |
| SDMGrad ($\lambda = 0.9$) | 41.03 | 67.16 | 0.5314 | 0.2271 | 25.89 | 20.97 | 27.22 | 53.58 | 66.48 | -2.17 |
| SDMGrad ($\lambda = 1.0$) | 39.94 | 66.27 | 0.5224 | 0.2155 | 26.51 | 21.95 | 25.15 | 51.54 | 64.94 | -0.06 |
| SDMGrad ($\lambda = 10$) | 39.81 | 66.11 | 0.5352 | 0.2232 | 27.05 | 22.57 | 24.53 | 50.24 | 63.59 | 1.82 |
| SDMGrad ($\lambda = 0.3, S = 1$) | 39.63 | 65.43 | 0.5296 | 0.2140 | 25.66 | 20.83 | 27.18 | 53.93 | 67.05 | -2.34 |

Table 6: Addtional supervised learning experiments on NYU-v2 dataset.

To compare with CAGrad-Fast[9], we choose $n = 4$ for MT10 benchmark. We cite the reported success rates of all baseline methods in Table 4, but independently run each experiment 5 times to calculate the average running time. All experiments on MT10 are run on RTX 2080Ti GPU. We also report addtional experiments results over $S = 1$ on MT10 in Table 7.

| Method | Metaworld MT10 | |
|---|---|---|
| | success (mean ± stderr) | time |
| SDMGrad | **0.84** ± 0.10 | 13.6 |
| SDMGrad (S=1) | 0.83 ± 0.05 | 11.2 |
| SDMGrad-OS | 0.82 ± 0.08 | 9.7 |
| SDMGrad-OS (S=1) | 0.80 ± 0.12 | **6.8** |

Table 7: Additonal reinforcement learning experiments on Metaworld MT10 benchmarks.

# B  Notations for Technical Proofs

In this part, we first summarize all the notations that we used in this paper in order to help readers understand. First, in multi-objective optimization, we have $K \geq 2$ different objectives and each of them has the loss function $L_i(\theta)$. Let $g_i$ denote the gradient of objective $i$ and $g_0$ denotes the target gradient. $w = (w_1, ..., w_K)^T \in \mathbb{R}^K$ $and$ $\mathcal{W}$ denotes the probability simplex. Other useful notations are listed as below:

$$\theta^* = \arg\min_{\theta \in \mathbb{R}^m} \left\{ L_0(\theta) \triangleq \frac{1}{K} \sum_{i=1}^{K} L_i(\theta) \right\}, \; g_0 = g_0(\theta) = G(\theta)\widetilde{w}$$

$$g_w = \sum_i w_i g_i \quad s.t. \quad \mathcal{W} = \{w : \sum_i w_i = 1 \ \ and \ \ w_i \geq 0\}$$

$$w_\lambda^* = \arg\min_{w \in \mathcal{W}} \frac{1}{2}\|g_w + \lambda g_0\|^2, \quad w^* = \arg\min_{w \in \mathcal{W}} \frac{1}{2}\|g_w\|^2, \quad w_t^* = \arg\min_{w \in \mathcal{W}} \frac{1}{2}\|G(\theta_t)w\|^2$$

$$w_{t,\lambda}^* = \arg\min_{w \in \mathcal{W}} F(w) = \arg\min_{w \in \mathcal{W}} \frac{1}{2}\|G(\theta_t)w + \lambda g_0(\theta_t)\|^2$$

$$\nabla_w F(w) = G(\theta_t)^T (G(\theta_t)w + \lambda g_0(\theta_t)), \nabla_w \widehat{F}(w) = G(\theta_t; \xi)^T (G(\theta_t, \xi')w + \lambda g_0(\theta_t, \xi')). \quad (11)$$

We use $\mathbb{E}[\cdot]_{A|B}$ to denote taking expectation over $A$ conditioning on $B$ and $\widetilde{\mathcal{O}}$ omits the order of $\log$.

## C Detailed proofs for convergence analysis with nonconvex Objectives

We now provide some auxiliary lemmas for proving Proposition 1 and Theorem 1

**Lemma 1.** *Let $d^*$ be the solution of*

$$\max_{d \in \mathbb{R}^m} \min_{i \in [K]} \langle g_i, d \rangle - \frac{1}{2}\|d\|^2 + \lambda \langle g_0, d \rangle,$$

*then we have*

$$d^* = g_{w_\lambda^*} + \lambda g_0.$$

*In addition, $w_\lambda^*$ is the solution of*

$$\min_{w \in \mathcal{W}} \frac{1}{2}\|g_w + \lambda g_0\|^2.$$

*Proof.* First, it can be seen that

$$\max_{d \in \mathbb{R}^m} \min_{i \in [K]} \langle g_i, d \rangle - \frac{1}{2}\|d\|^2 + \lambda \langle g_0, d \rangle$$

$$= \max_{d \in \mathbb{R}^m} \min_{w \in \mathcal{W}} \langle \sum_i w_i g_i, d \rangle - \frac{1}{2}\|d\|^2 + \lambda \langle g_0, d \rangle$$

$$= \max_{d \in \mathbb{R}^m} \min_{w \in \mathcal{W}} g_w{}^T d - \frac{1}{2}\|d\|^2 + \lambda \langle g_0, d \rangle. \quad (12)$$

Noting that the problem is concave w.r.t. $d$ and convex w.r.t $w$ and using the Von Neumann-Fan minimax theorem [50], we can exchange the min and max problems without changing the solution. Then, we can solve the following equivalent problem.

$$\min_{w \in \mathcal{W}} \max_{d \in \mathbb{R}^m} g_w{}^T d - \frac{1}{2}\|d\|^2 + \lambda \langle g_0, d \rangle \quad (13)$$

Then by fixing $w$, we have $d^* = g_w + \lambda g_0$. Substituting this solution to the eq. (13) and rearranging the equation, we turn to solve the following problem.

$$\min_{w \in \mathcal{W}} \frac{1}{2}\|g_w + \lambda g_0\|^2.$$

Let $w_\lambda^*$ be the solution of the above problem, and hence the final updating direction $d^* = g_{w_\lambda^*} + \lambda g_0$. Then, the proof is complete. $\qquad\square$

**Lemma 2.** *Suppose Assumption 2-3 are satisfied. According to the definition of $g_0(\theta)$ in eq. (11), we have the following inequalities,*

$$\|g_0(\theta)\| \leq C_g, \quad \mathbb{E}[\|g_0(\theta; \xi) - g_0(\theta)\|^2] \leq K\sigma_0^2.$$

*Proof.* Based on the definitions, we have

$$\|g_0(\theta)\| = \|G(\theta)\widetilde{w}\| \leq C_g,$$

where the inequality follows from the fact that $\|\widetilde{w}\| \leq 1$ and Assumption 3. Then, we have

$$\mathbb{E}_\xi[\|g_0(\theta; \xi) - g_0(\theta)\|^2] \leq \mathbb{E}_\xi[\|G(\theta; \xi) - G(\theta)\|^2] \leq K\sigma_0^2$$

where $\sigma_0^2 = \max_i \sigma_i^2$ and the proof is complete. $\qquad\square$

**Lemma 3.** *Suppose Assumptions 2-3 are satisfied and recall that $F(w) = \frac{1}{2}\|G(\theta_t)w + \lambda g_0(\theta_t)\|^2$ is a convex function. Let $w_\lambda^* = \arg\min_{w \in \mathcal{W}} \frac{1}{2}\|g_w + \lambda g_0\|^2$ and set step size $\beta_{t,s} = c/\sqrt{s}$ where $c > 0$ is a constant. Then for any $S > 1$, it holds that,*

$$\mathbb{E}[\|\nabla_w \widehat{F}(w)\|] \leq C_1,$$

$$\mathbb{E}[\|G(\theta_t)w_S + \lambda g_0(\theta_t)\|^2 - \|G(\theta_t)w_\lambda^* + \lambda g_0(\theta_t)\|^2] \leq (\frac{2}{c} + 2cC_1)\frac{2 + log(S)}{\sqrt{S}}$$

*where $C_1 = \sqrt{8(K\sigma_0^2 + C_g^2)^2 + 8\lambda^2(K\sigma_0^2 + C_g^2)^2} = \mathcal{O}(K + \lambda K)$, $\nabla_w \widehat{F}(w) = G(\theta_t; \xi)^T(G(\theta_t; \xi')w + \lambda g_0(\theta_t; \xi'))$.*

*Proof.* This lemma mostly follows from Theorem 2 in [51]. However, we did not take that $\mathbb{E}[\|\nabla_w \widehat{F}(w)\|]$ is bounded by a constant as an assumption. Therefore, we first provide a bound for it in our method. Based on the definition in Equation (11), $\nabla_w \widehat{F}(w) = G(\theta_t; \xi)^T(G(\theta_t; \xi')w + \lambda g_0(\theta_t; \xi'))$. According to the fact that $\mathbb{E}[X] \leq \sqrt{\mathbb{E}[X^2]}$, we have

$$\mathbb{E}[\|\nabla_w \widehat{F}(w)\|] \leq \sqrt{\mathbb{E}[\|\nabla_w \widehat{F}(w)\|^2]} = \sqrt{\mathbb{E}[\|G(\theta_t; \xi)^T(G(\theta_t; \xi')w + \lambda g_0(\theta_t; \xi'))\|^2]}$$

$$\overset{(i)}{\leq} \sqrt{2\mathbb{E}[\underbrace{\|G(\theta_t; \xi)^T G(\theta_t; \xi')w\|^2}_{A} + \lambda^2 \underbrace{\|G(\theta_t; \xi)^T g_0(\theta_t; \xi'))\|^2}_{B}]}, \tag{14}$$

where $(i)$ follows from the Young's inequality. Next, we provide bounds for $\mathbb{E}[A]$ and $\mathbb{E}[B]$, separately:

$$\mathbb{E}[A] \overset{(i)}{\leq} \mathbb{E}[\|(G(\theta_t; \xi)^T - G(\theta_t)^T + G(\theta_t)^T)(G(\theta_t; \xi') - G(\theta_t) + G(\theta_t))\|^2]$$

$$= \mathbb{E}[\|(G(\theta_t; \xi)^T - G(\theta_t)^T)(G(\theta_t; \xi') - G(\theta_t)) + (G(\theta_t; \xi)^T - G(\theta_t)^T)G(\theta_t)$$

$$+ G(\theta_t)^T(G(\theta_t; \xi') - G(\theta_t)) + G(\theta_t)^T G(\theta_t)\|^2]$$

$$\overset{(ii)}{\leq} 4\mathbb{E}[\|G(\theta_t; \xi)^T - G(\theta_t)^T\|^2\|G(\theta_t; \xi') - G(\theta_t)\|^2 + \|G(\theta_t; \xi)^T - G(\theta_t)^T\|^2\|G(\theta_t)\|^2$$

$$+ \|G(\theta_t)^T\|^2\|(G(\theta_t; \xi') - G(\theta_t)\|^2 + \|G(\theta_t)^T G(\theta_t)\|^2]$$

$$\overset{(iii)}{\leq} 4K^2\sigma_0^4 + 8K\sigma_0^2 C_g^2 + 4C_g^4 = 4(K\sigma_0^2 + C_g^2)^2, \tag{15}$$

where $(i)$ follows from Cauchy–Schwarz inequality and $w \in \mathcal{W}$ where $\mathcal{W}$ is the simplex, $(ii)$ follows from Young's inequality and $(iii)$ follows from Assumption 2 and Assumption 3. Then for term B, we have,

$$\mathbb{E}[B] = \mathbb{E}[\|(G(\theta_t; \xi)^T - G(\theta_t)^T + G(\theta_t)^T)(g_0(\theta_t; \xi') - g_0(\theta_t) + g_0(\theta_t))\|^2]$$

$$\overset{(i)}{\leq} 4\mathbb{E}[\|(G(\theta_t; \xi)^T - G(\theta_t)^T)(g_0(\theta_t; \xi') - g_0(\theta_t))\|^2 + \|(G(\theta_t; \xi)^T - G(\theta_t)^T)g_0(\theta_t)\|^2$$

$$+ \|G(\theta_t)^T(g_0(\theta_t; \xi') - g_0(\theta_t))\|^2 + \|G(\theta_t^T)g_0(\theta_t)\|^2]$$

$$\overset{(ii)}{\leq} 4K^2\sigma_0^4 + 8K\sigma_0^2 C_g^2 + 4C_g^4 = 4(K\sigma_0^2 + C_g^2)^2, \tag{16}$$

where $(i)$ follows from Young's inequality, $(ii)$ follows from Assumption 3 and Lemma 2. Then substituting eq. (15) and eq. (16) into eq. (14), we can obtain,

$$\mathbb{E}[\|\nabla_w \widehat{F}(w)\|] \leq \sqrt{8(K\sigma_0^2 + C_g^2)^2 + 8\lambda^2(K\sigma_0^2 + C_g^2)^2} = C_1.$$

Meanwhile, since $\mathbb{E}[\|\nabla_w \widehat{F}(w)\|] \leq C_1$, $\sup_{w,w'}\|w - w'\| \leq 1$ and by choosing step size $\beta_s = c/\sqrt{s}$ where $c > 0$ is a constant, we can obtain the following inequality from Theorem 2 in [51]:

$$\mathbb{E}[F(w_S) - F(w_\lambda^*)] \leq (\frac{1}{c} + cC_1)\frac{2 + log(S)}{\sqrt{S}} \tag{17}$$

Then after multiplying by 2 on both sides, the proof is complete. $\qquad\square$

## C.1 Proof of Proposition 1

**CA distance.** Now we show the upper bound for the distance to CA direction. Recall that we define the CA distance as $\|\mathbb{E}_{\zeta,w_{t,S}|\theta_t}[G(\theta_t;\zeta)w_{t,S} + \lambda g_0(\theta_t;\zeta)] - G(\theta_t)w_{t,\lambda}^* - \lambda g_0(\theta_t)\|$.

*Proof.* Based on the Jensen's inequality, we have

$$
\begin{aligned}
&\|\mathbb{E}_{\zeta,w_{t,S}|\theta_t}[G(\theta_t;\zeta)w_{t,S} + \lambda g_0(\theta_t;\zeta)] - G(\theta_t)w_{t,\lambda}^* - \lambda g_0(\theta_t)\|^2 \\
&\leq \mathbb{E}_{w_{t,S}|\theta_t}\left[\left\|\mathbb{E}_\zeta[G(\theta_t;\zeta)w_{t,S} + \lambda g_0(\theta_t;\zeta)] - G(\theta_t)w_{t,\lambda}^* - \lambda g_0(\theta_t)\right\|^2\right] \\
&\overset{(i)}{=} \mathbb{E}[\|G(\theta_t)w_{t,S} - G(\theta_t)w_{t,\lambda}^*\|^2] \\
&= \mathbb{E}[\|G(\theta_t)w_{t,S} + \lambda g_0(\theta_t) - G(\theta_t)w_{t,\lambda}^* - \lambda g_0(\theta_t)\|^2] \\
&= \mathbb{E}[\|G(\theta_t)w_{t,S} + \lambda g_0(\theta_t)\|^2 + \|G(\theta_t)w_{t,\lambda}^* + \lambda g_0(\theta_t)\|^2 \\
&\quad - 2\mathbb{E}\langle G(\theta_t)w_{t,S} + \lambda g_0(\theta_t), G(\theta_t)w_{t,\lambda}^* + \lambda g_0(\theta_t)\rangle] \\
&= \mathbb{E}[\|G(\theta_t)w_{t,S} + \lambda g_0(\theta_t)\|^2 + \|G(\theta_t)w_{t,\lambda}^* + \lambda g_0(\theta_t)\|^2] \\
&\quad - 2\mathbb{E}[\langle G(\theta)w_{t,S}, G(\theta_t)w_{t,\lambda}^* + \lambda g_0(\theta_t)\rangle] - 2\mathbb{E}[\langle \lambda g_0(\theta_t), G(\theta_t)w_{t,\lambda}^* + \lambda g_0(\theta_t)\rangle] \\
&\overset{(ii)}{\leq} \mathbb{E}[\|G(\theta_t)w_{t,S} + \lambda g_0(\theta_t)\|^2 + \|G(\theta_t)w_{t,\lambda}^* + \lambda g_0(\theta_t)\|^2] \\
&\quad - 2\mathbb{E}[\langle G(\theta)w_{t,\lambda}^*, G(\theta_t)w_{t,\lambda}^* + \lambda g_0(\theta_t)\rangle] - 2\mathbb{E}[\langle \lambda g_0(\theta_t), G(\theta_t)w_{t,\lambda}^* + \lambda g_0(\theta_t)\rangle] \\
&= \mathbb{E}[\|G(\theta_t)w_{t,S} + \lambda g_0(\theta_t)\|^2 + \|G(\theta_t)w_{t,\lambda}^* + \lambda g_0(\theta_t)\|^2] \\
&\quad - 2\mathbb{E}[\langle G(\theta)w_{t,\lambda}^* + \lambda g_0(\theta_t), G(\theta_t)w_{t,\lambda}^* + \lambda g_0(\theta_t)\rangle] \\
&= \mathbb{E}[\|G(\theta_t)w_{t,S} + \lambda g_0(\theta_t)\|^2 - \|G(\theta_t)w_{t,\lambda}^* + \lambda g_0(\theta_t)\|^2] \\
&\overset{(iii)}{\leq} \left(\frac{2}{c} + 2cC_1\right)\frac{2 + log(S)}{\sqrt{S}}
\end{aligned}
\tag{18}
$$

where $(i)$ omits the subscript of taking expectation over $w_{t,S}$ conditioning on $\theta_t$, $(ii)$ follows from optimality condition that

$$
\langle w, G(\theta_t)^T(G(\theta_t)w_{t,\lambda}^* + \lambda g_0(\theta_t))\rangle \geq \langle w_{t,\lambda}^*, G(\theta_t)^T(G(\theta_t)w_{t,\lambda}^* + \lambda g_0(\theta_t))\rangle.
\tag{19}
$$

$(iii)$ follows from Lemma 3 whenwe choose $\beta_{t,s} = c/\sqrt{s}$ where $c$ is a constant. Then take the square root on both sides, the proof is complete. $\qquad\square$

## C.2 Proof of Theorem 1

**Theorem 5** (Restatement of Theorem 1). *Suppose Assumptions 1-3 are satisfied. Set $\alpha_t = \alpha = \Theta((1 + \lambda)^{-1}K^{-\frac{1}{2}}T^{-\frac{1}{2}})$, $\beta_{t,s} = c/\sqrt{s}$ where $c$ is a constant, and $S = \Theta((1 + \lambda)^{-2}T^2)$. The outputs of the proposed SDMGrad algorithm satisfy*

$$
\frac{1}{T}\sum_{t=0}^{T-1}\mathbb{E}[\|G(\theta_t)w_{t,\lambda}^* + \lambda g_0(\theta_t)\|^2] = \widetilde{\mathcal{O}}((1 + \lambda^2)K^{\frac{1}{2}}T^{-\frac{1}{2}}).
$$

*Proof.* Recall that $d = G(\theta_t;\zeta)w_{t,S} + \lambda g_0(\theta_t;\zeta)$. According to Assumption 1, we have for any $i$,

$$
L_i(\theta_{t+1}) + \lambda L_0(\theta_{t+1}) \leq L_i(\theta_t) + \lambda L_0(\theta_t) + \alpha_t\langle g_i(\theta_t) + \lambda g_0(\theta_t), -d\rangle + \frac{l'_{i,1}\alpha_t^2}{2}\|d\|^2.
\tag{20}
$$

where $l'_{i,1} = l_{i,1} + \lambda \max_i l_{i,1} = \Theta(1 + \lambda)$. Then we bound the second and third terms separately on the right-hand side (RHS). First, for the second term, conditioning on $\theta_t$ and taking expectation, we have

$$
\begin{aligned}
&\mathbb{E}[\langle g_i(\theta_t) + \lambda g_0(\theta_t), -G(\theta_t;\zeta)w_{t,S} - \lambda g_0(\theta_t;\zeta)\rangle|\theta_t] \\
&= \mathbb{E}[\langle g_i(\theta_t) + \lambda g_0(\theta_t), -G(\theta_t)w_{t,S} - \lambda g_0(\theta_t)\rangle|\theta_t] \\
&= \mathbb{E}[\langle g_i(\theta_t) + \lambda g_0(\theta_t), G(\theta_t)w_{t,\lambda}^* - G(\theta_t)w_{t,S}\rangle - \langle g_i(\theta_t) + \lambda g_0(\theta_t), G(\theta_t)w_{t,\lambda}^* + \lambda g_0(\theta_t)\rangle|\theta_t]
\end{aligned}
$$

$$\overset{(i)}{\leq}\mathbb{E}[(l_i + \lambda C_g)\|G(\theta_t)w_{t,\lambda}^* - G(\theta_t)w_{t,S}\||\theta_t] - \mathbb{E}[\|G(\theta_t)w_{t,\lambda}^* + \lambda g_0(\theta_t)\|^2|\theta_t]$$

$$\overset{(ii)}{\leq}(l_i + \lambda C_g)\sqrt{\mathbb{E}[\|G(\theta_t)w_{t,\lambda}^* - G(\theta_t)w_{t,S}\|^2|\theta_t]} - \mathbb{E}[\|G(\theta_t)w_{t,\lambda}^* + \lambda g_0(\theta_t)\|^2|\theta_t]$$

$$\overset{(iii)}{\leq}(l_i + \lambda C_g)\sqrt{(\frac{2}{c} + 2cC_1)\frac{2 + log(S)}{\sqrt{S}}} - \mathbb{E}[\|G(\theta_t)w_{t,\lambda}^* + \lambda g_0(\theta_t)\|^2|\theta_t] \tag{21}$$

where $(i)$ follows from Cauchy-Schwarz inequality and optimality condition in eq. (19), $(ii)$ follows from the fact that $\mathbb{E}[X] \leq \sqrt{\mathbb{E}[X^2]}$ and $(iii)$ follows from eq. (18).

Then for the third term,

$$\mathbb{E}[\|d\|^2] = \mathbb{E}[\|G(\theta_t;\zeta)w_{t,S} + \lambda g_0(\theta_t;\zeta)\|^2]$$

$$= \mathbb{E}[\|G(\theta_t;\zeta)w_{t,S} - G(\theta_t)w_{t,S} + G(\theta_t)w_{t,S} + \lambda g_0(\theta_t;\zeta) - \lambda g_0(\theta_t) + \lambda g_0(\theta_t)\|^2]$$

$$\overset{(i)}{\leq}4\mathbb{E}[\|G(\theta_t;\zeta) - G(\theta_t)\|^2] + 4\mathbb{E}[\|G(\theta_t)\|^2] + 4\lambda^2\mathbb{E}[\|g_0(\theta_t;\zeta) - g_0(\theta_t)\|^2]$$
$$\quad + 4\lambda^2\mathbb{E}[\|g_0(\theta_t)\|^2]$$

$$\overset{(ii)}{\leq}\underbrace{4K\sigma_0^2 + 4C_g^2 + 4\lambda^2 K\sigma_0^2 + 4\lambda^2 C_g^2}_{C_2} \tag{22}$$

where $(i)$ follows from Young's inequality, and $(ii)$ follows from Assumption 3 and Lemma 2. Note that $C_2 = \mathcal{O}(K + K\lambda^2)$. Then taking expectation on eq. (20), substituting eq. (21) and eq. (22) into it, and unconditioning on $\theta_t$, we have

$$\mathbb{E}[L_i(\theta_{t+1}) + \lambda L_0(\theta_{t+1})]$$

$$\leq \mathbb{E}[L_i(\theta_t) + \lambda L_0(\theta_t)] + \alpha_t \mathbb{E}[\langle g_i(\theta_t) + \lambda g_0(\theta_t), -d\rangle] + \frac{l_{i,1}'\alpha_t^2}{2}\mathbb{E}[\|d\|^2]$$

$$\leq \mathbb{E}[L_i(\theta_t) + \lambda L_0(\theta_t)] + \alpha_t(l_i + \lambda C_g)\sqrt{(\frac{2}{c} + 2cC_1)\frac{2 + log(S)}{\sqrt{S}}}$$

$$\quad - \alpha_t \mathbb{E}[\|G(\theta_t)w_{t,\lambda}^* + \lambda g_0(\theta_t)\|^2] + \frac{l_{i,1}'\alpha_t^2}{2}C_2 \tag{23}$$

Then, choosing $\alpha_t = \alpha$, and rearranging the above inequality, we have

$$\alpha\mathbb{E}[\|G(\theta_t)w_{t,\lambda}^* + \lambda g_0(\theta_t)\|^2] \leq \mathbb{E}[L_i(\theta_t) + \lambda L_0(\theta_t) - L_i(\theta_{t+1}) - \lambda L_0(\theta_{t+1})]$$

$$+ \alpha(l_i + \lambda C_g)\sqrt{(\frac{2}{c} + 2cC_1)\frac{2 + log(S)}{\sqrt{S}}} + \frac{l_{i,1}'\alpha^2}{2}C_2.$$

Telescoping over $t \in [T]$ in the above inequality yields

$$\frac{1}{T}\sum_{t=0}^{T-1}\mathbb{E}[\|G(\theta_t)w_{t,\lambda}^* + \lambda g_0(\theta_t)\|^2]$$

$$\leq \frac{1}{\alpha T}\mathbb{E}[L_i(\theta_0) - \inf L_i(\theta) + \lambda(L_0(\theta_0) - \inf L_0(\theta))] + \frac{l_{i,1}'\alpha}{2}C_2$$

$$+ (l_i + \lambda C_g)\sqrt{(\frac{2}{c} + 2cC_1)\frac{2 + log(S)}{\sqrt{S}}},$$

If we choose $\alpha = \Theta((1 + \lambda)^{-1}K^{-\frac{1}{2}}T^{-\frac{1}{2}})$ and $S = \Theta((1 + \lambda)^{-2}T^2)$, we have

$$\frac{1}{T}\sum_{t=0}^{T-1}\mathbb{E}[\|G(\theta_t)w_{t,\lambda}^* + \lambda g_0(\theta_t)\|^2] = \widetilde{\mathcal{O}}((1 + \lambda^2)K^{\frac{1}{2}}T^{-\frac{1}{2}}),$$

where $\widetilde{\mathcal{O}}$ means the order of $logT$ is omitted. The proof is complete. $\qquad\square$

### C.3 Proof of Corollary 1

*Proof.* Since $\lambda > 0$ and $g_0(\theta_t) = G(\theta_t)\widetilde{w}$, we have

$$\mathbb{E}[\|G(\theta_t)w_{t,\lambda}^* + \lambda g_0(\theta_t)\|^2] = (1+\lambda)^2 \mathbb{E}[\|G(\theta_t)w'\|^2] \geq (1+\lambda)^2 \mathbb{E}[\|G(\theta_t)w_t^*\|^2]$$

where $w' = \frac{1}{1+\lambda}(w_{1,t,\lambda}^* + \lambda\widetilde{w}_1, w_{2,t,\lambda}^* + \lambda\widetilde{w}_2, ..., w_{K,t,\lambda}^* + \lambda\widetilde{w}_K)^T$ such that $w' \in \mathcal{W}$. According to parameter selection in Theorem 1 and by choosing a constant $\lambda$, we have

$$\frac{1}{T}\sum_{t=0}^{T-1}\mathbb{E}[\|G(\theta_t)w_t^*\|^2] = \widetilde{\mathcal{O}}(K^{\frac{1}{2}}T^{-\frac{1}{2}}). \tag{24}$$

To achieve an $\epsilon$-accurate Pareto stationary point, it requires $T = \widetilde{\mathcal{O}}(K\epsilon^{-2})$ and each objective requires $\widetilde{\mathcal{O}}(K^3\epsilon^{-6})$ samples in $\xi$ ($\xi'$) and $\widetilde{\mathcal{O}}(K\epsilon^{-2})$ samples in $\zeta$, respectively. Meanwhile, according to the choice of $S$ and $T$, we have the following result for CA distance,

$$\|\mathbb{E}_{\zeta,w_{t,S}|\theta_t}[G(\theta_t,\zeta)w_{t,S} + \lambda g_0(\theta_t;\zeta)] - G(\theta_t)w_{t,\lambda}^* - \lambda g_0(\theta_t)\| = \widetilde{\mathcal{O}}(\sqrt{\frac{K}{T}}) = \widetilde{\mathcal{O}}(\epsilon) \tag{25}$$

**Remark.** Our algorithm with a constant $\lambda$ helps mitigate gradient conflict and it guarantees an $\epsilon-$accurate Pareto stationary point and the CA distance takes the order of $\widetilde{O}(\epsilon)$ simultaneously. □

### C.4 Proof of Corollary 2

*Proof.* According to the inequality $\|a + b - b\|^2 \leq 2\|a + b\|^2 + 2\|b\|^2$, we have

$$\lambda^2\|g_0(\theta_t)\|^2 \leq 2\|G(\theta_t)w_{t,\lambda}^* + \lambda g_0(\theta_t)\|^2 + 2\|G(\theta_t)w_{t,\lambda}^*\|^2$$
$$\leq 2\|G(\theta_t)w_{t,\lambda}^* + \lambda g_0(\theta_t)\|^2 + 2C_g^2$$

where the last inequality follows from Assumption 3. Then we take the expectation on the above inequality and sum up it over $t \in [T]$ such that

$$\frac{1}{T}\sum_{t=0}^{T-1}\mathbb{E}[\|g_0(\theta_t)\|^2] \leq \frac{2}{\lambda^2 T}\sum_{t=0}^{T-1}\mathbb{E}[\|G(\theta_t)w_{t,\lambda}^* + \lambda g_0(\theta_t)\|^2] + \frac{C_g^2}{\lambda^2}$$
$$= \widetilde{\mathcal{O}}((\lambda^{-2} + 1)K^{\frac{1}{2}}T^{-\frac{1}{2}} + \lambda^{-2}),$$

where the last inequality follows from Theorem 1. If we choose $\lambda = \Theta(T^{\frac{1}{2}})$, then we have

$$\frac{1}{T}\sum_{t=0}^{T-1}\mathbb{E}[\|g_0(\theta_t)\|^2] = \widetilde{\mathcal{O}}(K^{\frac{1}{2}}T^{-\frac{1}{2}}).$$

To achieve an $\epsilon$-accurate stationary point, it requires $T = \widetilde{\mathcal{O}}(K\epsilon^{-2})$ and each objective requires $\widetilde{\mathcal{O}}(K^2\epsilon^{-4})$ samples in $\xi$ ($\xi'$) and $\widetilde{\mathcal{O}}(K\epsilon^{-2})$ samples in $\zeta$, respectively. Meanwhile, according to the choice of $\lambda$, $S$ and $T$, we have the following result for CA distance,

$$\|\mathbb{E}_{\zeta,w_{t,S}|\theta_t}[G(\theta_t,\zeta)w_{t,S} + \lambda g_0(\theta_t;\zeta)] - G(\theta_t)w_{t,\lambda}^* - \lambda g_0(\theta_t)\| = \widetilde{\mathcal{O}}(\sqrt{\frac{K(1+\lambda)^2}{T}}) = \widetilde{\mathcal{O}}(\sqrt{K})$$

**Remark.** With an increasing $\lambda$, our algorithm approaches GD and it has a faster convergence rate to the stationary point. However, the CA distance takes the order of $\widetilde{O}(\sqrt{K})$. □

### C.5 Proof of Theorem 2

Now we provide the convergence analysis with nonconvex objectives with objective sampling.

**Theorem 6** (Restatement of Theorem 2). *Suppose Assumptions 1-3 are satisfied. Set $\gamma = \frac{K}{n}$, $\alpha_t = \alpha = \Theta((1+\lambda^2)^{-\frac{1}{2}}\gamma^{-\frac{1}{2}}K^{-\frac{1}{2}}T^{-\frac{1}{2}})$, $\beta_{t,s} = c/\sqrt{s}$ and $S = \Theta((1+\lambda)^{-2}\gamma^{-2}T^2)$. Then by choosing a constant $\lambda$, the iterates of the proposed SDMGrad-OS algorithm satisfy*

$$\frac{1}{T}\sum_{t=0}^{T-1}\mathbb{E}[\|G(\theta_t)w_t^*\|^2] = \widetilde{\mathcal{O}}(K^{\frac{1}{2}}\gamma^{\frac{1}{2}}T^{-\frac{1}{2}}).$$

*Proof.* Recall that updating direction for $\theta_t$ is $d' = \frac{K}{n}H(\theta_t; \zeta, \widetilde{S})w_{t,S} + \frac{K}{n}\lambda h_0(\theta_t; \zeta, \widetilde{S})$. Similarly, we have

$$L_i(\theta_{t+1}) + \lambda L_0(\theta_{t+1}) \le L_i(\theta_t) + \lambda L_0(\theta_t) + \alpha_t \langle g_i(\theta_t) + \lambda g_0(\theta_t), -d' \rangle + \frac{l'_{i,1}\alpha_t^2}{2}\|d'\|^2. \quad (26)$$

Then for the inner product term on the RHS of eq. (26), conditioning on $\theta_t$ and taking expectation, we have

$$\mathbb{E}[\langle g_i(\theta_t) + \lambda g_0(\theta_t), -d' \rangle | \theta_t] = \mathbb{E}[\langle g_i(\theta_t) + \lambda g_0(\theta_t), -G(\theta_t)w_{t,S} - \lambda g_0(\theta_t) \rangle | \theta_t]$$

$$\le (l_i + \lambda C_g)\left(\sqrt{(\frac{2}{c} + 2cC_1)\frac{2 + log(S)}{\sqrt{S}}}\right) - \mathbb{E}[\|G(\theta_t)w_{t,\lambda}^* + \lambda g_0(\theta_t)\|^2 | \theta_t], \quad (27)$$

where the last inequality follows from eq. (21). Then following the same step as in eq. (22), we can bound the last term on the RHS of eq. (26) as

$$\mathbb{E}[\|d'\|^2] \le \underbrace{4\gamma^2(1 + \lambda^2)(n\sigma_0^2 + C_g^2)}_{C_2'}. \quad (28)$$

Then taking expectation on eq. (26), substituting eq. (27) and eq. (28) into it and unconditioning on $\theta_t$, we have

$$\mathbb{E}[L_i(\theta_{t+1}) + \lambda L_0(\theta_{t+1})] \le \mathbb{E}[L_i(\theta_t) + \lambda L_0(\theta_t)] + \alpha_t(l_i + \lambda C_g)\left(\sqrt{(\frac{2}{c} + 2cC_1)\frac{2 + log(S)}{\sqrt{S}}}\right)$$

$$- \alpha_t \mathbb{E}\|G(\theta_t)w_{t,\lambda}^* + \lambda g_0(\theta_t)\|^2 + \frac{l'_{i,1}\alpha_t^2}{2}C_2'$$

Then choosing $\alpha_t = \alpha$, telescoping the above inequality over $t \in [T]$, and rearranging the terms, we have

$$\frac{1}{T}\sum_{t=0}^{T-1}\mathbb{E}[\|G(\theta_t)w_{t,\lambda}^* + \lambda g_0(\theta_t)\|^2] \le \frac{1}{\alpha T}\mathbb{E}[L_i(\theta_0) - \inf L_i(\theta) + \lambda(L_0(\theta_0) - \inf L_0(\theta))] + \frac{l'_{i,1}\alpha}{2}C_2'$$

$$+ (l_i + \lambda C_g)\left(\sqrt{(\frac{2}{c} + 2cC_1)\frac{2 + log(S)}{\sqrt{S}}}\right)$$

If we choose $\alpha = \Theta((1 + \lambda^2)^{-\frac{1}{2}}\gamma^{-\frac{1}{2}}K^{-\frac{1}{2}}T^{-\frac{1}{2}})$, and $S = \Theta((1 + \lambda)^{-2}\gamma^{-2}T^2)$, we can get $\frac{1}{T}\sum_{t=0}^{T-1}\mathbb{E}[\|G(\theta_t)w_{t,\lambda}^* + \lambda g_0(\theta_t)\|^2] = \widetilde{\mathcal{O}}((1 + \lambda^2)K^{\frac{1}{2}}\gamma^{\frac{1}{2}}T^{-\frac{1}{2}})$. Furthermore, by choosing $\lambda$ as constant and following the same step as in Appendix C.3, we have

$$\frac{1}{T}\sum_{t=0}^{T-1}\mathbb{E}[\|G(\theta_t)w_t^*\|^2] = \widetilde{\mathcal{O}}(K^{\frac{1}{2}}\gamma^{\frac{1}{2}}T^{-\frac{1}{2}}).$$

To achieve an $\epsilon$-accurate Pareto stationary point, it requires $T = \widetilde{\mathcal{O}}(K\gamma\epsilon^{-2})$. In this case, each objective requires a similar number of samples $\widetilde{\mathcal{O}}(K^3\gamma\epsilon^{-6})$ in $\xi$ ($\xi'$) and $\widetilde{\mathcal{O}}(K\gamma\epsilon^{-2})$ samples in $\zeta$, respectively. As far as we know, this is the first provable objective sampling strategy for stochastic multi-objective optimization. $\square$

## D Lower sample complexity but higher CA distance

When we do not have requirements on CA distance, we can have a much lower sample complexity. In Algorithm 1, the update process for $w$ is to reduce the CA distance, which increases the sample complexity. Thus, we will set $S = 1$ to make Algorithm 1 more sample-efficient. In addition, we will use $w_{t+1} = w_{t,1}$ and $\beta_t$ instead of $\beta_{t,s}$ in Algorithm 1 for simplicity. The following proof is mostly motivated by Theorem 3 in [14].

## D.1 Proof of Theorem 3

**Theorem 7** (Restatement of Theorem 3). *Suppose Assumptions 1-3 are satisfied and $S = 1$. Set $\alpha_t = \alpha = \Theta(K^{-\frac{1}{2}}T^{-\frac{1}{2}})$, $\beta_t = \beta = \Theta(K^{-1}T^{-\frac{1}{2}})$ and $\lambda$ as constant. The iterates of the proposed SDMGrad algorithm satisfy,*

$$\frac{1}{T}\sum_{t=0}^{T-1}\mathbb{E}[\|G(\theta_t)w_t^*\|^2] = \mathcal{O}(KT^{-\frac{1}{2}}).$$

*Proof.* Now we define a new function, with a fixed weight $w \in \mathcal{W}$,

$$l'(\theta_t) = L(\theta_t)w + \lambda L_0(\theta_t). \tag{29}$$

For this new function, we have

$$l'(\theta_{t+1}) \leq l'(\theta_t) + \alpha_t\langle G(\theta_t)w + \lambda g_0(\theta_t), -d\rangle + \frac{l_1'\alpha_t^2}{2}\|d\|^2$$

$$= l'(\theta_t) + \alpha_t\langle G(\theta_t)w + \lambda g_0(\theta_t), -G(\theta_t;\zeta)w_{t+1} - \lambda g_0(\theta_t;\zeta)\rangle + \frac{l_1'\alpha_t^2}{2}\|d\|^2$$

where $l_1' = \max_i l_{i,1} + \lambda l_{i,1}$. Then taking expectations over $\zeta$ on both sides and rearranging the inequality, we have

$$\mathbb{E}[l'(\theta_{t+1})] - \mathbb{E}[l'(\theta_t)] \leq \alpha_t\mathbb{E}[\langle G(\theta_t)w + \lambda g_0(\theta_t), -G(\theta_t)w_{t+1} - \lambda g_0(\theta_t)\rangle] + \frac{l_1'\alpha_t^2}{2}\mathbb{E}[\|d\|^2]$$

$$= -\alpha_t\mathbb{E}[\langle G(\theta_t)w + \lambda g_0(\theta_t), G(\theta_t)w_{t+1} - G(\theta_t)w_t\rangle]$$

$$-\alpha_t\mathbb{E}[\langle G(\theta_t)w + \lambda g_0(\theta_t), G(\theta_t)w_t + \lambda g_0(\theta_t)\rangle] + \frac{l_1'\alpha_t^2}{2}\mathbb{E}[\|d\|^2]$$

$$= -\alpha_t\mathbb{E}[\langle G(\theta_t)w + \lambda g_0(\theta_t), G(\theta_t)w_{t+1} - G(\theta_t)w_t\rangle]$$

$$-\alpha_t\mathbb{E}[\langle G(\theta_t)w - G(\theta_t)w_t, G(\theta_t)w_t + \lambda g_0(\theta_t)\rangle]$$

$$-\alpha_t\mathbb{E}[\|G(\theta_t)w_t + \lambda g_0(\theta_t)\|^2] + \frac{l_1'\alpha_t^2}{2}\mathbb{E}[\|d\|^2]$$

$$\overset{(i)}{\leq} \alpha_t\underbrace{\mathbb{E}[\|(G(\theta_t)w + \lambda g_0(\theta_t))^T G(\theta_t)\|\|w_t - w_{t+1}\|]}_{C}$$

$$+ \alpha_t\underbrace{\mathbb{E}[\langle G(\theta_t)w_t - G(\theta_t)w, G(\theta_t)w_t + \lambda g_0(\theta_t)\rangle]}_{D}$$

$$-\alpha_t\mathbb{E}[\|G(\theta_t)w_t + \lambda g_0(\theta_t)\|^2] + \frac{l_1'\alpha_t^2}{2}\mathbb{E}[\|d\|^2], \tag{30}$$

where $(i)$ follows from Cauchy-Schwarz inequality. Then we provide bound for term C and term D, respectively. For term C,

$$\mathbb{E}[\|(G(\theta_t)w + \lambda g_0(\theta_t))^T G(\theta_t)\|\|w_t - w_{t+1}\|]$$

$$= \beta_t\mathbb{E}[\|(G(\theta_t)w + \lambda g_0(\theta_t))^T G(\theta_t)\|\|G(\theta_t;\xi)^T(G(\theta_t;\xi')w_t + \lambda g_0(\theta;\xi'))\|]$$

$$\leq \beta_t\mathbb{E}[\|(G(\theta_t)w + \lambda g_0(\theta_t))^T G(\theta_t)\|(\|G(\theta_t;\xi)^T(G(\theta_t;\xi')w_t\| + \lambda\|G(\theta_t;\xi)g_0(\theta;\xi')\|)]$$

$$\leq \beta_t(1 + \lambda)^2 C_g^2(K\sigma_0 + C_g)^2 = \beta_t C_3, \tag{31}$$

where $C_3 = \mathcal{O}((1 + \lambda)^2 K^2)$. Then for term D, we first follow the non-expansive property of projection onto the convex set,

$$\|w_{t+1} - w\|^2 \leq \|w_t - \beta_t G(\theta_t;\xi)^T(G(\theta_t;\xi')w_t + \lambda g_0(\theta_t;\xi')) - w\|^2$$

$$= \|w_t - w\|^2 - 2\beta_t\langle w_t - w, G(\theta_t;\xi)^T(G(\theta_t;\xi')w_t + \lambda g_0(\theta_t;\xi'))\rangle$$

$$+ \beta_t^2\|G(\theta_t;\xi)^T(G(\theta_t;\xi')w_t + \lambda g_0(\theta_t;\xi'))\|^2$$

Then taking expectation on the above inequality, we can obtain,

$$\mathbb{E}[\|w_{t+1} - w\|^2] \leq \mathbb{E}[\|w_t - w\|^2] - 2\beta_t\mathbb{E}[\langle w_t - w, G(\theta_t;\xi)^T(G(\theta_t;\xi')w_t + \lambda g_0(\theta_t;\xi'))\rangle]$$

$$+ \beta_t^2 \mathbb{E}[\|G(\theta_t; \xi)^T (G(\theta_t; \xi')w_t + \lambda g_0(\theta_t; \xi'))\|^2]$$
$$\leq \mathbb{E}[\|w_t - w\|^2] - 2\beta_t \mathbb{E}[\langle w_t - w, G(\theta_t)^T (G(\theta_t)w_t + \lambda g_0(\theta_t))\rangle] + \beta_t^2 C_1^2,$$

where the last inequality follows from Lemma 3. Then by rearranging the above inequality, we can obtain,

$$\mathbb{E}[\langle w_t - w, G(\theta_t)^T(G(\theta_t)w_t + \lambda g_0(\theta_t))\rangle] \leq \frac{1}{2\beta_t}\mathbb{E}[\|w_t - w\|^2 - \|w_{t+1} - w\|^2] + \frac{\beta_t}{2}C_1^2 \quad (32)$$

Then substituting eq. (31) and eq. (32) into eq. (30), we can obtain,

$$\begin{aligned} \mathbb{E}[l'(\theta_{t+1}) - l'(\theta_t)] \leq & \alpha_t \beta_t C_3 + \frac{\alpha_t}{2\beta_t}\mathbb{E}[\|w_t - w\|^2 - \|w_{t+1} - w\|^2] + \frac{\alpha_t \beta_t}{2}C_1^2 \\ & - \alpha_t \mathbb{E}[\|G(\theta_t)w_t + \lambda g_0(\theta_t)\|^2] + \frac{l_1'\alpha_t^2}{2}\mathbb{E}[\|d\|^2] \\ \overset{(i)}{\leq} & \alpha_t \beta_t C_3 + \frac{\alpha_t}{2\beta_t}\mathbb{E}[\|w_t - w\|^2 - \|w_{t+1} - w\|^2] + \frac{\alpha_t \beta_t}{2}C_1^2 \\ & - \alpha_t \mathbb{E}[\|G(\theta_t)w_t + \lambda g_0(\theta_t)\|^2] + \frac{l_1'\alpha_t^2}{2}C_2 \end{aligned} \quad (33)$$

Then we take $\alpha_t = \alpha$ and $\beta_t = \beta$ as constants, telescope and rearrange the above inequality,

$$\begin{aligned} \frac{1}{T}\sum_{t=0}^{T-1}\mathbb{E}[\|G(\theta_t)w_t + \lambda g_0(\theta_t)\|^2] \leq & \frac{1}{\alpha T}\mathbb{E}[l'(\theta_0) - l'(\theta_T)] + \frac{1}{2\beta T}\mathbb{E}[\|w_0 - w\|^2 - \|w_T - w\|^2] \\ & + \beta(C_3 + \frac{C_1^2}{2}) + \frac{l_1'\alpha}{2}C_2 \\ \overset{(i)}{\leq} & \mathcal{O}(\frac{1}{\alpha T} + \alpha K + \frac{1}{\beta T} + \beta K^2), \end{aligned} \quad (34)$$

where $(i)$ follows from that we choose $\lambda$ as a constant. If we choose $\alpha = \Theta(K^{-\frac{1}{2}}T^{-\frac{1}{2}})$ and $\beta = \Theta(K^{-1}T^{-\frac{1}{2}})$, we can get $\frac{1}{T}\sum_{t=0}^{T-1}\mathbb{E}[\|G(\theta_t)w_t + \lambda g_0(\theta_t)\|^2] = \mathcal{O}(KT^{-\frac{1}{2}})$. Furthermore, following the same steps as in Appendix C.3, we have

$$\frac{1}{T}\sum_{t=0}^{T-1}\mathbb{E}[\|G(\theta_t)w_t^*\|^2] = \mathcal{O}(KT^{-\frac{1}{2}}).$$

To achieve an $\epsilon$-accurate Pareto stationary point, it requires $T = \mathcal{O}(K^2\epsilon^{-2})$. In this case, each objective requires a similar number of samples $\mathcal{O}(K^2\epsilon^{-2})$ in $\xi(\xi')$ and $\zeta$, respectively. $\qquad\square$

**Convergence under objective sampling.** We next analyze the convergence of SDMGrad-OS.

**Theorem 8** (Restatement of Theorem 4). *Suppose Assumptions 1-3 are satisfied and $S = 1$. Set $\gamma = \frac{K}{n}$, $\alpha_t = \alpha = \Theta(K^{-\frac{1}{2}}\gamma^{-\frac{1}{2}}T^{-\frac{1}{2}})$, $\beta_t = \beta = \Theta(K^{-1}\gamma^{-1}T^{-\frac{1}{2}})$ and $\lambda$ as a constant. The iterates of the proposed SDMGrad-OS algorithm satisfy,*

$$\frac{1}{T}\sum_{t=0}^{T-1}\mathbb{E}[\|G(\theta_t)w_t^*\|^2] = \mathcal{O}(K\gamma T^{-\frac{1}{2}}).$$

*Proof.* In SDMGrad-OS, the vector for updating $\theta_t$ is $d' = \frac{K}{n}H(\theta_t; \zeta, \widetilde{S})w_{t+1} + \frac{\lambda K}{n}h_0(\theta_t; \zeta, \widetilde{S})$. Using the same function defined in eq. (29), we have

$$l'(\theta_{t+1}) \leq l'(\theta_t) + \alpha_t \langle G(\theta_t)w + \lambda g_0(\theta_t), -d'\rangle + \frac{l_1'\alpha_t^2}{2}\|d'\|^2.$$

Then by taking expectation over $\zeta$ and $\widetilde{S}$, we have

$$\mathbb{E}[l'(\theta_{t+1}) - l'(\theta_t)] \leq \alpha_t \mathbb{E}[\langle G(\theta_t)w + \lambda g_0(\theta_t), -G(\theta_t)w_{t+1} + \lambda g_0(\theta_t)\rangle] + \frac{l_1'\alpha_t^2}{2}\mathbb{E}[\|d'\|^2]$$

$$
\begin{aligned}
=&\alpha_t E[\langle G(\theta_t)w + \lambda g_0(\theta_t), G(\theta_t)(w_t - w_{t+1})\rangle] \\
&+ \alpha_t \mathbb{E}[\langle G(\theta_t)w_t - G(\theta_t)w, G(\theta_t)w_t + \lambda g_0(\theta_t)\rangle] \\
&- \mathbb{E}[\|G(\theta_t)w_t + \lambda g_0(\theta_t)\|^2] + \frac{l_1' \alpha_t^2}{2} \mathbb{E}[\|d'\|^2] \\
\leq&\alpha_t \mathbb{E}[\|(G(\theta_t)w + \lambda g_0(\theta_t))^T G(\theta_t)\|\|w_t - w_{t+1}\|] \\
&+ \alpha_t \mathbb{E}[\langle G(\theta_t)w_t - G(\theta_t)w, G(\theta_t)w_t + \lambda g_0(\theta_t)\rangle] \\
&- \mathbb{E}[\|G(\theta_t)w_t + \lambda g_0(\theta_t)\|^2] + \frac{l_1' \alpha_t^2}{2} \mathbb{E}[\|d'\|^2]
\end{aligned}
\tag{35}
$$

Then following the same steps in eq. (31) and eq. (32), we can obtain,

$$
\begin{aligned}
\mathbb{E}[l'(\theta_{t+1}) - l'(\theta_t)] \leq& \alpha_t \beta_t C_3' + \frac{\alpha_t}{2\beta_t} \mathbb{E}[\|w_t - w\|^2 - \|w_{t+1} - w\|^2] + \frac{\alpha_t \beta_t}{2} C_1'^2 \\
&- \alpha_t \mathbb{E}[\|G(\theta_t)w_t + \lambda g_0(\theta_t)\|^2] + \frac{l_1' \alpha_t^2}{2} C_2',
\end{aligned}
\tag{36}
$$

where $C_1'^2 = 4\gamma^4(1+\lambda^2)(n\sigma_0^2 + C_g)^2$, $C_2' = 4\gamma^2(1+\lambda^2)(n\sigma_0^2 + C_g^2)$, and $C_3' = \gamma^2(1+\lambda)^2 C_g^2(n\sigma_0^2 + C_g)^2$. Then we take $\alpha_t = \alpha$ and $\beta_t = \beta$ as constants and telescope the above inequality,

$$
\begin{aligned}
\frac{1}{T} \sum_{t=0}^{T-1} \mathbb{E}[\|G(\theta_t)w_t + \lambda g_0(\theta_t)\|^2] \leq& \frac{1}{\alpha T} \mathbb{E}[l'(\theta_0) - l'(\theta_T)] + \frac{1}{2\beta T} \mathbb{E}[\|w_0 - w\|^2 - \|w_T - w\|^2] \\
&+ \beta(C_3' + \frac{C_1'^2}{2}) + \frac{l_1' \alpha}{2} C_2' \\
\overset{(i)}{\leq}& \mathcal{O}(\frac{1}{\alpha T} + \alpha\gamma K + \frac{1}{\beta T} + \beta\gamma^2 K^2),
\end{aligned}
\tag{37}
$$

where $(i)$ follows from that we choose $\lambda$ as constant. Similarly, if we choose $\alpha = \Theta(K^{-\frac{1}{2}}\gamma^{-\frac{1}{2}}T^{-\frac{1}{2}})$ and $\beta = \Theta(K^{-1}\gamma^{-1}T^{-\frac{1}{2}})$, we can get $\frac{1}{T}\sum_{t=0}^{T-1}\mathbb{E}[\|G(\theta_t)w_t + \lambda g_0(\theta_t)\|^2] = \mathcal{O}(K\gamma T^{-\frac{1}{2}})$. Furthermore, following the same step as in Appendix C.3, we have

$$
\frac{1}{T} \sum_{t=0}^{T-1} \mathbb{E}[\|G(\theta_t)w_t^*\|^2] = \mathcal{O}(K\gamma T^{-\frac{1}{2}}).
$$

To achieve an $\epsilon$-accurate Pareto stationary point, it requires $T = \mathcal{O}(\gamma^2 K^2 \epsilon^{-2})$. In this case, each objective requires a similar number of samples $\mathcal{O}(\gamma^2 K^2 \epsilon^{-2})$ in $\xi(\xi')$ and $\zeta$, respectively. $\qquad\square$

