# OpenReview forum: "Direction-oriented Multi-objective Learning: Simple and Provable Stochastic Algorithms"
_NeurIPS.cc/2023/Conference — NeurIPS 2023 poster_

### Official Review · Reviewer_81UB · 2023-06-23

**Soundness:** 2 fair
**Presentation:** 2 fair
**Contribution:** 2 fair
**Rating:** 2
**Confidence:** 4

**Summary:**

This paper proposes a gradient manipulation method named SDMGrad for multi-task learning (MTL). SDMGrad improves the previous MGDA method by using two constraints. The first one is to constrain the common descent direction nearby the one computed with a specific preference (such as the average direction), which is similar to the previous CAGrad method. The second one is to add a regularization term for the computed weights, which is the same as the previous MoCo method. To reduce the computational cost, partial tasks are randomly selected to compute their gradients in SDMGrad, which is called SDMGrad-OS and is similar to the previous RLW method.

Theoretically, a stochastic non-convex convergence analysis for the proposed methods is provided. Empirically, experiments on multi-task supervised learning and reinforcement learning are conducted but some important baselines are missing.

**Strengths:**

1. This paper proposes a method for MTL.
2. This paper provides a stochastic non-convex convergence analysis for the proposed methods.
3. The code is provided.

**Weaknesses:**

1. The novelty of the proposed methods is limited. This paper aims to improve the previous MGDA method by adding several components. However, those components are similar to some existing work.
2. Lack of some important baselines, especially Nash-MTL (ICML 2022), which outperforms the proposed SMDGrad in all benchmark datasets.
3. The regularization term with a coefficient $\frac{\rho}{2}$ in Eq. (9) is one of the proposed improvements. However, the paper does not mention how to set $\rho$. Besides, from the code in the supplementary material, it seems $\rho=0$ in the implementation.

**Questions:**

1. The novelty of the proposed methods is limited. The proposed SDMGrad method improves the previous MGDA method by using two constraints.
- constraining the common descent direction nearby the one computed with a specific preference, as shown in Eq. (6). This is very similar to CAGrad in Eq. (7). The only difference is the norm constraint used in CAGrad while the angle constraint used in SDMGrad. Thus, what is the motivation and advantage to use Eq. (6)?
- adding a regularization term for the computed weights in Eq. (9), which is the same as MoCo.
- SDMGrad-OS reduces the computational cost in SDMGrad by randomly sampling tasks, which is similar to RLW [1]. In RLW, loss/gradient weights are randomly sampled from a distribution (such as Normal and Bernoulli) at each iteration. If the sample distribution is Bernoulli-family, it is equivalent to randomly sampling task losses/gradients to optimize at each iteration.

&ensp;&ensp; In a word, it is hard to find something new or interesting in Section 4.

2. What is the meaning of "propose a new direction-oriented multi-objective problem" in Lines 3-4, Line 139, and Line 313? Maybe "formulation" is more appropriate than "problem".
3. $L_i$ in the last of Line 23 is quite confusing. It is better to use more clear notation.
4. Lines 24-25: "MTL aims to solve an average loss for all $K$ tasks" is wrong.
5. Line 27: why MOO **cannot** find a common parameter $\theta$ that achieves optima for all objectives? MOO can, but rarely. The description of "parameter $\theta$ that minimizes all objective functions" is wrong.
6. Eq. (4): what is the meaning of $h_{t, i}$?
7. Line 133: $\mathbb{R}^K\rightarrow\mathbb{R}^m$
8. Lines 145-147: in what scenarios? It is better to provide some examples here.
9. Line 176 in the text and Step 5 in Algorithm 1: why two different sampled data is used for the update of $\omega$? From the code in the supplementary material, the same sampled data is used in the implementation.
10. Proposition 1: it seems the bound is meaningless when $\rho\rightarrow0$.
11. Lines 447-448 in Appendix: the experimental setting of adding zero-mean Gaussian noise is following MoCo but without citation. Besides, MoCo does not been compared in this toy example section.
12. Section B in Appendix: "$g_0$ denotes the average gradient" in Line 483 but $g_0=G_0(\theta)=G(\theta)\tilde{\omega}$ in Eq. (13). What does $\tilde{\omega}$ denote? Is $G_0$ a vector here? It is confusing because capital $G$ denotes a matrix and lowercase $g$ denotes a vector in Eq. (13). The last line in Eq. (13): $G(\theta_t)\rightarrow G(\theta_t)\omega$.
13. Eq. (17) in Appendix: Is it possible to provide a detailed derivation of step (i)?
14. Hyperparameter selection.
- how to set $\rho$? From the code in the supplementary material, it seems $\rho=0$ in the implementation. So why? If so, the regularization term in Eq. (9) and the bound in Proposition 1 are meaningless.
- $\lambda$ in the proposed SDMGrad is searched for the best. How about the hyperparameters of baselines? From Table 8 in Appendix, CAGrad performs significantly better than the results in Table 4 in the main text.
- how to set $n$ for SMDGrad-OS?
15. Line 289: the definition of $\Delta m$ is wrong since some task has more than one metric (for example those tasks in Cityscapes and NYU-v2).
16. Lack of some important baselines.
- Nash-MTL [2] outperforms the proposed SMDGrad in all benchmark datasets, i.e., Cityscapes, NYU-v2, MT10, and MT50. Besides, the running time of Nash-MTL is shorter than SMDGrad-OS in MT10.
- It is better to compare with more baselines such as RLW [1], GradNorm [3], IMTL [4], and so on.
17. SegNet is out-of-dated and performs unsatisfactory. It is better to conduct experiments using a more powerful backbone, such as resnet50 and transformer.
18. The results of SMDGrad-OS in Cityscapes and NYU-v2 datasets should be reported.
19. The claim of SMDGrad-OS can deal with the case where the number of objectives is large (Lines 10-11) while MoCo does not (Lines 44-47). However, there is no experiment to support this.
20. Lines 94-96: there is no experiment to support that the proposed SMDGrad is model-agnostic.





----
[1] Reasonable Effectiveness of Random Weighting: A Litmus Test for Multi-Task Learning. TMLR 2022.

[2] Multi-Task Learning as a Bargaining Game. ICML 2022.

[3] GradNorm: Gradient Normalization for Adaptive Loss Balancing in Deep Multitask Networks. ICML 2018.

[4] Towards Impartial Multi-task Learning. ICLR 2021.

---

> ### Author Rebuttal · Authors · 2023-08-09
>
> **Answer to Q1** We would like to clarify the novelty of our algorithmic designs and the difference from previous works.
> 1. As we discussed in lines 152-158, optimizing the constraint-based regularization (see eq. (7)) in CAGrad involves the evaluations of product $\\|h_0\\|\\|g_w\\|$ and the ratio $\\|h_0\\|/\\|g_w\\|$, which can heavily complicate the design of unbiased stochastic gradient/multi-gradient in $w$ and $\theta$ updates. As a result, CAGrad does not have performance guarantees in stochastic setting. As a comparison, our angle-based regularization admits very simple optimization steps in the stochastic setting (see line 5 and 7 of our Algorithm 1), while guaranteeing the convergence for general regularization constant $\lambda$. Thus, developing a simple and provable stochastic MOO method under such direction-oriented mechanism is the motivation and advantage of our design in eq. (6).
> 2. Our regularization term in Eq. (9) and the one in MoCo serves different purposes, and also lead to different analyses. In specific, our regularizer is to ensure the last-iterate convergence for solving Eq. (9), whereas the regularizer in MoCo is to ensure the Lipschitz continuity of the solution $w^*_{\rho,0}(\theta)$ (under our notations) w.r.t. $\theta$.
> 3. We agree that the task sampling is not new, but it is new to show that such sampling guarantees the near-unbiasedness of the gradient/multi-gradient in lines 5 and 7 of our Algorithm, and achieves an improved convergence guarantee. Previous works do not have such results.
>
> **Answer to Q2, 3, 7.** Many thanks. We will follow your suggestions to improve the presentation.
>
> **Answer to Q4.** We will revise this sentence to “The objective function of MTL is taking as the average loss over K objectives”.
>
> **Answer to Q5.** Sorry for the improper wording. We will revise it to “MOO rarely finds a common parameter $\theta$ that achieves optima for all individual objective functions simultaneously”.
>
> **Answer to Q6.** Please refer to our answers to Q3 from the reviewer ft1r.
>
> **Answer to Q8.** The most relevant example is multi-task learning, whose objective function is an average loss for all tasks. Another possible case is that we have prior knowledge about the importance of different tasks, and the target is to optimize a weighted sum of loss functions for all tasks.
>
> **Answer to Q9.** In theory, please refer to our answers to Q2 from the reviewer ZeY5. In experiments, we found that using two different mini-batches performs similarly to using the same mini-batch. The result can be found in Table 1 in the global response.
>
> **Answer to Q10.** Note that in theory, the upper bounds are often derived in a uniform sense, i.e., hold for a class of objectives satisfying the assumptions. Thus, the $\rho>0$ is necessary here to ensure the worst case in this class to achieve a performance guarantee. However, in practice, the problem may not be the worst case, and hence $\rho\rightarrow 0$ may still work well.
> It is also worth mentioning that for another class of objectives under a bounded function value assumption, we do not need this $\rho$ (Theorem 3 and 4).
>
> **Answer to Q11.** We will add the citation of MoCo. The results of reproduced MoCo can be found in Figure 1 in the global response PDF.
>
> **Answer to Q12.** Since we use $g_0$ to denote average gradient, $\widetilde{w}$ denotes a vector whose elements are all $\frac{1}{K}$, where $K$ is the number of objectives. Yes, $G_0$ is a vector. To remove the confusion, we will revise $G_0(\theta)$ to $g_0(\theta)$ in the revision.
>
> **Answer to Q13.** Note that eq. (9) is strongly convex w.r.t. w. Using the property of a smooth $\mu$-strongly-convex function $f$ that $\forall x, y, (\nabla f(x)-\nabla f(y))^T(x-y)\geq\mu\\|x-y\\|^2$, we can derive $(i)$. We will clarify it in our revision.
>
> **Answer to Q14.** Our selection takes following steps:
> The parameter $\rho$ is used only for theoretical analysis and for guaranteeing our method to work well in the worst case. In the experiments (which are not necessarily the worst case), we find the performance when $\rho=0$ is good enough. Thus, we make this selection for simplicity.
> From the formulation of our proposed SDMGrad, we know that it becomes close to SGD when $\lambda$ is large, and close to MGDA with a small $\lambda$. We first try to identify the large and small values of $\lambda$ where the performance shows consistency with our formulation. In experiments, we find $\lambda=0.1$ and $\lambda=10$ work well. Next, we narrow the range by trying different values in [0.1, 10]. Specifically, we search with $\lambda \in [0.1, 1, 2, 5]$ and evaluate which choice is better. For $n$, we make the same selection as the baseline CAGrad.
>
> **Answer to Q15.** Thanks! We revise the definition as:
> $\Delta_m=\frac{1}{K}\sum_{k=1}^K(-1)^\delta_k (M_{m,k}-M_{b,k})/M_{b,k}$, where $K$ is the number of metrics, $M_{b,k}$ is the value of metric $M_k$ obtained by the baseline and $M_{m,k}$ obtained by the compared method. $\delta_k=1$ if a higher value is better for metric $M_k$ and 0 otherwise.
>
> **Answer to Q16.** For supervised learning, we add baselines including RLW, IMTL-G, and Nash-MTL in supervised learning and reinforcement learning. It can be seen from Table 3 in the global response PDF that our SDMGrad and SDMGrad-OS achieve higher success rate than NashMTL, while using much less time.
>
> **Answer to Q17.** We follow former works using SegNet as backbone and it is fair to compare.
>
> **Answer to Q18.** The numbers of tasks in Cityscapes and NYU-v2 are 2 and 3 respectively, which are already very small, and hence we do not apply task sampling in these two datasets.
>
> **Answer to Q19.** Please refer to added experiment in the global response.
>
> **Answer to Q20.** Following the CAGrad paper, all gradient manipulation methods such as CAGrad, MoCo, PCGrad, and MGDA are called as model agnostic because they manipulate gradients rather than models to avoid conflict.

---

> > ### Comment · Reviewer_81UB · 2023-08-11
> > **Thanks for reply**
> >
> > Thanks for the response. It addresses some of my concerns. But I still have many concerns.
> >
> > ---
> > **Response to "Answer to Q4"**: Does it mean only $\frac{1}{K}\sum_{i=1}^K \ell_i$ (where $\ell_i$ is the loss of i-th task) is the objective function of MTL? How about $\frac{1}{K}\sum_{i=1}^K w_i\ell_i$, where $w_i$ is the task weight of i-th task.
> >
> > **Response to "Answer to Q8"**: Same question as above.
> >
> > **Response to "Answer to Q9"**: Please see **Response to "Answer to Q14"** below.
> >
> > **Response to "Answer to Q11"**: It seems MoCo and the proposed SDMGrad perform comparably since they are both stochastic methods. Thus, this toy example cannot demonstrate the advantages of the proposed method over MoCo.
> >
> > **Response to "Answer to Q14"**:
> >
> > - This answer means **the proposed method is designed for theoretical analysis and is not used in practice**. In other words, in practice, SGD is used to solve Eq. (8) and **we do not need anything introduced in Section 4.2**.
> >
> > - It is better to provide a comprehensive ablation study of $\lambda$ (only two values of $\lambda$ are conducted in the paper).
> >
> > - **This question has not been answered.**  Q14: "How about the hyperparameters of baselines? From Table 8 in Appendix, CAGrad performs significantly better than the results in Table 4 in the main text."
> >
> > - I cannot understand it. "For $n$, we make the same selection as the baseline CAGrad." in Answer to Q14.
> >
> > **Response to "Answer to Q16"**:
> >
> > - What are the meanings of SGD and Unit. Scal. in Tables 1, 2, and 3 in the global response PDF?
> >
> > - It seems $\Delta_m$ of RLW in Table 2 in the global response PDF is 7.78, according to Table 2 in the Nash-MTL paper.
> >
> > - **Nash-MTL outperforms the proposed method on both Cityscapes and NYU-v2 datasets, according to the results of Tables 1 and 2 in the global response PDF**.
> >
> > - It seems the running time of Nash-MTL reported in Table 3 in the global response PDF is not convincing. Table 5 in the Nash-MTL paper shows that Nash-MTL can be as efficient as PCGrad by tuning the hyperparameter (the frequency of task weights updates). It seems the computational cost of PCGrad is similar to the proposed method. Thus, I do not think Nash-MTL is 7-9 times slower than the proposed method.
> >
> > **Response to "Answer to Q18"**: If my understanding is correct, the proposed SDMGrad-OS method can work when the number of tasks is small. It is important to evaluate it comprehensively. If SDMGrad-OS cannot work with small task numbers, it is necessary to explain why and the definition of "small".
> >
> > **Response to "Answer to Q19"**: Which experiment?
> >
> > **New Question**: Line 435 in Appendix: what is the detail of rescaling? Lines 435-437 in Appendix: MoCo uses softmax for projection operation (see Page 30 of MoCo paper in the ICLR version).
> >
> > ---
> >
> > **All in All, I appreciate the theoretical contributions of this paper. However, I have two major concerns about this paper.**
> >
> > - **The proposed method is inconsistent with the implementation**. There are **four inconsistencies** as follows.
> >
> > 	1. In the implementation, each task's gradient is normalized by its norm before Line 5 of Algorithm 1.
> >
> > 	2. $\rho$ in Line 5 of Algorithm 1 is set as $0$ in implementation.
> >
> > 	3. Two mini-batches strategy in Line 5 of Algorithm 1 is not used in the implementation.
> >
> > 	4. $\alpha_t$ in Line 7 of Algorithm 1 is set as $\frac{\alpha_t}{1+\lambda}$ in the implementation, where $\lambda$ is the same as in Line 5 of Algorithm 1 and is a hyperparameter that needs to be carefully tuned.
> >
> >   **The second and third points indicate that the proposed method in Section 4.2 is for better convergence properties but is not used in practice**.  I know **the first point** is a "potential" trick in MTL and significantly affects the performance. Thus, I think it should be mentioned in the paper. **The last point** means tuning $\lambda$ has the same effect as directly tuning the learning rate $\alpha_t$, which also significantly affects the performance. According to the empirical results [1], the most simple method, $\min\sum_{i=1}^K \ell_i$, can perform comparably or even better than all those multi-objective optimization methods for both supervised learning and reinforcement learning by tuning some hyperparameters (like the learning rate).
> >
> >   [1] In Defense of the Unitary Scalarization for Deep Multi-Task Learning. NeurIPS 2022.
> >
> > - **The experimental results cannot demonstrate the effectiveness of the proposed method.**
> >
> > 	1. **Nash-MTL outperforms the proposed method** on both Cityscapes and NYU-v2 datasets, according to the results of Tables 1 and 2 in the global response PDF.
> >
> > 	2. **The hyperparameters of the proposed method are carefully tuned, while the baselines are not**. See the results of CAGrad in Tables 3 and 4 in the paper and Tables 7 and 8 in Appendix.

---

> > > ### Author Response · Authors · 2023-08-14
> > > **Thanks for your feedback (Part1)**
> > >
> > > We thank the reviewer for further reply and for appreciating our theoretical contributions.
> > > ___
> > >
> > > **Response to "Answer to Q4" and "Answer to Q8"**: Here, we refer to the commonly used average loss $\frac{1}{K}\sum_{i=1}^K \ell_i$, following the CAGrad paper. However, we can also use the weighted average loss $\frac{1}{K}\sum_{i=1}^K w_i\ell_i$ when the preferences $w_i,i=1,...,K$ are known. In fact, our framework also covers this case by setting the weight $\widetilde{w}_i$ in our $g_0=\sum_i\widetilde{w}_i\nabla l_i$ to be $\frac{1}{K}w_i$. In other words, our framework covers both formulations.
> > >
> > > **Response to "Answer to Q9" and "Answer to Q14"**:
> > > 1. We think you have a misunderstanding of our answer to Q14. As we emphasized in response, the parameter selection (like $\rho>0$) is to ensure that our method can **also** work well in the worst-case scenario. This type of theoretical results (i.e., convergence upper bounds) are very common in optimization, and is often used to evaluate the theoretical performance of different algorithms.
> > > However, the experiments are not necessarily the worst cases, and hence in practice, people usually search the hyperparameters.
> > > In our experiments, we found that the choice of $\rho=0$ performs similarly to a small positive $\rho>0$, so we set $\rho=0$ for simple implementation. To totally resolve your concern here, we have added an additional experiment (see the table below) with $\rho=0.01$, which achieves a performance similar to $\rho=0$.
> > > | Method | Segmentation $\uparrow$ |  | Depth $\downarrow$ |  | $\Delta \mathrm{m}$% $\downarrow$ |
> > > | :---: | :---: | :---: | :---: | :---: | :---: |
> > > |  | mloU | Pix Acc | Abs Err | Rel Err |  |
> > > | Independent | 74.01 | 93.16 | 0.0125 | 27.77 |  |
> > > | SDMGrad ($\lambda=0.3,\rho=0.01$) |75.15| 93.45 | 0.0161 | 29.59 | 8.42 |
> > > | SDMGrad ($\lambda=0.3,\rho=0$) |75| 93.43 | 0.0135 | 35.35 | 8.39 |
> > > ||||||
> > > 2. Following your suggestion, we provide a comprehensive ablation study of $\lambda$ on Cityscapes in the following table.
> > > | Method | Segmentation $\uparrow$ |  | Depth $\downarrow$ |  | $\Delta \mathrm{m}$% $\downarrow$ |
> > > | :---: | :---: | :---: | :---: | :---: | :---: |
> > > |  | mloU | Pix Acc | Abs Err | Rel Err |  |
> > > | Independent | 74.01 | 93.16 | 0.0125 | 27.77 |  |
> > > | SDMGrad ($\lambda=0.1$) |75.53| 92.94 | 0.0165 | 37.4 | 16.99 |
> > > | SDMGrad ($\lambda=0.2$) |73.78| 92.95 | 0.0141 | 36.53 | 11.17 |
> > > | SDMGrad ($\lambda=0.3$) |75| 93.43 | 0.0135 | 35.35 | 8.39 |
> > > | SDMGrad ($\lambda=0.4$) |75.24| 93.52 | 0.0133 | 38.73 | 10.94 |
> > > | SDMGrad ($\lambda=0.5$) |74.8| 93.42 | 0.0138 | 37.05 | 10.62 |
> > > | SDMGrad ($\lambda=0.6$) |74.63| 93.46 | 0.014 | 37.3 | 11.32 |
> > > | SDMGrad ($\lambda=0.7$) |75.49| 93.58 | 0.0142 | 37.33 | 11.46 |
> > > | SDMGrad ($\lambda=0.8$) |75.18| 93.52 | 0.0137 | 37.89 | 11.03 |
> > > | SDMGrad ($\lambda=0.9$) |75.2| 93.53 | 0.0139 | 39.51 | 12.92 |
> > > ||||||
> > > 3. The results of all baselines in Table 4-8 are directly quoted from their original papers. CAGrad has a hyperparameter $c$ similar to our $\lambda$ in SDMGrad. The value of $c$ in CAGrad is set to 0.2 in Table 3, and 0.4 in Table 4.
> > > The results of CAGrad shown in Table 3-4 are the ones with the best average training loss. While the results of CAGrad shown in Table 7-8 are the ones with the best performance drop $\Delta_m$. This is consistent with the presentation in the original CAGrad paper. We report both two results of the proposed SDMGrad in Table 3-4. In fact, we have discussed this in Lines 291-292. We will clarify this in revision.
> > > 4. The original CAGrad paper proposed a speedup strategy via sampling a subset of tasks. Following CAGrad, we set $n=4$ for MT10 and $n=8$ for MT50.
> > >
> > > **Response to "Answer to Q11"**: In this toy example, our goal is to show that our SDMGrad method converges to the target solution, and hence does not fall short compared to MoCo. However, note that in other experiments on Cityscapes, NYU-v2, and MT10, our SDMGrad clearly outperforms MoCo a lot.
> > >
> > > **Response to "Answer to Q16"**:
> > > 1. SGD means that SGD is used to optimize the average loss of all tasks during the training. Unit. Scal. denotes the unitary scalarization method proposed in [1].
> > > 2. Sorry for the carelessness. Yes, the $\Delta_m$ of RLW should be 7.78, and 10.11 is the MR (mean rank) of RLW. We will revise it.
> > > 3. NashMTL achieves a better performance drop than the proposed SDMGrad on both Cityscapes and NYU-v2. However, SDMGrad performs better than or similar to NashMTL in the segmentation task. Moreover, SDMGrad outperforms NashMTL both in success rate and time efficiency on MT10.
> > > 4. The official implementations of NashMTL on MT10 are not released by the original paper, and hence we use the most recent re-implementation and reproduced results in [2]. The reproduced results in Table 3 are consistent with those in [2]. We will clarify this in revision.
> > > [1] In defense of the unitary scalarization for deep multi-task learning. NeurIPS.
> > > [2] FAMO: Fast Adaptive Multitask Optimization. arXiv.

---

> > > > ### Author Response · Authors · 2023-08-14
> > > > **Thanks for your feedback (Part2)**
> > > >
> > > > **Response to "Answer to Q18"**: We have clarified in our previous response that the total number of tasks in Cityscapes and NUY-v2 are **2 and 3**. Thus, task sampling is unnecessary here, and it may also take the risk that no tasks are sampled.
> > > >
> > > > **Response to "Answer to Q19"**: It refers to the experiment on MT10 shown by Table 3 in the attached PDF.
> > > >
> > > > **Response to New Question**: During the training process, the gradient norms of  tasks may change over time. Thus, directly solving Eq.(9) may trigger numerical problems. Thus, rescalling here means that we normalize the gradient of each task before solving the Eq.(9) to stablize the training. This is a strategy inspired by CAGrad.
> > > > The softmax used in MoCo is not proper here. Take the 2-task case as an example. Suppose the task weights now are $[0.75, 0.25]$, which satisfies the simplex constraint. The task weights keep unchanged after the Euclidean projection. However, the task weights will be $[0.62, 0.38]$ after softmax projection.
> > > >
> > > > **Response to Concern 1**:
> > > >  - **To your first point**: The reason why we normalize gradients of each task is explained in the response to the new question. We have discussed the trick in the supplements.
> > > >  - **To your second and third points**: As we explained in previous answers, the experiments are not necessarily the worst cases, and proper but useful implementations are also often preferred in practice. In our response, we have also provided the empirical results on Cityscapes that 1) use the two mini-batches strategy and 2) use a positive $\rho=0.01$ (see new table in previous answers).
> > > >  - **To your last point**: The selection of $\frac{\alpha_t}{1+\lambda}$ directly follows from CAGrad. In the experiments, we find this strategy works quite well.
> > > >
> > > > **Comparison to Unitary Scalarization**:
> > > > In our response, we have compared our method with SGD and unitary scalarization in [1] in our experiments (see Table 1, 2, 3 in the attached pdf of global response), where it can be seen that our method is much better.
> > > > [1] In Defense of the Unitary Scalarization for Deep Multi-Task Learning. NeurIPS.
> > > >
> > > > **Response to Concern 2**:
> > > >  - NashMTL achieves a better performance drop than the proposed SDMGrad on both Cityscapes and NYU-v2. However, SDMGrad performs better than or similar to NashMTL in the segmentation task. Moreover, SDMGrad outperforms NashMTL a lot both in success rate and time efficiency on MT10 from our experiments.
> > > >  - The hyperparameters in one of our baselines CAGrad are also carefully tuned. More details about such tuning can be seen in Table 5-6 in the Appendix of their paper. For fair comparison, we also select the results with the best average training loss and the best performance drop.

---

> > > > > ### Comment · Reviewer_81UB · 2023-08-16
> > > > > **Thanks for your further reply**
> > > > >
> > > > > Thanks for the further response with more experimental results. **However, it raises more concerns.**
> > > > >
> > > > > ---
> > > > >
> > > > > **Response to "Answer to Q4" and "Answer to Q8":** As you mentioned, the weighted sum loss $\frac{1}{K}\sum_{i=1}^K w_i\ell_i$ is also the objective function of MTL. Thus, the sentence “The objective function of MTL is taking as the average loss over K objectives” in **Answer to Q4** and **Answer to Q8** is wrong.
> > > > >
> > > > > ---
> > > > >
> > > > > **Response to "Answer to Q9" and "Answer to Q14":**
> > > > >
> > > > > 1: The regularization term in Eq. (9) is one of the components of the proposed method and its motivation is slightly different from one in MoCo (as you mentioned in **Answer to Q1**). Although you claimed that a small positive $\rho$ performs similarly with $\rho=0$, **actually $\rho=0.01$ is worse than $\rho=0$ according to your results**, which means **this proposed component should not be used in empirical experiments**.
> > > > >
> > > > >  *The same problem for the two mini-batch strategy.* As your answers to Q2 from the reviewer ZeY5, the two mini-batch strategy is to ensure an unbiased estimation. While you claimed the results with and without this strategy are similar (**actually sometimes better and sometimes worse**). Does it mean the bias problem is unimportant in practice? **Why it can be simply omitted in the implementation?**
> > > > >
> > > > > 3: **The presentation of results in Tables 3 and 4 of the paper may cause misunderstanding.** For the proposed SDMGrad, the results with the best hyperparameters searched with both the best average training loss and the best performance drop are presented in Tables 3 and 4. However, for the baseline CAGrad, only the result with the best average training loss is presented in these Tables. How about the results of other baselines? Are they with the best average training loss or the best performance drop?
> > > > >
> > > > > 4: Your response indicates that **The speedup strategy of sampling a subset of tasks in CAGrad is very similar to SDMGrad-OS.** However, it does **not be discussed and compared** in this paper.
> > > > >
> > > > > ---
> > > > >
> > > > > **Response to "Answer to Q16":**
> > > > >
> > > > > 3: NashMTL **largely outperforms** the proposed SDMGrad on both Cityscapes and NYU-v2.
> > > > >
> > > > >   > However, SDMGrad performs better than or similar to NashMTL in the segmentation task.
> > > > >
> > > > >   It is meaningless since the overall performance is more important in MTL. For example, in Table 4 of the paper, MGDA achieves undesired overall performance, although it largely outperforms other methods in the Surface Normal task.
> > > > >
> > > > >   > Moreover, SDMGrad outperforms NashMTL both in success rate and time efficiency on MT10.
> > > > >
> > > > >   SDMGrad slightly outperforms NashMTL in success rate on MT10. But I do not think the comparison of time efficiency is fair. Please see the reply below.
> > > > >
> > > > > 4:
> > > > > - The running time cannot be directly copied from other papers since it may be computed on different devices. Thus, the running time of both baselines and the proposed method must be computed in the same device for a fair comparison.
> > > > >
> > > > > - It seems the specific value of the running time of NashMTL does not be reported in FAMO. Thus, how was the NashMTL running time of 86.2 computed?
> > > > >
> > > > > - Figure 3 and Table 4 in FAMO also show that NashMTL can be much more efficient without a drop in success rate via tuning an important hyperparameter. Thus, I do not think Nash-MTL is 7-9 times slower than the proposed method.
> > > > >
> > > > > ---
> > > > >
> > > > > **Response to "Answer to Q18":** How many tasks are necessary to use SDMGrad-OS?
> > > > >
> > > > > ---
> > > > >
> > > > > **Response to New Question:**
> > > > >
> > > > > - Does gradient normalization affect the theoretical analysis?
> > > > >
> > > > > - > This is a strategy inspired by CAGrad.
> > > > >
> > > > >   I have checked the official implementation of CAGrad. It seems CAGrad does not use gradient normalization.
> > > > >
> > > > > ---
> > > > >
> > > > > **Response to Concern 1:**
> > > > >
> > > > > - It is necessary to **clearly mention these four points** in the paper's main text, especially in Algorithm 1. Otherwise, readers cannot completely understand the proposed method.
> > > > >
> > > > > - It is necessary to **conduct a comprehensive ablation study**. There are two tricks followed from CAGrad in the implementation and they work well (the first and last points). While two proposed components can be simply omitted in the implementation (the second and third points). Thus, do those two tricks contribute most to the performance of the proposed method?
> > > > >
> > > > > ---
> > > > >
> > > > > Since the rebuttal raises more concerns, I decrease my score to 2.

---

> > > > > > ### Author Response · Authors · 2023-08-20
> > > > > > **Thanks for your further feedback (Part 1)**
> > > > > >
> > > > > > We thank the reviewer for further feedback.
> > > > > > ___
> > > > > >
> > > > > > **Response to "Answer to Q4" and "Answer to Q8"**: Ok, we will revise the sentence to  “The objective function of MTL is **often** taking as the average loss over K objectives”.
> > > > > >
> > > > > >
> > > > > > **Response to "Answer to Q9" and "Answer to Q14"**:
> > > > > > 1. **To your first and second points**:
> > > > > >    - We have emphasized before that the experiments are not necessarily the worst cases, and hence in practice, it is possible to choose $\rho=0$. In theory, it is necessary. In addition, in the table we provide, the choice of $\rho=0.01$ is overall better than $\rho=0$ with better performance on Segmentation and Rel Err. The performances on $\Delta m$ are also very close to each other (8.42 v.s. 8.39).
> > > > > >    - We cannot say the bias problem is not important in practice, but it may not be a big issue in the experiments of this paper. However, to achieve a performance guarantee (that can hold also in worst cases), this strategy is necessary. In our experiments, since we found that the same and different mini-batches achieve similar performance, we adopt the same mini-batch for a simpler implementation.
> > > > > >    - We have implemented $\rho>0$ and the two-mini-batch strategy in the codes and will add them in the revision.
> > > > > > 2. **To your third point**:
> > > > > >    - We reported the result for CAGrad with the best performance drop in Table 7 and 8 in Appendix due to the space limitations.
> > > > > >    - Other baselines do not involve a regularization parameter like SDMGrad and CAGrad, and
> > > > > > hence have only one result, as reported by their corresponding papers.
> > > > > > 3. **To your last point**:
> > > > > >    - The primary goal regarding the task sampling is to show our proposed SDMGrad framework admits a **provable** SDMGrad-OS method with such a task sampling strategy. As a comparison, CAGrad uses it as a practical acceleration without any guarantee.
> > > > > >    - Based on your question, we also provide such a comparison in the following table. We apply the speedup strategy used in CAGrad to the proposed SDMGrad, denoted as SDMGrad-Fast.
> > > > > > | Method | Metaworld MT10 |  |
> > > > > > | :---: | :---: | :---: |
> > > > > > |  | success (mean $\pm$ stderr) | time |
> > > > > > | SDMGrad | $0.84 \pm 0.049$ | 13.3 |
> > > > > > | SDMGrad-OS | $0.82 \pm 0.04$ | 9.7 |
> > > > > > | SDMGrad-FAST | $0.82 \pm 0.075$ | 10.2 |
> > > > > > ||||
> > > > > >
> > > > > >
> > > > > > **Response to "Answer to Q16"**:
> > > > > > 1. **To your first point**:
> > > > > >    - It can be seen from Table 1 and 2 in the global response that SDMGrad performs quite comparable to or slightly worse than NashMTL, and both of them outperform other baselines like SGD, Unit. Scal., RLW, IMTL-G a lot. However, it is also worthy to mention that SDMGrad admits a provable convergence in the stochastic case, whereas NashMTL does not have this feature.
> > > > > >    - On the other hand, SDMGrad clearly outperforms NashMTL in success rate and time efficiency on MT10, and the time efficiency comparison to NashMTL is fair (see the response to the next question).
> > > > > > 2. **To your second point**:
> > > > > >    - We **do not** copy running time from other papers, but run the re-implementation of NashMTL in FAMO paper on our own machines. Other papers do not have this running time.
> > > > > >    - We independently conduct the experiments of the proposed SDMGrad and NashMTL on MT10, running each experiment 5 times on the GTX 2080Ti. Then the running time is calculated by taking the average of each round.
> > > > > >    - The hyperparameter you mention is the period for applying NashMTL. Based on your suggestion, we also compare our method to NashMTL (every 100) with this trick. Our experiments show that the running time of NashMTL (every 100) is 9.9s/epoch in our device, which is still worse than 9.7s/epoch of our SDMGrad-OS. In addition, from Fig. 4 in the FAMO paper, the time cost of NashMTL increases dramatically when the number of tasks increases, and is much worse than other baselines when the number of tasks is large.
> > > > > >
> > > > > >
> > > > > > **Response to "Answer to Q18"**: We think the necessity to use the proposed objective sampling strategy is dependent on the specific task. Besides, the proper size of the sampled subset is also needed to tune. However, in the case of Cityscapes and NYU-v2, where the number of tasks is 2 and 3 respectively, we think it unnecessary to use the objective sampling.
> > > > > >
> > > > > >
> > > > > > **Response to New Question**:
> > > > > >  - The gradient normalization does not affect the theoretical analysis.The purpose of the gradient in solving Eq.(9) is to avoid potential numeric issues. Since the operation does not change the solution of Eq.(9), the theoretical analysis remains unaffected.
> > > > > >  - The gradient normalization strategy of CAGrad is detailed in the `cagrad.py` script under the folder `mtrl/mtrl_files`.
> > > > > >
> > > > > > **Response to Concern 1**:
> > > > > >  - Thank you for your suggestions and we will incorporate more implementation details in our revision.
> > > > > >  - These tricks are overall useful to stabilize the training, and to have a fair comparison to CAGrad with these strategies, we also adopt them in our method.  However, we will surely take your suggestions into consideration in our revisions.

---

> > > > > > > ### Author Response · Authors · 2023-08-20
> > > > > > > **Thanks for your further feedback (Part 2)**
> > > > > > >
> > > > > > > Finally, we want to emphasize that the main contribution of our paper is to develop a **provable** MOO algorithm with **promising theoretical guarantee**, as well as competitive practical performance. We wish the reviewer also recognizes our theoretical contributions, which are overlooked in this discussion. For the experiments, we have tried our best to provide all the empirical results you requested in such a short time. Although our method cannot outperform NashMTL on some datasets, our method still has great advantages such as **promising theoretical guarantees, better efficiency,  and better success rate in MT10.**

---

> > > > > > > > ### Comment · Reviewer_81UB · 2023-08-20
> > > > > > > >
> > > > > > > > Thanks for further clarification. As mentioned before, I appreciate the theoretical contribution of this paper and your efforts in providing more experimental results in the rebuttal. But there are many significant problems in this paper.
> > > > > > > >
> > > > > > > > 1. The implementation is highly inconsistent with the proposed algorithm presented in the paper, and the results show the proposed method is ineffective.
> > > > > > > >
> > > > > > > >  - As you claimed and the results show, $\rho=0$ performs similarly to a small positive $\rho>0$, which means the proposed regularization term is ineffective in practice.
> > > > > > > >
> > > > > > > >  - The most important thing in this paper is to ensure an unbiased estimate of the task weight $\omega$. To achieve it, a two mini-batch data strategy (i.e., sample data $\xi, \xi^{'}$ in Line 4 of Algorithm 1) is needed to guarantee the equation between Lines 175 and 176 is held. Besides, as shown in Algorithm 1, new $\xi, \xi^{'}$ are sampled in each SGD step in each training iteration. However, both of them are not implemented. In your code, a mini-batch is sampled in each iteration and used in all SGD steps, which means the task weight $\omega$ is still biased.
> > > > > > > >
> > > > > > > >     Besides, as you claimed, the same and different mini-batches achieve similar performance, which also means the two mini-batch data strategy is ineffective in practice.
> > > > > > > >
> > > > > > > >  - Without these two components (or with these two empirically ineffective components), the proposed algorithm is just used SGD to solve Eq. (8), which is very similar to MGDA and CAGrad.
> > > > > > > >
> > > > > > > >   - In your code, the gradient normalization trick is used on NYUv2, Cityscapes, MT10, and MT50 datasets. But CAGrad uses it on MT10 and MT50, not NYUv2 and Cityscapes.
> > > > > > > >
> > > > > > > >  - You can implement $\rho>0$ and the two-mini-batch strategy in the code in the revision, but the results you have provided still cannot be demonstrated their effectiveness.
> > > > > > > >
> > > > > > > > 2. Baselines and Results.
> > > > > > > >
> > > > > > > >  - Other baselines like MoCo also have hyperparameters. Are their results with the best performance drop or the best training loss?
> > > > > > > >
> > > > > > > >  - If you think "SDMGrad performs *quite comparable to or slightly worse* than NashMTL" (in terms of $\Delta_m$, Nash-MTL 6.82 vs. SDMGrad 8.39 on Cityscapes, -4.04 vs. -2.58 on NYUv2), how about the success rates and running times on MT10 (Nash-MTL 9.9s vs. SDMGrad-OS 9.7s, Nash-MTL 0.80 vs. SDMGrad-OS 0.82)?
> > > > > > > >
> > > > > > > >  - > In addition, from Fig. 4 in the FAMO paper, the time cost of NashMTL increases dramatically when the number of tasks increases, and is much worse than other baselines when the number of tasks is large.
> > > > > > > >
> > > > > > > >  	No experiments can show the proposed method is still more efficient than NashMTL when the number of tasks is large. Thus, this clarification is meaningless.
> > > > > > > >
> > > > > > > > 3. SDMGrad-OS.
> > > > > > > >
> > > > > > > >  - I appreciate SDMGrad-OS has theoretical guarantees while CAGrad-Fast does not.
> > > > > > > >
> > > > > > > >  - The basic idea of CAGrad-Fast and SDMGrad-OS is too similar, but CAGrad-Fast does not be discussed and compared in the paper. What is more, as you replied in the rebuttal, it seems you clearly knew CAGrad-Fast before.
> > > > > > > >
> > > > > > > >  - Why SDMGrad-FAST is slower than SDMGrad-OS?
> > > > > > > >
> > > > > > > >
> > > > > > > > Again, I appreciate the theoretical contribution of this paper. But I do not appreciate that a carefully-designed method is useful for theoretical analysis but is not implemented or even ineffective in practice. For example, $\rho$ and sampling two mini-batch $\xi, \xi^{'}$ in each SGD step in each training iteration, especially the latter, since it ensures the task weight $\omega$ is unbiased and it is the core contribution of this paper.

---

### Official Review · Reviewer_5qpL · 2023-06-29

**Soundness:** 4 excellent
**Presentation:** 4 excellent
**Contribution:** 3 good
**Rating:** 6
**Confidence:** 4

**Summary:**

The contributions are as follows:
- First, this work gives a new framework of direction-oriented multi-objective optimization.
- Second, they propose an algorithm, SDMGrad (and an objective sampling version when the number of objectives is large)
- Third, they give a convergence analysis for their algorithm and show that it has improved complexity compared to previous work
- Finally, they show good empirical performance on both supervised learning (semantic segmentation + depth estimation) and reinforcement learning (robot manipulation task) paradigms

**Strengths:**

Overall, the paper is clearly written, the new algorithm is straightforwardly effective, and the convergence rates beat existing bounds with fewer convergence assumptions. More specifically:
- The proposed formulation is a generalization of existing algorithms, CAGrad and SGD
- Algorithm 1 is intuitive to understand and memory-efficient to implement
- The convergence rate beats the previous MoCo by epsilon^2, and even is able to handle the case where the number of objectives is large (via the objective sampling version)
- Experiments are conducted well and show the effectiveness of the method in practice

**Weaknesses:**

- I am unsure about the applicability of the newly proposed formulation in the context of multi-objective optimization. Could the authors share some examples where this direction based regularization would be useful?
- Also, I am not sure how the -1/2||d||^2 term will affect the regularization, as it seems added for the convenience of the algorithm and convergence proof.
- Combining the above two points, properly choosing $$\lambda$$ also seems like it is important but would add an extra hyperparameter (making things more costly, empirically). For the examples you could give above, can you also share how this parameter would be chosen?
- I believe the work would benefit from a short discussion of the theorems and proof challenges in the main text. Currently, there are four theorems listed with little discussion of any proof details.

**Questions:**

- Could you outline, in intuitive terms, a sketch of the proof of Theorem 1? Is the proof inspired by any previous work (such as MoCo or CR-MOGM)?
- Is there any chance to remove or relax the bounded gradient assumption?
- Why can the convergence rate (of both your algorithm and MoCo) improve so much with a bounded function assumption?
- Do you have any sense how things would change if you remove the smoothness assumption?

**Limitations:**

No limitation section is included.

---

> ### Author Rebuttal · Authors · 2023-08-09
>
> **Q1. Could the authors share some examples where this direction based regularization would be useful?**
>
> **A:** This direction based regularization would be useful if one target is to minimize a specific objective function, e.g., the average loss in MTL. A more specific example is provided in A.1 of the appendix, where the goal is also to find the minimizer (the black point in the Pareto front) of the average loss of two tasks. From Fig. 2, it can be seen that MGDA without such regularization converges to different points in the Pareto front when starting from different points. As a comparison, our method can converge to this target black point from different initializations. This supports the importance of our regularization.
>
> **Q2.  Also, I am not sure how the -1/2$\\|d\\|^2$ term will affect the regularization, as it seems added for the convenience of the algorithm and convergence proof.**
>
> **A:** The regularization $-1/2||d||^2$ is necessary here to ensure the boundedness of the magnitude of the direction $d$. Without this regularizer, it can be seen that eq. (6) that the maximizer $d^*$ can go to infinity, and hence makes the problem meaningless. This regularizer is not new, and has also been used in works such as MGDA, CR-MOGM [1], etc.
>  [1] Zhou, Shiji, et al. "On the convergence of stochastic multi-objective gradient manipulation and beyond." Advances in Neural Information Processing Systems 35 (2022): 38103-38115.
>
> **Q3. Choosing $\lambda$ also seems like it is important but would add an extra hyperparameter (making things more costly, empirically). For the examples you could give above, can you also share how this parameter would be chosen?**
>
> **A:** From the formulation of our proposed SDMGrad, we know that it becomes close to SGD when $\lambda$ is large, and close to MGDA with a small $\lambda$. We first try to identify the large and small values of $\lambda$ where the performance shows consistency with our formulation. In experiments, we find that $\lambda=0.1$ and $\lambda=10$ work well. Next, we narrow the range by trying different values in [0.1, 10]. Specifically, we search with $\lambda \in [0.1, 1, 2, 5]$ and evaluate which choice is better. Overall, this grid search takes small efforts and the performance improvement is robust to certain ranges (e.g., [0.1,1] used in our search) within $[0.1,10]$.
>
> **4. Currently, there are four theorems listed with little discussion of any proof details. Could you outline, in intuitive terms, a sketch of the proof of Theorem 1? Is the proof inspired by any previous work (such as MoCo or CR-MOGM)?**
>
> **A:** Great suggestion! We summarize our proofs as following three main steps:
> 1. Characterization of the last-iterate convergence of SGD in solving eq. (9).
> 2. Use Proposition 1 to upper-bound the bias of the update vector $d$. The main step here is to use an intermediate quantity $G(\theta_t)w_{t, \rho, \lambda}$ to split the bound into the error in solving eq. (9) and the bias induced by smoothing.
> 3. Combine the previous two steps with a decent lemma of each objective to derive the final convergence.
>
> We will provide a proof sketch in the revision.
>
> **Q5. Is there any chance to remove or relax the bounded gradient assumption?**
>
> **A:** Interesting question! It is hard to remove this assumption given the current framework. This is because when approximating the true update direction  $d^*=G(\theta)w^*_\lambda+\lambda G_0(\theta)$, we cannot get the exact $w^*_\lambda$ but an estimate $\hat w$, and hence if the gradient $G$ is unbounded, one approximation error $\\|G(\theta)(w^*_\lambda-\hat w)\\|$ can be uncontrollable. This is why the assumption is necessary here.
>
> **Q6. Why can the convergence rate (of both your algorithm and MoCo) improve so much with a bounded function assumption?**
>
> **A:** Good question! Without bounded function assumption, we need to add a quadratic term $\rho\\|w\\|^2$ to smooth the problem w.r.t. $w$, which makes the complexity to be proportional to $\frac{1}{\rho}$ (see proof of Theorem 1). Since the smoothing factor $\rho$ is sufficiently small, the complexity becomes large. When having this assumption, we do not require such a smoothing trick, and hence improve the complexity quite a lot.
>
> **Q7.Do you have any sense how things would change if you remove the smoothness assumption?**
>
> **A:** Interesting question! First we note that solving the eq. (9) w.r.t. $w$ keeps unchanged. The main change lies in optimization w.r.t. the variable $\theta$, i.e., line 7 of our Algorithm 1. Some new challenges will arise here. For example, it is not clear if $d=G(\theta_t;\zeta)w_{t,S}+\lambda G_0(\theta_t;\zeta)$ still achieves a small bias for approximating true direction $d^*$. Decent lemma for each objective in our proof may not work here. We guess some techniques in [1] or some proximal methods may be helpful here, and we would like to leave it for future study.
>
> [1] Ohad Shamir et al. "Stochastic Gradient Descent for Non-smooth Optimization: Convergence Results and Optimal Averaging Schemes".

---

> > ### Comment · Reviewer_5qpL · 2023-08-15
> >
> > Thanks for the response! After reading other reviews and your clarifications, I have decided to maintain my original score of 6.

---

> > > ### Author Response · Authors · 2023-08-15
> > > **Thanks for the feedback**
> > >
> > > We thank the reviewer 5qpL for the feedback! We will take your suggestions in our revision!
> > >
> > > Best,
> > > Authors

---

> > > > ### Comment · Reviewer_5qpL · 2023-08-20
> > > >
> > > > I have been following the discussion between authors and reviewer 81UB. Actually, I completely agree with the points that they mentioned about the empirical experiments. In my initial review, I didn't carefully look into the exact methodology of the experiments, but after rereading and looking at the code, I agree with the criticisms that are mentioned, and believe this experimental section, as written, has many flaws.
> > > >
> > > > However, I still appreciate the theoretical contributions and find that they are enlightening and interesting. I would have been happy to accept this paper without any experimental results, but the problems with the experiments make me less willing to want to accept the paper as is. There is such a mismatch between theory and experiments that the experiments honestly don't really add much to the paper. From my perspective, this work should only be accepted if the experiments are moved to the Appendix, or removed altogether.
> > > >
> > > > I will not lower my score for now, but wanted to make this note.

---

> > > > > ### Comment · Reviewer_81UB · 2023-08-21
> > > > >
> > > > > Dear Reviewer 5qpL,
> > > > >
> > > > > Thanks for agreeing with my concerns about this paper. In my view, there is not a simple mismatch between theory and experiments but a major mistake in principle, although this paper provides some valuable theoretical results. Firstly, as you agreed, the implemented code highly mismatches the proposed algorithm. Secondly, the authors also provided the results with (possibly) correct implementation in the rebuttal, while those results show the proposed algorithm is ineffective in practice (i.e., the results of using the proposed method are similar to the ones not using, as the authors repeatedly emphasized). Note that many experiments are conducted on five commonly-used multi-task benchmark datasets. However, the authors do not clarify the inconsistent implementation and the ineffective results in the paper, which will cause a big misunderstanding for readers.
> > > > >
> > > > > I appreciate the theoretical contributions of this paper. But I kindly remind you that NeurIPS is a machine learning conference instead of a mathematical journal. Thus, it is necessary to evaluate the theoretical results and clearly show the advantages, such as weaker assumption and better convergence rate, on a toy example at least and on real-world applications if possible. However, this paper fails. The toy example in Appendix cannot demonstrate the advantages of the proposed method over MoCo, as the authors recognized in the rebuttal. As for the evaluation on the real-world applications, there are also many problems (please refer to the discussion in the rebuttal for detail).
> > > > >
> > > > > I believe this paper can be accepted after a major revision (such as adding a better evaluation with a matched implementation on the toy example and removing the experiment part as you suggested) in the future. But for now, I insist on rejecting this paper.
> > > > >
> > > > > Best, Reviewer 81UB

---

> > > > > > ### Comment · Reviewer_5qpL · 2023-08-21
> > > > > >
> > > > > > Thanks for your reply. To me, the theoretical ideas and results themselves already have merit. I believe NeurIPS can and should be a venue to share nice theoretical ideas without the need for experimental rigor, but if the experiments are there, I'd want them to be rigorous.
> > > > > >
> > > > > > Historically, I find that theory papers accepted to NeurIPS have often had poor/incomplete experiments that don't match the theory. I agree it shouldn't be this way, but the bar for NeurIPS is not high, and I feel it would be unfair for this work to be overly scrutinized for this reason.
> > > > > >
> > > > > > I am on the same page with you about the problems in the evaluation. My take is to accept this paper if and only if the authors can provide a clear revision plan that would highlight the mismatches and the caveats in the experiments (ideally putting them all in the Appendix).

---

> > > > > > > ### Comment · Reviewer_81UB · 2023-08-22
> > > > > > >
> > > > > > > Thanks for your reply. I totally agree with you that some theory papers may have poor experimental results but are still valuable. As I emphasized, this paper has a major mistake in principle. In other words, as Reviewer QQfF mentioned, the authors should *honestly* report the experimental results with a matched implementation. In my view, it must be finished in the initial submission rather than the camera-ready version. Otherwise, if no reviewer points out this problem in the review period, it will cause a big misunderstanding to reviewers and readers (after accepting), and even lead to a negative impact on the conference and community.

---

### Official Review · Reviewer_ZeY5 · 2023-06-29

**Soundness:** 3 good
**Presentation:** 3 good
**Contribution:** 3 good
**Rating:** 5
**Confidence:** 3

**Summary:**

Authors presented a new stochastic gradient method for multi-objective optimization (MOO) problem, called SDMGrad.
Compared with previous SMG, they claimed SDMGrad dose not need to increase the batch size linearly with the iteration numbers.
Compared with previous MoCo, they claimed SDMGrad could be applied into situations with large number of objectives.

**Strengths:**

The authors presented a new stochastic gradient method for multi-objective optimization (MOO) problem, called SDMGrad.
Detailed background is provided to help understand the history of MOO.
And various experiments show that the new method can outperform the previous methods.
But I need to ask some questions to fully understand the novelty of this method.

**Weaknesses:**

I need to ask some questions to fully understand the novelty of this method.
Also, some technically details are presented very clear, I will also post some questions.

**Questions:**

1. Could you detail the advantages and novelty of SDMGrad over MoCo?
First, Line 5 in Algorithm 1 seems similar with Equation 4 of MoCo.
Second, authors commented MoCo "a relatively strong assumption that the number T of iterations is much larger than the number K of objectives." Well, in the real industrial systems, the number T of iterations could be millions, and the number K of objectives (maybe 3 or 10?) is obviously much smaller than T. So this is a natural assumption to me and hence weaken the necessity to improve MoCo from this perspective.

2. Why the equation in the Line 5 of Algorithm 1 is an unbiased estimation? (Line 175)

3. The equation after line 175 is the derivative of Equation 9, right? Why we need two different data samples: \xi and \xi\prime?
and how do we implement this in the mini-batch training? sample two mini-batches in each s?

4.  Line 5 in Algorithm 1, why we need \prod_{W}? what is the meaning of this? In Equation 3, W is a set with infinite possibilities of w. How does this multiply work? And this is element-wise multiple?




**Limitations:**

No social impact.

---

> ### Author Rebuttal · Authors · 2023-08-09
>
> **Q1. Could you detail the advantages and novelty of SDMGrad over MoCo? First, Line 5 in Algorithm 1 seems similar with Equation 4 of MoCo. Second, authors commented MoCo "a relatively strong assumption that the number T of iterations is much larger than the number K of objectives." Well, in the real industrial systems, the number T of iterations could be millions, and the number K of objectives (maybe 3 or 10?) is obviously much smaller than T. So this is a natural assumption to me and hence weaken the necessity to improve MoCo from this perspective.**
>
> **A:** Good question! After reading MoCo paper carefully, we guess that you refer to eq. (10) of the ArXiv version of MoCo paper.
> For the first question, line 5 of our method and eq. (10) of MoCo have some key differences. In eq. (10) of MoCo, an auxiliary tracking variable $Y_k$ is used as a stochastic estimator for the full gradient $G(\theta_k)$, and the direction bias $\\|d(x_k)-Y_kw_k\\|$ can be shown to decrease iteratively. As a comparison, our approach directly uses the stochastic gradients $G(\theta_t; \xi), G(\theta_t; \xi^\prime)$ in $w$ updates, and show that the direction bias is sufficiently small at each iteration.
> For the second question, we wanted to claim that the assumption that $T$ is large at an order of $K^{10}$ is relatively strong. For example, $T$ can be as large as $10^{10}$ even though we choose a relatively small $K=10$. We will make this clear in the revision.
>
> **Q2.  Why the equation in the Line 5 of Algorithm 1 is an unbiased estimation? Why we need two different data samples: \xi and \xi\prime? and how do we implement this in the mini-batch training? sample two mini-batches in each s?**
>
> **A:** Great questions! Since we use two independent sample data $\xi$ and $\xi^\prime$, $G(\theta_t;\xi)$ and $G(\theta_t; \xi^\prime)w_{t,s}+\lambda G_0(\theta_t;\xi^\prime)$ are independent w.r.t.~$\xi,\xi^\prime$. Based on the fact that $\mathbb{E}[AB]=\mathbb{E}[A]\mathbb{E}[B]$ if A and B are independent, we can get
>
> $\mathbb{E}[G(\theta_t;\xi)^T(G(\theta_t; \xi^\prime)w_{t,s}+\lambda G_0(\theta_t; \xi^\prime))+\rho w_{t,s}] = \mathbb{E}[G(\theta_t;\xi)^T]\mathbb{E}[G(\theta_t; \xi^\prime)w_{t,s}+\lambda G_0(\theta_t; \xi^\prime)]+\rho w_{t,s}$
>
> $=G(\theta_t)^T(G(\theta_t)w_{t,s}+\lambda G_0(\theta_t))+\rho w_{t,s}$. Therefore, it is an unbiased estimation w.r.t. data sampling. Then, if there is one sample data $\xi$, then $G(\theta_t; \xi)$ and $G(\theta_t; \xi)w_{t,s}+\lambda G_0(\theta_t; \xi)$ have correlation with each other, and hence it could lead a biased estimation. This is the reason why we use double sampling in the gradients.
> In experiments, we found that using two different mini-batches performs similarly to using the same mini-batch. Thus, for simple implementation, we use the same mini-batch for gradient constructions. We also provide additional experiments with two different mini-batches Table 1 shown in above to support this observation.
>
> **Q3. Line 5 in Algorithm 1, why we need \prod_{W}? what is the meaning of this?**
>
> **A:** This notation $\prod_{\mathcal{W}}$ denotes the projection on the probability simplex. We will clarify this in the revision.

---

> > ### Comment · Reviewer_ZeY5 · 2023-08-16
> > **following questions**
> >
> > Thanks for the reply of authors!
> >
> > I just have a few more questions to confirm if my understanding is right and to determine if a rating change is needed:
> >
> > (1) The motivation of stochastic MOO algorithms?
> > Why we want stochastic MOO? because the true gradient or full gradient is hard to get (as claimed in MoCo)? What is the definition of true gradient or full gradient?
> >
> > I have this question because MGDA and CAGrad has been introduced and worked before these stochastic MOO algorithms, I wonder how did MGDA or CAGrad to get objective gradients? maybe they just use the gradient of a mini-batch? hard to believe MGDA or CAGrad was using a full gradient (the gradient of the whole dataset)?
> >
> > (2) To understand this paper, I go back to read MoCo and seems MoCo is to design an unbiased estimate of the model parameter's gradient; and this paper is to design an unbiased estimate of the gradient of the task weight (w), am I right?
> >
> > (3) In previous methods like MGDA and CAGrad, how did they solve w from equation (8)? this paper is the first one that use SGD-style to solve w?

---

> > > ### Author Response · Authors · 2023-08-17
> > >
> > > Thanks for the further questions
> > >
> > > Dear Reviewer ZeY5,
> > >
> > > We thank you a lot for the feedback and the additional questions! Our further responses are listed below.
> > >
> > > **Q1. The motivation of stochastic MOO algorithms? Why we want stochastic MOO? because the true gradient or full gradient is hard to get (as claimed in MoCo)? What is the definition of true gradient or full gradient?
> > > I have this question because MGDA and CAGrad has been introduced and worked before these stochastic MOO algorithms, I wonder how did MGDA or CAGrad to get objective gradients? maybe they just use the gradient of a mini-batch? hard to believe MGDA or CAGrad was using a full gradient (the gradient of the whole dataset)?**
> > >
> > > **A:** Great question! The biggest motivation for studying the stochastic MOO is that full-batch gradient (which is calculated using all data samples) requires large memory or computation and may be computationally infeasible in large-sample scenarios. In MGDA and CAGrad works, they describe and analyze their methods in the deterministic case using full gradients, but, as also noted by the reviewer, use the mini-batch sampling in the implementations. However, there is no theoretical guarantee for their approaches in the stochastic case. In fact, there are some theoretical works (e.g., [1] and MoCo paper) showing some counterexamples under which MGDA and CAGrad cannot converge to a Pareto stationary point. In addition, there are some empirical evidences such as Fig. 2 in [1] and Fig.3 in MoCo paper showing such divergence behaviors.
> > >
> > > [1] Suyun Liu and Luis Nunes Vicente. "The Stochastic Multi-gradient Algorithm for Multi-objective Optimization and its Application to Supervised Machine Learning."
> > >
> > >
> > > **Q2. To understand this paper, I go back to read MoCo and seems MoCo is to design an unbiased estimate of the model parameter's gradient; and this paper is to design an unbiased estimate of the gradient of the task weight (w), am I right?**
> > >
> > > **A:** Exactly! In MoCo, the tracking variable $Y_k$ is used as a stochastic estimator for the full gradient $G(\theta_k)$, and the direction bias $\\|d(x_k)-Y_kw_k\\|$ can be shown to decrease iteratively. Our approach is to design an unbiased estimate of the gradient of the task weight (w), which in turn leads to a near-unbiased multi-gradient estimation at each iteration.
> > >
> > > **Q3. In previous methods like MGDA and CAGrad, how did they solve w from equation (8)? this paper is the first one that use SGD-style to solve w?**
> > >
> > > **A:** Good question! In CAGrad, they solved the constrained problem in eq. (7), which designed a constraint to regularize the update direction close to the average gradient. It is equivalent to solving the Lagrangian of this constrained problem (see page 3 in CAGrad paper), where they used projected gradient descent to solve $w^*$ from the minimization problem $w^*=\arg\min_{w\in\mathcal{W}} g_w^Tg_0+\sqrt{\phi}\\|g_w\\|$ with $\phi=c^2\\|g_0\\|^2$. However, their theoretical guarantee highly relies on the access to the full gradients, and in the stochastic setting, the analysis is not clear because the norms $\\|g_w\\|\\|g_0\\|$ complicates the design of unbiased gradient/multi-gradient estimates. Similarly, for MGDA, they used projected gradient descent to solve eq.(3) in our paper and achieved guaranteed convergence in the deterministic case.
> > > However, in the stochastic case, MoCo paper provides a counterexample (see Section 2.3 in MoCo paper), which shows that directly using mini-batch gradients in MGDA leads to a **biased** multi-gradient estimate.
> > >
> > > To the best of our knowledge, our work is the first one to use SGD with near-unbiased gradient estimates to solve $w$.

---

> > > > ### Comment · Reviewer_ZeY5 · 2023-08-18
> > > > **more reply**
> > > >
> > > > Q3:
> > > > https://github.com/Cranial-XIX/CAGrad seems in the implementation of CAGrad, they also use SGD to solve w?

---

> > > > > ### Author Response · Authors · 2023-08-19
> > > > >
> > > > > **Q. https://github.com/Cranial-XIX/CAGrad seems in the implementation of CAGrad, they also use SGD to solve w?**
> > > > >
> > > > > **A:** Thank you for the question. It is noted that CAGrad only uses the stochastic method to update $w$ on reinforcement learning experiments in real implementation. However, there is no theoretical guarantee since their theory is conducted in deterministic settings. It is possible that CAGrad can not converge to the Pareto optimal solution for stochastic problems. For example, [1] provides an example showing that parameters actually move away from Pareto set using CAGrad in the stochastic settings. To be more precise, our work is the first one to provide a theoretical guarantee by using SGD with near-unbiased gradient estimates to solve $w$.
> > > > >
> > > > > Additionally, in their SGD implementation, they use softmax to project $w$ onto the probability simplex, which is different from our choice of Euclidean projection. Take the 2-task case as an example. Suppose the task weights now are [0.75,0.25], which already satisfies the simplex constraint. The task weights keep unchanged after the Euclidean projection. However, the task weights will be [0.62,0.38] after softmax projection.
> > > > >
> > > > > [1] Zhou S, Zhang W, Jiang J, et al. On the convergence of stochastic multi-objective gradient manipulation and beyond

---

### Official Review · Reviewer_tz6S · 2023-06-30

**Soundness:** 2 fair
**Presentation:** 3 good
**Contribution:** 2 fair
**Rating:** 4
**Confidence:** 4

**Summary:**

This paper proposes a direction-oriented multi-objective gradient descent algorithm under stochastic gradient settings. The authors show that the algorithm can benefit from the direction-oriented mechanism and ensure optimal convergence. In addition, an objective sampling strategy is applied to the proposed algorithm for the case of many objectives.

**Strengths:**

This paper studies an important problem in multi-objective optimization. It has solid analysis and experiments.



**Weaknesses:**

1. The key benefit of MoCo compared with CR-MOGM is that the direction can gradually be unbiased to the one calculated by full-batch gradients (Lemma 2). However, there is no respective analysis of the direction bias in the analysis.

2. I check that CR-MOGM has the same sample complexity as SDMGrad in the non-convex setting. Note that $O(T^{-1/4})$ in CR-MOGM is for the first-order convergence of $\|\|G_t w_t\|\|$, and for the second-order convergence $\|\|G_t w_t\|\|_2^2$ is $O(T^{-1/2})$. Usually, the first-order one is more often used.

3. The idea is very similar to [1], which studies multi-objective online learning that can be easily reduced to stochastic settings. [1] also uses a regularization to restrict the direction to be closed to a prior preference.

4. CR-MOGM has not been compared in the experiments.

5. I am not convinced with the motivation to face many objectives problems as stated in Line 46. It is known that the many objective problem is hard to be solved by Pareto optimization, since when the number of objectives becomes too large, the Pareto set will cover nearly the majority of the space, then optimizing towards Pareto optimality is meaningless.


[1] Jiang et al. Multi-Objective Online Learning. ICLR 23.

**Questions:**

No

**Limitations:**

Novelty is limited.

---

> ### Author Rebuttal · Authors · 2023-08-09
>
> **Q1. There is no respective analysis of the direction bias in the analysis.**
>
> **A:** Our analysis of the direction bias can be found in Proposition 1. It can be seen that the bias is upper-bounded by an exponentially decaying term $4C_g^2(1-2\beta_t\rho)^S$ plus two small terms $\frac{\beta_tC_g^2C_1}{\rho}$ and $\rho$. This error bound is controllable by selecting stepsize $\beta_t$ and smoothing constant $\rho$ properly small.
>
> **Q2. CR-MOGM has the same sample complexity as SDMGrad in the non-convex setting.**
>
> **A:** Thanks for pointing this out for us! We have double checked the sample complexity of CR-MOGM and the second order convergence is indeed $\mathcal{O}(T^{-\frac{1}{2}})$. We will revise it accordingly.
>
> **Q3. The idea is similar to [1].
> [1] Jiang et al. Multi-Objective Online Learning. ICLR 23.**
>
> **A:** Thanks for pointing this paper out for us! However, we believe our work has substantial differences from this work:
>
> 1. First, we focus on different objective functions. [1] studied the multi-objective online convex optimization problem with convex objective functions, whereas we focus on the stochastic multi-objective optimization with nonconvex objective functions.
>
> 2. Second, our regularizer is different from theirs. In specific, the $l_1$-regularizer in [1] is to use time-varying historical information for stabilizing the algorithm performance, whereas our regularizer $\lambda\langle g_0, d\rangle$ is to regularize the update direction $d$ close to a fixed direction $g_0$.
>
> 3. Third, algorithms are different. The method in [1] is motivated by mirror descent, whereas ours is an SGD-type approach with a double-sampling scheme.
>
> We will cite this relevant paper in the revision and provide a detailed discussion for comparison.
>
> **Q4. CR-MOGM has not been compared in the experiments.**
>
> **A:** The official codes of CR-MOGM have not been released, so we implement them by ourselves. We provide the preliminary results of CR-SDMGrad on Cityscapes and NYU-v2 in Table 1-2 in global response. It can be seen that our method performs comparably to CR-SDMGrad on Cityscapes, but on NYU-v2, our method is significantly better.
>
> **Q5. I am not convinced with the motivation to face many objectives problems as stated in Line 46. It is known that the many objective problem is hard to be solved by Pareto optimization, since when the number of objectives becomes too large, the Pareto set will cover nearly the majority of the space, then optimizing towards Pareto optimality is meaningless.**
>
> **A:** Sorry for the confusion and let us clarify the motivation here. In lines 44-46, we mentioned that MoCo needs an assumption that the number $T$ of iterations is large at an order of $K^{10}$, where $K$ is the number of objectives. This is a strong assumption even for a small number $K=10$ (in this case, $T$ is as large as $10^{10}$). Then, our motivation is to remove this requirement rather than the need to face many objectives. We will clarify this in the revision.

---

> > ### Comment · Area_Chair_FdG5 · 2023-08-21
> > **Thanks for the rebuttal.**
> >
> > Although the review did not engage, I'll carefully read and consider it during the decision period.
> >
> > AC

---

> ### Author Response · Authors · 2023-08-15
> **Look forward to the reviewer's feedback**
>
> Dear Reviewer tz6S,
>
> Thanks so much for your time and efforts in the review. While the discussion period has started for a while, we have not received your feedback on our response. We really appreciate it if you can let us know whether our response resolves your concerns (e.g., comparison to CR-MOGM) or not. Your further comments and questions are appreciated!
>
> Best,
> Authors

---

### Official Review · Reviewer_ft1r · 2023-07-06

**Soundness:** 3 good
**Presentation:** 3 good
**Contribution:** 2 fair
**Rating:** 6
**Confidence:** 3

**Summary:**

This paper introduces the stochastic direction-oriented multi-objective gradient descent (SDMGrad) and SDMGrad-OS (OS stands for objective sampling). The idea of SDMGrad is to make the objective “direction-oriented”, which is done by regularizing the descent direction towards a specific direction. This direction is chosen to be the average of all gradients, which would allow the regularizer to interpolate the algorithm between multiple-gradient descent and regular gradient descent. This formulation is similar to CAGrad, but results in a simpler stochastic analysis and a simpler algorithm. The experiments show that, for a well-tuned regularization parameter, SDMGrad’s performance is competitive with the state-of-the-art. The objective sampling procedure in SDMGrad-OS makes it run significantly faster than SDMGrad when the number of objectives is large. Analysis for both methods is provided with good sample complexities.

**Strengths:**

- The idea is clear, the formulation is sound, and the algorithm is simple to implement.
- The formulation allows for a better stochastic analysis and simpler algorithm. The sample complexity achieved by the algorithm is good.
- SDMGrad generalizes MGDA and GD, and is a better formulation of CAGrad.
- The code is available, so experiments can be reproduced.
- Performance is good overall, and can be better for fine-tuned lambda.
- Objective sampling helps when the number of objective is large. The analysis also justifies this procedure, which is a good contribution.

**Weaknesses:**

- Need to tune lambda to get better results. It does not seem to be easy to pick a good starting point as the algorithm is not very robust to the choice of lambda. Also, it is not clear how to choose $\rho$.
- The algorithm interpolates between MGDA and GD (with $w$ regularization), so the algorithmic novelty is limited.
- In equation 4, $h_{t,i}$ is not explained.
- The improvements are not significant, and that is for the best choices of $\lambda$. For other choices of $\lambda$, it might be worse. It seems to be on par with CAGrad and MoCO, which is competitive.
- In the appendix, the authors mention Young's inequality for some steps, but I think it's Cauchy-Schwartz, though I might be mistaken.
- Note that the Lipschitzness (bounded gradient) assumption of $L_i$ would not work with strongly convex losses.

**Questions:**

- Which method do you use to project the weights onto the simplex? Is the projection costly?
- Have you considered relaxing the simplex constraint? Would it make sense to use the closed-form minimum $w$ for unconstrained $w$?
- Have you considered a different regularizer for $w$? How do you set $\rho$ in practice?

**Limitations:**

None.

---

> ### Author Rebuttal · Authors · 2023-08-09
>
> **Q1.  Need to tune lambda to get better results. It does not seem to be easy to pick a good starting point as the algorithm is not very robust to the choice of lambda. Also, it is not clear how to choose $\rho$.**
>
> **A:** Good question! From the formulation of our proposed SDMGrad, we know that it becomes close to SGD when $\lambda$ is large, and close to MGDA with a small $\lambda$. We first try to identify the large and small values of $\lambda$ where the performance shows consistency with our formulation. In experiments, we find that $\lambda=0.1$ and $\lambda=10$ work well. Next, we narrow the range by trying different values in [0.1, 10]. Specifically, we search with $\lambda \in [0.1, 1, 2, 5]$ and evaluate which choice is better. Overall, this grid search takes small efforts and the performance improvement is robust to certain ranges (e.g., [0.1,1] used in our search) within $[0.1,10]$.
>
> The parameter $\rho$ is used for theoretical analysis and guarantees that the proposed method works well in the worst case. In the experiments (which are not necessarily the worst case), we find the performance when $\rho=0$ is good enough. Thus, we make this choice for simple implementation.
>
> **Q2. The algorithm interpolates between MGDA and GD (with w regularization), so the algorithmic novelty is limited.**
>
> **A:** Let us explain the algorithmic novelty of our method as follows.
> First, our method has substantial differences from existing stochastic MGDA-type methods. For example, MoCo constructs an additional auxiliary sequence $Y_k$ to approximate the gradients in MOO with decreasing estimation error, whereas our approach leverages a double-sampling mechanism that admits a near-unbiased multi-gradient estimation at each iteration. This type of approach is new in the literature.
> Second, our regularization is new and contains careful designs. Compared to the most relevant CAGrad method that uses a constraint to regularize the update direction close to $g_0$, our regularization not only enjoys such direction-oriented benefit, but also admits a provable algorithmic design in the challenging stochastic setting.
> Third, our double-sampling-based approach is flexible to incorporate the objective sampling for better efficiency. To the best of our knowledge, this is the first result for a stochastic MOO algorithm with objective sampling.
>
> **Q3. What is $h_{t,i}$ in equation 4?**
>
> **A:**  $h_{t,i}$ is defined in eq. (6) of MoCo paper in the ICLR version, where $h_{t,i}$ denotes a stochastic estimator of $\nabla L_i(\theta)$ at t-th iteration. We will clarify this in the revision.
>
> **Q4. In proof, Young’s inequality or Cauchy-Schwarz.**
>
> **A:** Thanks! We will double check the proofs and make the revisions accordingly.
>
> **Q5. Note that the Lipschitzness (bounded gradient) assumption of $L_i$ would not work with strongly convex losses.**
>
> **A:** Note that in our setting,  the function $L_i(\theta)$ is generally nonconvex rather than strongly convex. The problem in eq. (9) is strongly convex w.r.t. $w$ rather than $\theta$.
>
> **Q6. Which method do you use to project the weights onto the simplex? Is the projection costly?**
>
> A: The computation of the projection to the probability simplex we use is the Euclidean projection proposed by [1], which involves solving a convex problem via quadratic programming. The implementation used in our experiments follows the repository in [2], which is very efficient in practice.
> [1] Weiran Wang, and Miguel Á. Carreira-Perpiñán. Projection onto the probability simplex: An efficient algorithm with a simple proof, and an application. arXiv preprint arXiv: 1309.1541
>
> [2] Adrien Gaidon. Compute Euclidean projections on the simplex or L1-ball.
>
> **Q7. Have you considered relaxing the simplex constraint? Would it make sense to use the closed-form minimum w for unconstrained w?**
>
> **A:** Good question! It is possible to use the closed-form minimum $w$ if the full gradients are used. For example, for the two-objective example provided by the MoCo paper (see eq. (5) therein), a closed-form solution is provided. This solution is accurate when full gradients are taken. However, when only stochastic gradients are available, they show that a large bias can be induced. Thus, it may be challenging to apply this idea in the stochastic setting, which is the focus of this paper. However, we would like to leave such exploration for future study.
>
> **Q8. Have you considered a different regularizer for w? How do you set it up $\rho$ in practice?**
>
> **A:** Great question! Since the main purpose of the regularizer for $w$ is to make the problem in eq. (9) strongly convex such that the theoretical convergence guarantee can be established. For this reason, we use the simplest quadratic regularizer. However, it is possible to use other strongly-convex regularizers for a stronger theoretical guarantee. We would like to leave it for future study.
>
> In experiments, we tune $\rho$ and find the best range to be $[0,0.1]$. For a simple implementation, we simply set $\rho=0$ in all experiments.

---

> > ### Comment · Reviewer_ft1r · 2023-08-13
> >
> > Thank you for clarifying my concerns. I have read the rebuttals and decided to update my score accordingly.

---

> > > ### Author Response · Authors · 2023-08-13
> > > **Thanks so much for your updates**
> > >
> > > Dear Reviewer,
> > >
> > > Thanks so much for your updates and for raising your score! We will take your suggestions into our revisions!
> > >
> > > Best,
> > > Authors

---

### Official Review · Reviewer_QQfF · 2023-07-10

**Soundness:** 3 good
**Presentation:** 3 good
**Contribution:** 2 fair
**Rating:** 7
**Confidence:** 3

**Summary:**

This paper proposes stochastic variants of the MGDA for multi-objective optimization (MOO) problem, named "Stochastic Direction-Oriented Multi-objective Gradient descent" (SDMGrad) and SDMGrad-OS with efficient sampling. Optimization convergence analysis to the Pareto stationary point is provided, with improved complexities compared to prior works on stochastic MOO.
The proof of optimization convergence mainly follows that of [12].
Experiments on multi-task supervised learning and reinforcement learning justified the superior performance of the proposed method.

**Strengths:**

1. The paper proposes a stochastic variant of MGDA to address the MOO problem.

2. Optimization convergence analysis is provided with improved complexity over [12] without bounded function values assumption, and with improved complexity over [11] with bounded function values assumption.

3. Experiments on MTL benchmarks demonstrated the superior performance of the proposed methods.

**Weaknesses:**

## Soundness
### Lack of justification of some claims

See __Questions__-1,2.


### Some steps of the proof is ambiguous or unclear

See __Questions__-3,4.

### Theoretical and practical benefits of the proposed algorithm over the simple unitary scalarization (GD or SGD) baseline are unclear

1. The algorithm is not very efficient. In Theorems 1, 2, with a constant $\lambda$, it requires the inner loop iterations $S = \mathcal{O}(T^3)$, and each step of the inner loop needs to compute a projection to the simplex, which adds more time complexity to the MGDA based algorithms as they already require computing $K$ gradients at each outer iteration comparing to $1$ gradient for GD or SGD.
How does your algorithm compare to the simple SGD baseline in terms of convergence to Pareto stationary point in clock time? It would be better if some evaluations and discussions regarding this can be provided.
E.g., in Table 6 or other experiments, compare with the time and performance of the simple SGD baseline.

5. What is the theoretical benefit of this proposed algorithm compared to SGD of unitary scalarization, as the latter can also converge to Pareto stationary (PS) point and is more efficient?
For example, in [12], in addition to the convergence to PS point, Lemma 2 is provided to justify the convergence of MoCo to the desired MGDA direction, does similar results apply for the proposed algorithm also?

2. In the experiments, one baseline method with unitary scalarization is needed, as it has been shown in prior works [a,b] that unitary scalarization outperforms MGDA-based algorithms in some practical applications.
I understand that experiments with varying $\lambda$ are provided, which becomes close to GD with larger $\lambda$, but they are still not the same. Instead I would expect an ablation study with varied $\beta_t$, and especially $\beta_t = 0$.

>[a] Vitaly Kurin et al. "In Defense of the Unitary Scalarization for Deep Multi-Task Learning"

>[b] Derrick Xin et al. "Do Current Multi-Task Optimization Methods in Deep Learning Even Help?"

### Technical novelty is limited

Novelty in terms of proof techniques is limited, the proof mainly follows that of [12].

## Minor

1. Appendix C.2, line 530, "satisfies" -> "satisfy"
2. Move Table 4 to Section 6.2
3. In Appendix D.1, LHS of Equation below line 591, "$E$" -> "$\mathbb{E}$"



================UPDATE==========================

To avoid unnecessary misunderstandings, and unevidenced accusations, I change the word *honest* to *accurate* in my previous comment and also update my final comments and suggestions below.

I appreciate the reviewers carefully checking the code and I also appreciate the authors clarifying what they implemented in the rebuttal.

My current score is based on the theoretical contributions and also conditioned on the authors can correctly implement and report the results with double sampling. Otherwise it could be confusing.

=================================================

**Questions:**

1. In Section 4.2, line 170-172, why is the case "the objective in eq. (8) is merely convex w.r.t. $w$, it can only guarantee the convergence of at least one iterate (not necessarily the last one) to the solution $w_\lambda^*$"?
In fact, there are works that prove the last iterate convergence of convex objectives, see [c].
Since this serves as a motivation for the algorithm design, it is important to clarify it.
And although it claims that "to ensure the more practical last-iterate convergence" in line 172, no guarantee of convergence of objective in eq. (9) for last iterate is provided in the paper.

>[c] Ohad Shamir et al. "Stochastic Gradient Descent for Non-smooth Optimization: Convergence Results and Optimal Averaging Schemes".

2. The use of smoothing constant $\rho$ seems to be redundant since you already have another regularization term $\lambda g_0$, which makes the update direction close to $g_0$. Using the regularization term $\rho ||w||^2$ has a similar effect of making the update direction close to a uniform combination of gradients of all objectives.
Since the motivation "to ensure the more practical last-iterate convergence" is questionable, see __Question__-1,
more justification is needed for this design.

3. The expectation operations used throughout the paper and proof are very unclear.
The same notation $\mathbb{E}[\cdot]$ is used for almost all expectations, whether conditioning on $\theta_t$ or not, except for Eq. (11) in the main paper where $\mathbb{E}[\cdot\mid \theta_t]$ is used. This makes the arguments or proof steps ambiguous.
   - In Proposition 1, Equation (11), the LHS has two expectations, one conditioned on $\theta_t$, and the other not. What are the specific distributions for these two expectations? There is some ambiguity since there are multiple random variables, e.g. $\zeta, w_{t,S}, \theta_t$.
Also in the proof of Proposition 1, Appendix C.1, line 522, how is the condition on $\theta_t$ removed in the first equation of (21)?
This does not seem correct if the inner expectation is not conditioned on $\theta_t$.
    - In most part of the proof, $\mathbb{E}[\cdot]$ is used for both conditioning on $\theta_t$ and without conditioning on $\theta_t$.
    E.g. line 533 in the Appendix, Eq (24) is "conditioning on $\theta_t$", and later on in line 538 Eq(26) is "unconditioning on $\theta_t$".

1. In Appendix C.2, line 533-534, I did not see how the inequality (iii) is derived from eq. (21)?

**Limitations:**

I did not see specific discussions on the limitations and broader societal impacts.

This does not decrease my score, and it would be better if the authors provide some.

---

> ### Author Rebuttal · Authors · 2023-08-09
>
> **Q1. How does your algorithm compare to the simple SGD baseline in terms of convergence to Pareto stationary point in clock time?**
>
> **A:** For the SGD baseline, we use the implementation by [1].  The experiments on Cityscapes, NYU-v2, and MT10 are provided in  Table 1-3 in the global response PDF. It can be seen that SGD outperforms others on a specific task but achieves a worse overall performance than our SDMGrad method. The clock time comparison is provided in Table 3, where it can be seen that the SGD baseline is faster than SDMGrad but slower than SDMGrad-OS.
>
> [1] Liu, Bo, et al. "Conflict-averse gradient descent for multi-task learning." Advances in Neural Information Processing Systems 34 (2021): 18878-18890.
>
> **Q2. What is the theoretical benefit of this proposed algorithm compared to SGD of unitary scalarization, as the latter can also converge to Pareto stationary (PS) point and is more efficient?**
>
> **A:** Great question! Although SGD of unitary scalarization can achieve a faster convergence rate, our method is more flexible and general to enjoy the theoretical advantages of both SGD and MGDA. On the one hand, our method reduces to this SGD type of method for a large $\lambda$ (which achieves a much higher efficiency as shown in Corollary 2). On the other hand, for a smaller $\lambda$, our method with iteratively optimized weights can enjoy the benefits of MGDA type methods in mitigating the gradient conflict during the optimization process. Indeed, it has been shown in [2] that the weight changing methods like MGDA strike a better tradeoff among optimization, generalization, and gradient conflict than static weighting methods like SGD.
>
> [2] Chen, Lisha, et al. "Three-Way Trade-Off in Multi-Objective Learning: Optimization, Generalization and Conflict-Avoidance." arXiv preprint arXiv:2305.20057 (2023).
>
> **Q3. In the experiments, one baseline method with unitary scalarization is needed, as it has been shown in prior works [a,b] that unitary scalarization outperforms MGDA-based algorithms in some practical applications. And compare the case with $\beta_t=0$.**
>
> **A**: The comparison results of unitary scalarization [a] are shown in Tables 1 and 2 in the global response. In general, it can be seen that our SDMGrad outperforms unitary scalarization under different metrics. From Table 3 in the global response, it can be seen our SDMGrad-OS is faster than unitary scalarization due to the efficient task sampling.
> The ablation study of $\beta_t=0$ is also shown in Tables 1 and 2. It can be seen that our method with $\beta_t=0$ performs poorly in these experiments. This confirms the importance of the $w$ updates.
>
> [a] Kurin, Vitaly, et al. "In defense of the unitary scalarization for deep multi-task learning." Advances in Neural Information Processing Systems 35 (2022): 12169-12183.
>
> **Q4. In Section 4.2, line 170-172, why is the case "the objective in eq. (8) is merely convex w.r.t. w, it can only guarantee the convergence of at least one iterate (not necessarily the last one) to the solution "?**
>
> **A:** Sorry about the confusion and thanks for pointing this reference out for us! After careful checking, we find that the analysis in [c] (Theorem 2 therein) on convex functions relies on additional assumptions like bounded estimators and domains, which may be restrictive in our MOO setting. To make our claim more rigorous, we will revise our sentence to “To ensure the last-iterate convergence, one possible approach is to add a quadratic regularization term for smoothing.”
> The results for the guarantee of last-iterate convergence of objective in eq. (9) can be found in Lemma 3 of the appendix. We will clarify this in the main body.
>
> **Q5.The use of smoothing constant $\rho$ seems to be redundant since you already have another regularization term $\lambda g_0$, which makes the update direction close to $g_0$. Using the regularization term $\rho\\|w\\|^2$ has a similar effect of making the update direction close to a uniform combination of gradients of all objectives.**
>
> **A:** We want to clarify that these two regularization terms serve different purposes. The regularizer $\lambda\langle g_0,d\rangle$ makes the update direction close to $g_0$, whereas the quadratic term is to make the eq.(9) strongly convex to establish the convergence rate guarantee. In other words, the smoothing term $\rho\\|w\\|^2$ is necessary here because the regularization $\lambda\langle g_0,d\rangle$ cannot make the problem in eq. (9) strongly convex.
>
> **Q6. In Proposition 1, Equation (11), the LHS has two expectations, what are the specific distributions for these two expectations? In the proof of Proposition 1, Appendix C.1, line 522, how is the condition on $\theta_t$ removed in the first equation of (21)?  In most parts of the proof,  $\mathbb{E}[\cdot]$is used for both conditioning on $\theta_t$ and without conditioning on $\theta_t$. E.g. line 533 in the Appendix, Eq (24) is "conditioning on $\theta_t$", and later on in line 538 Eq(26) is "unconditioning on $\theta_t$".**
>
> **A:** Sorry for the confusion. In eq. (11), the inner expectation is conditioning on $\theta_t$ but the outer expectation takes the randomness over $\zeta, w_{t,S}, \theta_t$.
> In line 522, the inner expectation is indeed conditioned on $\theta_t$, and we remove the condition on $\theta_t$ based on the fact that $\mathbb{E}[\mathbb{E}[A|B]]=\mathbb{E}[A]$. We will make this clear in the proof.
> To make it clearer, we will use $\mathbb{E}[\cdot]$ to denote the expectation unconditioning on $\theta_t$, and use $\mathbb{E}[\cdot |\theta_t]$ to denote the expectation conditioning on $\theta_t$.
>
> **Q7. In Appendix C.2, Line 533-534, I did not see how the inequality (iii) is derived from eq. (21)?**
>
> **A:** Sorry for the confusion. The detailed steps can be found in the global response.

---

> > ### Comment · Reviewer_QQfF · 2023-08-17
> > **Thanks for the response**
> >
> > Thank you very much for the detailed response. It addresses most of my concerns. I have a few more questions below.
> >
> >
> > **Regarding Q1.** What is the intuition that SDMGrad-OS can be faster than SGD in Table 3? At each iteration during training, SGD needs to compute one gradient, while SDMGrad-OS needs to compute more than one (because of performing double sampling for $\xi$ and $\xi'$, as well as updating $w_ {t,s}$). Could you provide some intuitive explanation?
> >
> > **Regarding Q4.** Since here we are considering the subproblem convergence in this context, the domain of $w$ is bounded within a simplex, and the gradient estimators also seem can be bounded under Assumptions 2 and 3. Therefore I think analysis in [c] can be applied. Or are there any other challenges in applying analysis in [c]?
> >
> > Also, in Lemma 3, you provide the convergence in terms of $\mathbb{E}[||w_S-w_{\rho, \lambda}^*||^2]$, where $w_{\rho, \lambda}^*$ is the solution to the smoothed problem with $\rho > 0$, instead of the original problem with $\rho = 0$. I understand the original problem with $\rho=0$ could have multiple solutions, but ideally it would be more useful if the convergence in of $w_S$ to the solution set with $\rho = 0$ is analyzed. Is it possible to extend your analysis to the convergence to the solution set with $\rho = 0$ that matches the original problem?

---

> > > ### Author Response · Authors · 2023-08-17
> > > **Thanks for the feedback!**
> > >
> > > We thank the reviewer for recognizing our response and for the further questions. Our answers are listed as follows.
> > >
> > > **Q1. What is the intuition that SDMGrad-OS can be faster than SGD in Table 3? At each iteration during training, SGD needs to compute one gradient, while SDMGrad-OS needs to compute more than one (because of performing double sampling for $\xi$ and $\xi^\prime$, as well as updating $w_{t,s}$). Could you provide some intuitive explanation?**
> > >
> > > **A:** Thanks for the question. SDMGrad-OS can be faster than SGD for two reasons. On the one hand, SDMGrad-OS adopts the objective sampling at each iteration, whereas the SGD method, implemented in CAGrad paper, does not have this feature.  On the other hand, in our implementation, the number $S$ of iterations in updating $w$ is set to be relatively small and the gradient computation is efficient in our case. Thus, SDMGrad-OS is overall faster than SGD in Table 3.
> > >
> > > **Q2. Since here we are considering the subproblem convergence in this context, the domain of $w$ is bounded within a simplex, and the gradient estimators also seem can be bounded under Assumptions 2 and 3. Therefore I think analysis in [c] can be applied. Or are there any other challenges in applying analysis in [c]? Also, in Lemma 3, you provide the convergence in terms of $\\mathbb{E}[\\|w_S-w_{\\rho,\\lambda}^\ast\\|^2]$, where $w_{\\rho,\\lambda}^\ast$ is the solution to the smoothed problem with $\\rho>0$, instead of the original problem with $\\rho=0$. I understand the original problem with $\\rho=0$ could have multiple solutions, but ideally it would be more useful if the convergence in of $w_S$ to the solution set with $\\rho$=0 is analyzed. Is it possible to extend your analysis to the convergence to the solution set with $\\rho=0$ that matches the original problem?**
> > >
> > > **A:** Thanks for pointing it out. After a careful checking of this literature, we notice that the assumptions in [c] can be satisfied in our method. Suppose $\\hat{g}_t$ to be the stochastic gradient estimator w.r.t $w$ in our problem. The domain of $w$ is bounded within a simplex and $\\mathbb{E}[\\|\\hat{g}_t\\|]$ can also be bounded due to the fact that $\mathbb{E}[\\|\\hat{g}_t\\|]\\leq\\sqrt{\\mathbb{E}[\\|\\hat{g}_t\\|^2]}\\leq\\sqrt{3C_1}$, where $C_1$ is defined in our lemma 3 in Appendix. Thus, to solve the original question with $\rho=0$, one possible solution might use the analysis from the work that you mentioned and another possible solution would be choosing the minimum $w$ instead of the last iterate. We thank the reviewer for pointing the work [c] out for us. However, it may still need to take some time for making sure that the technique in [c] is applicable in our analysis, and we would like to leave it as part of our future work.

---

> > > > ### Comment · Reviewer_QQfF · 2023-08-18
> > > > **Applying the idea in SDMGrad-OS to other MOO algorithms**
> > > >
> > > > Thank you very much for the reply.
> > > >
> > > > The idea of SDMGrad-OS seems to be useful in practice since it can improve the efficiency of SDMGrad. Is it possible to apply this idea to other stochastic MOO algorithms such as MoCo without affecting its theoretical convergence?

---

> > > > > ### Author Response · Authors · 2023-08-19
> > > > >
> > > > > **Q. The idea of SDMGrad-OS seems to be useful in practice since it can improve the efficiency of SDMGrad. Is it possible to apply this idea to other stochastic MOO algorithms such as MoCo without affecting its theoretical convergence?**
> > > > >
> > > > > **A:** Good question! We believe our objective sampling method is applicable to other stochastic MOO algorithms. For example, in MoCo, to keep the unbiasedness of the estimate, we need to set a new unbiased estimate $h_{k,m}^\prime=\frac{K}{n}h_{k,m}$, where $n$ is the number of sampled objectives and $K$ is the total number of objectives. In this way, we guess that the theoretical convergence can also be guaranteed, but surely it requires extra effort for a careful check.

---

> > > > > > ### Comment · Reviewer_QQfF · 2023-08-19
> > > > > > **Thanks for the explanation**
> > > > > >
> > > > > > Thank you very much for the explanation.
> > > > > >
> > > > > > The principle behind SDMGrad-OS seems practically useful to improve efficiency of stochastic MOO, which overcomes a bottleneck in computation of stochastic MOO compared to the simple SGD baseline.
> > > > > >
> > > > > > I have updated my score accordingly.

---

> > > > > > > ### Author Response · Authors · 2023-08-19
> > > > > > >
> > > > > > > Dear Reviewer,
> > > > > > >
> > > > > > > Thanks so much for your updates and for raising your score! We will take your suggestions into our revisions!
> > > > > > >
> > > > > > > Best, Authors

---

### Author Rebuttal · Authors · 2023-08-09

**To all reviewers:**

We thank all reviewers for their time and valuable comments! Based on the reviewers’ suggestions, we have added the following additional experiments:
1. Comparison to additional baselines including SGD, unitary scalarization, RLW, IMTL, and NashMTL.
2. Running time comparison to unitary scalarization and MoCo.
3. Comparison to CR-MOGM.
4. Comparison between one data sample and two different data samples in the inner loop of SDMGrad.

Please refer to the attached PDF file for these experiment results.

**We also add details of how the inequality $(iii)$ is derived in Appendix C.2, Line 533-534.**

The detailed steps are shown as follows.

$\\|G(\\theta_t)w_{t,\\lambda}^*+\\lambda G_0(\\theta_t)- G(\\theta_t)w_{t,\\rho,\\lambda}^*-\\lambda G_0(\\theta_t)\\|^2$
 $=\\|G(\\theta_t)w_{t,\\lambda}^*+\\lambda G_0(\\theta_t) \\|^2+\\| G(\\theta_t)w_{t,\\rho,\\lambda}^*+\\lambda G_0(\\theta_t)\\|^2-2\\langle G(\\theta_t)w_{t,\\lambda}^*+\\lambda G_0(\\theta_t) , G(\\theta_t)w_{t,\\rho,\\lambda}^*+\\lambda G_0(\\theta_t)\\rangle$
 $\\overset{(i)}{\\leq}\\|G(\\theta)w_{t,\\rho,\\lambda}^*+\\lambda G_0(\\theta_t)\\|^2-\\|G(\\theta)w_{t,\\lambda}^*+\\lambda G_0(\\theta_t)\\|^2$

 $\\overset{(ii)}{\\leq}\\frac{1}{2}\\rho,$

where $(i)$ follows from optimality condition.
 $\\langle G(\\theta_t)w_{t,\\lambda}^*+\\lambda G_0(\\theta_t) , G(\\theta_t)w_{t,\\rho,\\lambda}^*+\\lambda G_0(\\theta_t)\\rangle$
$\\geq\\langle G(\\theta_t)w_{t,\\rho, \\lambda}^*+\\lambda G_0(\\theta_t) , G(\\theta_t)w_{t,\\rho,\\lambda}^*+\\lambda G_0(\\theta_t)\\rangle=\\| G(\\theta_t)w_{t,\\rho,\\lambda}^*+\\lambda G_0(\\theta_t)\\|^2$.

Step $(ii)$ follows from $(iv)$ in eq.(21). It uses the optimality that $\\|G(\theta)w_{t,\rho,\lambda}^*+\lambda G_0(\theta_t)\\|^2-\\|G(\theta)w_{t,\lambda}^*+\lambda G_0(\theta_t)\\|^2\leq\frac{1}{2}\rho$.

Then the inequality (iii) can be derived.

---

### Author Response · Authors · 2023-08-21
**Thank all reviewers**

Dear Reviewers,

We thank you all for the great time and efforts in the review and rebuttal process! We also appreciate the valuable comments that help us to refine our paper! After the rebuttal, we have added quite a few additional experimental results such as comparison to various baselines, running time comparison, some ablation studies, the implementations of two-mini-batch sampling and the choice of $\rho>0$. We are sorry about the concerns on our implementations. We will definitely follow the suggestions to revise our experiments and implementations.

Best,
Authors

---

### Decision · Program_Chairs · 2023-09-21

**Decision:**

Accept (poster)

**Comment:**

The paper is proposing a stochastic MOO algorithm with a theoretical analysis as well as experiments on classification and RL settings. It was reviewed by 6 reviewers and among them consensus was accepting paper. Although authors provided a rebuttal, the disagreement was still there. I carefully read the paper, reviews and the rebuttal. The major issues against acceptance was about empirical study. Specifically,

- There were minor mismatch between algorithmic description and empirical study
- Evaluation lacked some datasets/settings.

Overall, the mismatch is not a major issue as it is common to make some simplifications or engineering choices while implementing the methods. Authors clearly provided a source code and there is no re-production issue. I fail to see the importance the mismatch as both theoretical study and implementation is shared publicly. I also think we can not judge empirical study of a theoretical contribution same as empirical paper. The provided empirical study is enough to justify theoretical study (the convergence). Theory does not say anything about comparison to other methods.

It is a good theoretical contribution and should be shared with the community. I strongly recommend authors to incorporate reviewer feedback in their camera-ready.